# MITIGATING PRIVACY RISK VIA FORGET SET-FREE UNLEARNING

**Aviraj Newatia***
University of Toronto, Vector Institute
Toronto, ON, Canada
`avinewatia@cs.toronto.edu`

**Michael Cooper**
University of Toronto, Vector Institute
Toronto, ON, Canada
`coopermj@cs.toronto.edu`

**Viet Nguyen**
University of Toronto, Vector Institute
Toronto, ON, Canada
`viet@cs.toronto.edu`

**Rahul G. Krishnan**
University of Toronto, Vector Institute
Toronto, ON, Canada
`rahulgk@cs.toronto.edu`

## ABSTRACT

Training machine learning models requires the storage of large datasets, which often contain sensitive or private data. Storing data is associated with a number of potential risks which increase over time, such as database breaches and malicious adversaries. Machine unlearning is the study of methods to efficiently remove the influence of training data subsets from previously-trained models. Existing unlearning methods typically require direct access to the "forget set"—the data to be forgotten-and organisations must retain this data for unlearning rather than deleting it immediately upon request, increasing risks associated with the forget set. We introduce *partially-blind unlearning*—utilizing auxiliary information to unlearn without explicit access to the forget set. We also propose a practical framework RELOAD , a partially-blind method based on gradient optimization and structured weight sparsification to operationalize partially-blind unlearning. We show that RELOAD efficiently unlearns, approximating models retrained from scratch, and outperforms several forget set-dependent approaches. On language models, RELOAD unlearns entities using <0.025% of the retain set and <7% of model weights in <8 minutes on Llama2-7B. In the corrective case, RELOAD achieves unlearning even when only 10% of corrupted data is identified. [1]

## 1 MOTIVATION

In many facets of modern life, individuals consent for institutions to collect and use their personal data. Patients allow their data to be stored in electronic health records, internet surfers allow their browsing behaviour to be used to customize their searches, and citizens respond to public surveys, file their taxes, and register to vote using online government services. Frequently, institutions leverage machine learning (ML) models to derive insights, generate knowledge, or extract value from user data (Shinde & Shah, 2018; Pi, 2021; Sarker, 2021; Rahman et al., 2024). However, the act of collecting and storing a user's data poses inherent risk to the user. For example, cybercriminals may breach an institution's data security to commit identity theft (Anderson et al., 2008) or leak user data (Kenny, 2018; Zou et al., 2018), or patient records in an electronic health record system may be improperly accessed by curious healthcare workers (Long, 2016). These kind of breaches have the potential to cause users financial, clinical, and reputational harm.

Informally, modern machine learning systems expose the user to two types of risk: *dataset risk* represents the user risk associated with an institution storing a user's data, while *model risk* represents the additional risk to the user when their data is used to train a machine learning model. Whereas an ordinary data breach is an example of dataset risk, the reconstruction of user data from model weights

---

*This work was completed while the author was a student at the University of Toronto and Vector Institute.
[1] A software implementation of our work can be found on the project page.

by malicious actors (Shokri et al., 2017; Haim et al., 2022; Oz et al., 2024) is an example of model risk. If we assume that instances of dataset risk and model risk each occur with some nonnegative rate, $\mathcal{R}_\mathcal{D}, \mathcal{R}_\mathcal{M}$, respectively, we can represent the instantaneous risk borne by an individual at time $t$ as $\mathcal{R}$ as $\mathcal{R}(t) = \mathcal{R}_\mathcal{D}(t) + \mathcal{R}_\mathcal{M}(t)$. Since both risk functions are nonnegative, the cumulative data risk and model risk, $\int_0^T \mathcal{R}_\mathcal{D}(t)\, dt$ and $\int_0^T \mathcal{R}_\mathcal{M}(t)\, dt$, increase with $T$. This captures the simple intuition that risk compounds: the longer personal data remains stored by an institution or embedded within a model, the greater a user's cumulative risk exposure.

"Right to be forgotten" provisions such as the GDPR (European Parliament & Council of the European Union), allow users to demand that their data be deleted from both data stores and machine learning models (via machine unlearning, Bourtoule et al. (2019))—in theory, eliminating both database and model risk. In practice, because unlearning methods directly employ the "forget set" (Graves et al., 2021; Thudi et al., 2022; Chundawat et al., 2022; Fan et al., 2023)—the user data meant to be deleted—institutions must retain the user data to perform unlearning after a request for deletion is made. This is typically not a fast process: because instance-wise unlearning is expensive on the ever-larger that are becoming commonplace, institutions often accumulate deletion requests and process them in batches using conventional forget set-dependent methods (Hu et al., 2023). However, as long as this data is retained—often for a long time, on the order of months—the user remains exposed to both $\mathcal{R}_\mathcal{D}(t)$ and $\mathcal{R}_\mathcal{M}(t)$ for all $t$ between when the request for deletion is made and when unlearning has been completed.

This work develops a procedure for machine unlearning that does not require access to the forget set. Such a procedure would allow user data to be immediately removed when a request for deletion is made, elimi-

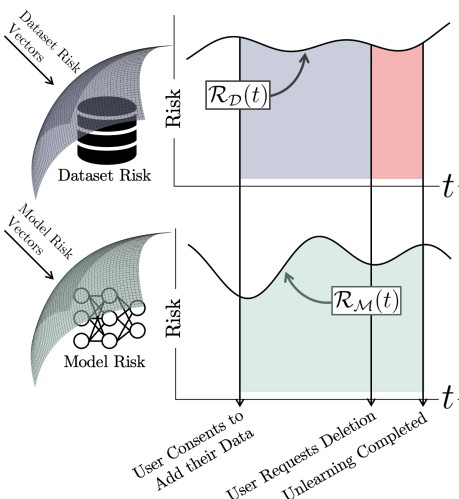

Figure 1: Conventional unlearning algorithms admit a cumulative user risk totalling the sum of the green, blue, and red regions. By allowing user data to be deleted immediately once a request for deletion is made, RELOAD eliminates the user risk associated with the red region.

nating the continued accumulation of dataset risk $\mathcal{R}_\mathcal{D}(t)$ after that time (the red region of Figure 1). Our algorithm, RELOAD, combines insights from three families of unlearning algorithms—gradient-based, structured sparsity-based, and finetuning-based algorithms—to implement a procedure that performs unlearning using only minimal auxiliary information about the forget set, rather than the forget set itself. An organisation using RELOAD would be able to immediately delete user data once a request for deletion is made without inhibiting downstream unlearning. Our work makes the following three contributions:

1. We establish and motivate the *partially-blind unlearning* (PBU) setting, capturing the intuition of unlearning without the forget set while leveraging auxiliary information. We characterise it in terms of input requirements, privacy risks, and approximation quality.

2. We introduce the RELOAD algorithm as an efficient algorithm to enable partially-blind unlearning. Rather than requiring the forget set, RELOAD only requires cached gradients from the final step of training.

3. We show that RELOAD consistently outperforms baselines across both standard and corrective unlearning scenarios (Goel et al., 2024). Surprisingly, RELOAD achieves state-of-the-art results *even when compared to algorithms that make use of the forget set*. We then extend the RELOAD algorithm to perform entity-level unlearning in language models (LMs).

Technology privacy law must be constrained by technological limitations: lawmakers cannot demand the implementation of technically-infeasible solutions. Although modern technology privacy law like GDPR permits temporary retention of data to facilitate downstream unlearning, we do not believe this provides the strongest privacy outcomes for users. Our work advances the frontier of

machine unlearning to lay the technological groundwork for stricter data-deletion timelines under right-to-be-forgotten legislation.

## 2 METHOD

### 2.1 SETTING AND NOTATION

Let $\mathcal{D} = \{z_i\}_{i=1}^N$ with $z_i \in \mathcal{Z}$ represent (i.i.d.) training data from individuals on which an organisation seeks to train a machine learning model. For a class of models $\mathcal{M} := \{M_\theta : \theta \in \Theta\}$ parametrised by a family $\Theta$, denote the parameters that minimise the empirical risk on $\mathcal{D}$ by

$$\theta^* := \arg\min_{\theta \in \Theta} \mathcal{L}(\theta; \mathcal{D}),$$

where $\mathcal{L} : \Theta \times \mathcal{Z} \to \mathbf{R}^+$ is a differentiable, additive risk (loss) function, and denote the trained model as $M_{\theta^*} \in \mathcal{M}$. For simplicity, we shorten $\mathcal{L}(\theta; \mathcal{D})$ to $\mathcal{L}(\mathcal{D})$. Let $\mathcal{D}_{forget} \subset \mathcal{D}$ represent the *forget set*, the subset of training data corresponding to individuals who have requested deletion of their information from the organisation's system. Complementarily, define the *retain set* $\mathcal{D}_{retain} := \mathcal{D} \setminus \mathcal{D}_{forget}$. In an ideal world where $M_{\theta^*}$ is currently in deployment, upon the deletion of $\mathcal{D}_{forget}$, the organisation should deploy a new model, model $M_{\theta^\sim}$, trained on the retain set with

$$\theta^\sim := \arg\min_{\theta \in \Theta} \mathcal{L}(\theta; \mathcal{D}_{retain}),$$

Thus, *machine unlearning (MU)* aims to transform $M_{\theta^*}$ into a model $M_{\tilde{\theta}}$ close to $M_{\theta^\sim}$ in some appropriate model-distances (e.g. predictive divergence or weight distance) without costly retraining an entirely new model. A classical unlearning algorithm is a function $\mathcal{A}_{MU}$ mapping $\mathcal{A}_{MU}(M_{\theta^*}, \mathcal{D}_{retain}, \mathcal{D}_{forget})$ to weights $\tilde{\theta} \in \Theta$ such that $M_{\tilde{\theta}} \approx M_{\theta^\sim}$ typically by directly using $\mathcal{D}_{forget}$ in the update rule (e.g., targeted gradient steps, reweighting, or pointwise correction). As previously discussed, classical unlearning approaches that require direct access to $\mathcal{D}_{forget}$ continually compound data subjects' cumulative risk, yet completely eliminating the influence of $\mathcal{D}_{forget}$ from $M_{\theta^*}$ without any information about the forget set is impossible: we cannot unlearn from nothing. While we must avoid retaining the raw forget set data, we can leverage **auxiliary information** that was collected during the original training process serving as a privacy-preserving proxy that enables effective unlearning while allowing immediate deletion of the sensitive data.

Thus, denote by $\mathcal{I}_{\mathcal{D}}$ any auxiliary object derived from training on $\mathcal{D}$ that may be retained to help perform unlearning without keeping raw examples from $\mathcal{D}_{forget}$. Examples include (but are not limited to) aggregated gradient statistics or feature summaries. We further impose the design desideratum that $\mathcal{I}_{\mathcal{D}}$ be chosen so that *recovering individual examples in $\mathcal{D}_{forget}$ from $\mathcal{I}_{\mathcal{D}}$ is difficult* in a practical sense (e.g. low instance-level leakage or computational hardness). We do **not** require an impossibility claim; rather this is a constraint on acceptable choices of $\mathcal{I}_{\mathcal{D}}$.

**Definition 1** (Partially-blind unlearning). *An unlearning algorithm $\mathcal{A}_{\mathrm{PBU}}$ operates in the* partially-blind unlearning (PBU) *setting if it has access to (i) the trained model $M_{\theta^*}$, (ii) the retain set $\mathcal{D}_{retain}$, and (iii) auxiliary training information $\mathcal{I}_{\mathcal{D}}$. The algorithm outputs $M_{\tilde{\theta}} = \mathcal{A}_{\mathrm{PBU}}(M_{\theta^*}, \mathcal{D}_{retain}, \mathcal{I}_{\mathcal{D}})$ such that $M_{\tilde{\theta}} \approx M_{\theta^\sim}$, where $\theta^\sim$ is the retraining solution on $\mathcal{D}_{retain}$.*

### 2.2 THE RELOAD ALGORITHM

The RELOAD algorithm is a PBU algorithm leveraging auxiliary object $\mathcal{I}_{\mathcal{D}} := \nabla_\theta \mathcal{L}(\mathcal{D})$, the gradient of the loss function evaluated on $(\theta^*, \mathcal{D})$ at the final epoch of training, in order to eliminate the influence of the forget set from the trained model. RELOAD comprises three stages combining insights from the approaches of Thudi et al. (2022), Fan et al. (2023), and Warnecke et al. (2023). Figure 2 provides a summary of the method.

**1. Ascent (Step (3) in Figure 2).** With the trained model $M_{\theta^*}$, cached gradients $\nabla_\theta \mathcal{L}(\mathcal{D})$, and the retain set $\mathcal{D}_{retain}$, RELOAD performs a single gradient ascent step (Thudi et al. (2022)) in the direction of $\nabla_\theta \mathcal{L}(\mathcal{D}) - \nabla_\theta \mathcal{L}(\mathcal{D}_{retain})$ with learning rate $\eta_p$:

$$\theta' \leftarrow \theta^* + \eta_p(\nabla_\theta \mathcal{L}(\mathcal{D}) - \nabla_\theta \mathcal{L}(\mathcal{D}_{retain}))$$

**2. Re-initialisation (Step (4) in Figure 2).** RELOAD identifies weights responsible for characterizing information about $\mathcal{D}_{forget}$ (Fan et al. (2023)) and re-initialises them. Concretely, *knowledge values*

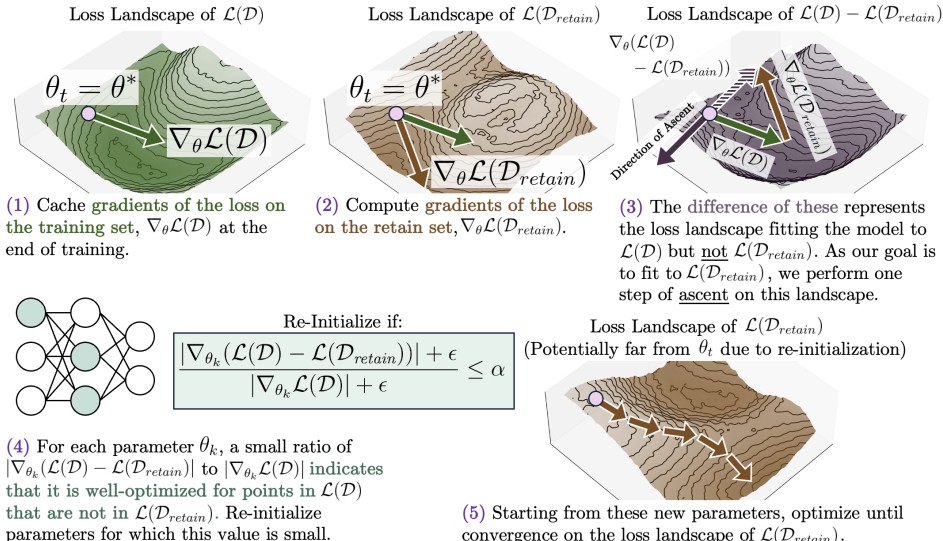

Figure 2: Overview of the RELOAD algorithm for partially-blind approximate unlearning. RELOAD marries a gradient-based unlearning step modified for the PBU setting (Steps (1) through (3)) with a weight saliency-based selective reinitialisation (Step (4)) and subsequent fine-tuning (Step (5)). Because the partially-blind unlearning setting prohibits taking gradients with respect to $\mathcal{D}_{forget}$, RELOAD exploits the linearity of differentiation to treat $\nabla_\theta(\mathcal{L}(\mathcal{D}) - \mathcal{L}(\mathcal{D}_{retain}))$ as a proxy for $\nabla_\theta \mathcal{L}(\mathcal{D}_{forget})$ at the location in weight space corresponding to $\theta_t$. This allows us to apply one gradient ascent step in this direction. Intuitively, this update in Step (3) removes some information about $\mathcal{D}_{forget}$ from *all* network weights, while the reinitialisation in Step (4) reinitialises those weights with a uniquely strong correspondence to $\mathcal{D}_{forget}$ (for which a single ascent step will not fully remove this information). RELOAD achieves state-of-the-art performance on a collection of unlearning tasks, often outperforming baselines with direct access to $\mathcal{D}_{forget}$.

of each weight $\theta_k$ are computed for the updated weights $\theta'$. For a small $\epsilon$, the knowledge value of $\theta_k$ is:

$$KV_{\theta_k} := \frac{|\nabla_{\theta_k}\mathcal{L}(\mathcal{D}) - \nabla_{\theta_k}\mathcal{L}(\mathcal{D}_{retain})| + \epsilon}{|\nabla_{\theta_k}\mathcal{L}(\mathcal{D})| + \epsilon}$$

where a low knowledge value indicates that $\theta_k$ carries stronger knowledge of $\mathcal{D}_{forget}$. Let $KV := \{KV_{\theta_k} : \theta_k \in \theta'\}$ denote the set of all knowledge values of $\theta'$. For a quantile hyperparameter $\alpha$, all weights $\theta_k$ with $KV_{\theta_k} \leq \text{Quantile}_\alpha(KV)$ are re-initialized. Denote by $\theta^\dagger$ the weights post selective reinitialization.

**3. Finetuning (Step (5) in Figure 2).** RELOAD finetunes $M_{\theta^\dagger}$ (Warnecke et al. (2023)) until convergence by minimising $\mathcal{L}(\mathcal{D}_{retain})$ via iterative gradient-based optimization starting from $\theta^\dagger$ and obtain $\tilde{\theta}$.

## 2.3 ALGORITHMIC INSIGHTS

**Partial blindness of $\nabla_\theta \mathcal{L}(\mathcal{D})$.** RELOAD uses gradients $\nabla_\theta \mathcal{L}(\mathcal{D})$ cached upon the completion of the last training epoch as auxiliary training information. While prior work has demonstrated that inputs can be reconstructed from gradient information (Geiping et al., 2020; Zhao et al., 2020; Vero et al., 2023; Wu et al., 2023a; Gao et al., 2021), these reconstruction methods either require knowing batch sizes (unavailable in our setting) or make restrictive assumptions like no duplicate labels (Xue et al., 2023). Moreover, reconstructed images are typically unrecognizable with only a small portion showing limited fidelity (Geiping et al., 2020). Thus, cached summed gradients represent a valid choice for $\mathcal{I}_\mathcal{D}$ in the partially-blind setting.

**Direction of Movement.** The central challenge of partially-blind unlearning is that taking repeated gradients of $\mathcal{L}(\mathcal{D}_{forget})$ is impossible without access to $\mathcal{D}_{forget}$. However, from cached gradients of

$\mathcal{D}$ at the conclusion of model training, $\nabla_\theta \mathcal{L}(\mathcal{D})$, we can infer $\nabla_\theta \mathcal{L}(\mathcal{D}_{forget})$ (Appendix A.1).

$$\nabla_\theta \mathcal{L}(\mathcal{D}_{forget}) = \nabla_\theta \mathcal{L}(\mathcal{D}) - \nabla_\theta \mathcal{L}(\mathcal{D}_{retain}).$$

Therefore, a gradient-based descent update in the direction of $\nabla_\theta \mathcal{L}(\mathcal{D}_{forget})$ moves the model weights such that they better fit to $\mathcal{D}_{forget}$; because our goal is *unlearning* $\mathcal{D}_{forget}$, RELOAD instead begins with a single gradient *ascent* update step (in the opposite direction). This informs Step **(2 - 3)** in Figure 2.

**Targeted Weight Adjustments.** Taking a gradient step in this direction is insufficient for unlearning for two reasons: we are limited to a single step without access to $\mathcal{D}_{forget}$, and network modularity theory (Rodriguez et al., 2019) suggests that a small subset of weights contains disproportionate information about $\mathcal{D}_{forget}$. While one ascent step removes some information about $\mathcal{D}_{forget}$ across all weights, it cannot fully remove information from the subset most responsible for characterizing the forget set.

We therefore perform selective reinitialisation based on weight importance to ensure full removal of information from the subset most responsible for characterizing the forget set. The relative magnitude of $\nabla_{\theta_k} \mathcal{L}(\mathcal{D}_{forget})$ compared to $\nabla_{\theta_k} \mathcal{L}(\mathcal{D})$ represents how much weight $\theta$ is responsible for characterizing $\mathcal{D}_{forget}$. A small relative magnitude indicates that $\theta_k$ is well-optimized to characterise instances in $\mathcal{D}_{forget}$ while a large relative magnitude indicates that $\theta_k$ poorly characterises these instances. We call this the *knowledge value* of weight $\theta_k$, formally defined as,

$$KV_{\theta_k} := \frac{|\nabla_{\theta_k} \mathcal{L}(\mathcal{D}_{forget})| + \epsilon}{|\nabla_{\theta_k} \mathcal{L}(\mathcal{D})| + \epsilon} = \frac{|\nabla_{\theta_k}(\mathcal{L}(\mathcal{D}) - \mathcal{L}(\mathcal{D}_{retain}))| + \epsilon}{|\nabla_{\theta_k} \mathcal{L}(\mathcal{D})| + \epsilon}$$
$$= \frac{|\nabla_{\theta_k} \mathcal{L}(\mathcal{D}) - \nabla_{\theta_k} \mathcal{L}(\mathcal{D}_{retain})| + \epsilon}{|\nabla_{\theta_k} \mathcal{L}(\mathcal{D})| + \epsilon}, \tag{1}$$

where $\epsilon$ is a small Laplace smoothing constant. By selectively reinitialising all weights $\theta_k$ if $KV_{\theta_k} \leq \text{Quantile}_\alpha(KV)$, where $\alpha$ controls the aggressiveness of re-initialisation selection, we can remove the influence of the weights uniquely responsible for encoding information about $\mathcal{D}_{forget}$. This informs Step **(4)** in Figure 2. This thinking extends on lines of work in gradient-based input saliency maps (Smilkov et al., 2017) and saliency unlearning by Fan et al. (2023). We ablate knowledge value formulas in Appendix C.1.

Due to this tight coupling of components, the produced effect is a large modification to weights which strongly characterise instances in $\mathcal{D}_{forget}$ combined with a smaller modification to the remainder of the weights, enabling model-wide removal of characterisation of the instances in $\mathcal{D}_{forget}$. We ablate components of the RELOAD in Appendix C.2 and C.8 and confirm that these components are non-redundant and essential.

## 3 EMPIRICAL RESULTS AND ANALYSIS

Our empirical evaluation has five complementary objectives to assess RELOAD's capabilities. (1) **Methodological Inspection** empirically verifies each component of RELOAD's unlearning procedure to ensure well-founded design choices. (2) **Classical Unlearning** evaluates RELOAD's performance on forgetting individuals' private data points, assessing effectiveness at approximating models trained only on the retain set. (3) **Entity Unlearning** examines forgetting specific entities or concepts in LMs, testing RELOAD's ability to remove knowledge about individuals from the TOFU dataset (Maini et al., 2024). (4) **Corrective Unlearning** (Goel et al., 2024) investigates mitigating training data aberrations when only a subset of affected samples can be identified, focusing on challenging scenarios where fewer than 80% of corrupted samples are identified—representing realistic conditions with incompletely diagnosed data quality issues. (5) **Ablations of RELOAD Components** provides a complete picture of the algorithm's stability across model types, gradient caching techniques, and more.

### 3.1 METHODOLOGICAL INTROSPECTION

To introspect on RELOAD, we focus on the simplest unlearning task: unlearning a class of data from a trained model. In this case, we unlearn the class "8" from a ResNet-18 model trained on the

SVHN dataset. We further ablate on the number of ascent steps, knowledge value formulas, and hyperparameters in Appendix C.

Figure 3 visualises selected feature maps of the ResNet-18 model at different stages of RELOAD and their t-SNE representations (van der Maaten & Hinton, 2008), colored by their predicted label[2]. The experiment demonstrates the importance of the reinitialisation step (Step (4) in Figure 2), as even after a single ascent step, the model still finds "8" to be the most probable class. Only after the important weights are identified and reinitialised does the model emits a lower-entropy distribution classifying the digit as a "2". This suggests that the primary utility of the ascent step in our algorithm is in amending the representations of $\mathcal{D}_{forget}$ in the later layers of the network, while the selective weight reinitialisation step modifies the representations produced by earlier layers. The findings of this experiment provide a degree of empirical confirmation of the intuition presented in Section 2.3.

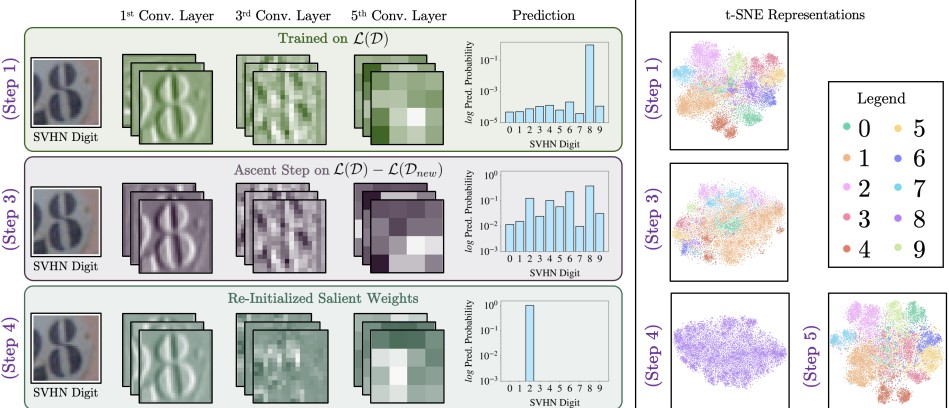

Figure 3: Introspecting on selected feature maps of a ResNet-18 model when using RELOAD to unlearn the class "8". *(Left)* The feature maps (activations) of the first channel fo the first, third, and fifth convolutional layers after the application of each algorithm step in Figure 2. After Step (3), the activations of the model remain largely unchanged, although the logits represent a considerably more uniform distribution over the digits. After Step (4), the activation of the first convolutional layer is largely unchanged—which is expected, as the earlier layers of CNNs tend to correspond to broad feature detectors (Zeiler & Fergus, 2014))—although the feature maps change and the predictive distribution contentrates around "2". *(Right)* t-SNE visualisations of the 5[th] Conv. Layer of a ResNet-18 trained on the same task in a different training run. At Step (1), classes largely cluster into nice separated regions with some overlap. The Ascent Step (3) disrupts the cluster separation, preserving some clustering structure while breaking apart much of the original class boundaries. Reinitialisation (Step (4)) reveals that the model collapses onto one label at this stage (on the left '2', on the right, "8"—though in our experience it is coincidental that this run happens to concentrate on the class we desire to unlearn). Finally, Finetuning (Step (5)) recovers much of the original separability, restoring the distinct class clusters seen in the initial t-SNE. Importantly, we observe that no samples are predicted to be "8".

## 3.2 CLASSICAL UNLEARNING EXPERIMENTS

**Baselines.** We compare RELOAD against baseline approaches of GA (Thudi et al., 2022), FT (Warnecke et al., 2023), SSD (Foster et al., 2023), SCRUB (Kurmanji et al., 2023), CF-$k$ (Goel et al., 2022), EU-$k$ (Goel et al., 2022), SalUn (Fan et al., 2023), and Fisher (Golatkar et al., 2020). FT, CF-$k$, EU-$k$, and Fisher are partially-blind algorithms, whereas the others require direct access to $\mathcal{D}_{forget}$. We also present results from ground-truth retraining from scratch, $M_{\theta\sim}$, to evidence the performance of unlearning algorithms. More details on baselines are provided in Appendix B.1.

**Evaluation.** We assess the performance similarity between our learned model and a baseline version of $M_{\theta\sim}$ trained naively from scratch using two key metrics. First, we calculate the forget accuracy difference ($\Delta$FA, $\downarrow$), representing the performance gap between our method and the baseline $M_{\theta\sim}$

---

[2]t-SNE visualizations in Figure 3 are of a separate, independent run from the left-hand side figure.

on $\mathcal{D}_{forget}$. Second, we compute the difference in forget membership inference attack success rates ($\Delta$FMIA, $\downarrow$), which quantifies how well the membership inference attack from (Shokri et al., 2017) can identify $\mathcal{D}_{forget}$ samples in each model's training data relative to the baseline identification rate on $M_{\theta\sim}$. Additional evaluation metrics, experimental details, and hyperparameter configurations are detailed in Appendices B.3, B.7, and B.8 respectively.

| | 10% Random | | 100 In-Class | | | Entity Unlearning 1% | |
|---|---|---|---|---|---|---|---|
| Algorithm | $\Delta$FA ($\downarrow$) | $\Delta$FMIA ($\downarrow$) | $\Delta$FA ($\downarrow$) | $\Delta$FMIA ($\downarrow$) | Algorithm | FQ ($\uparrow$) | MU ($\uparrow$) |
| GA | $18.77 \pm 2.43$ | $0.21 \pm 0.06$ | $23.33 \pm 1.06$ | $0.07 \pm 0.06$ | GA | $0.0068$ | $-0.0233$ |
| SSD | $74.17 \pm 2.04$ | $0.15 \pm 0.21$ | $68.67 \pm 1.97$ | $0.38 \pm 0.14$ | Grad Diff | $0.0143$ | $-0.0198$ |
| SCRUB | $18.85 \pm 2.39$ | $0.20 \pm 0.06$ | $27.55 \pm 1.43$ | $0.07 \pm 0.06$ | NPO-RT | $0.5786$ | $-0.1361$ |
| CF-$k$ | $18.01 \pm 2.60$ | $0.20 \pm 0.06$ | $21.84 \pm 0.88$ | $0.07 \pm 0.06$ | Pref Opt | $0.0971$ | $-0.0021$ |
| SalUn | $13.14 \pm 2.53$ | $7.39 \pm 2.60$ | $12.08 \pm 3.13$ | $\mathbf{0.02 \pm 0.02}$ | ECO (Zero-Out) | $\mathbf{0.9900}$ | $+0.0000$ |
| Fisher | $22.99 \pm 2.30$ | $7.27 \pm 2.48$ | $10.72 \pm 1.98$ | $0.03 \pm 0.04$ | Original | $0.0030$ | $+0.0000$ |
| **RELOAD** | $\mathbf{0.30 \pm 0.50}$ | $\mathbf{0.01 \pm 0.01}$ | $\mathbf{3.44 \pm 1.46}$ | $\mathbf{0.02 \pm 0.02}$ | **RELOAD** | $0.4046$ | $\mathbf{+0.0748}$ |

Table 1: Benchmarking RELOAD against baselines in the uncorrelated 10% setting, 100 in-class samples setting, and 1% forgetting entity unlearning setting. Best performances are **bolded**.

**RELOAD effectively unlearns both random and correlated samples.** We evaluate RELOAD under two regimes: randomly selecting 10% of CIFAR-100 training samples for $\mathcal{D}_{forget}$, and selecting 100 samples from a single class to assess unlearning of correlated data. Partial results are reported in Table 1 while complete results are deferred to tables in Appendix B.4. In both settings, RELOAD achieves strong performance across key metrics. For random sample unlearning, RELOAD attains the highest RA while maintaining the lowest $\Delta$FA, $\Delta$FE, $\Delta$FMIA, and FSKL, suggesting superior approximation of $M_{\theta\sim}$ compared to baselines. For correlated sample unlearning, RELOAD achieves the lowest $\Delta$FMIA and FSKL and performs competitively on other metrics, closely approximating $M_{\theta\sim}$. While Fisher marginally outperforms RELOAD in the correlated setting, it requires over twice the computational time as retraining, making RELOAD more practical. Methods like CF-$k$ and EU-$k$ achieve higher RA in the correlated setting due to minimal weight updates, but perform poorly on critical unlearning metrics like $\Delta$FA and $\Delta$FE. Surprisingly, these results demonstrate that RELOAD's superior approximation of $M_{\theta\sim}$ enables more effective unlearning than methods that explicitly leverage $\mathcal{D}_{forget}$ during the unlearning process We observe that RELOAD outperforms both GA and FT consistently, which form parts of the RELOAD algorithm. This supports the utility of how RELOAD is designed, and that each step of the algorithm plays a critical role in unlearning.

## 3.3 ENTITY UNLEARNING WITH LMS

**Baselines and Evaluation.** We compare RELOAD against baselines from Maini et al. (2024) on entity unlearning using TOFU's synthetic author biography dataset (Maini et al., 2024). We evaluate using forget quality (KS test $p$-value between unlearned and retrained model distributions) and model utility (performance on retained data and real-world knowledge). Experiments use Phi 1.5 (Li et al., 2023) and Llama-2-7B-Chat (Touvron et al., 2023) with open-source fine-tuned models (locuslab, 2025; Unlearning, 2025a;b;c).

**RELOAD unlearns select entities.** In this experiment, we task each algorithm with unlearning a subset of the fictitious authors in TOFU from a TOFU fine-tuned model. We observe that RELOAD effectively unlearns when the number of entities to forget is small. As evidenced in Table 1, RELOAD outperforms many existing unlearning methods in the 1% forgetting case, effectively forgetting entities ($p$-value $\gg 0.05$) and *improving* model utility over the Original and Retrained (Retain) models. However, in the 5% and 10% forgetting cases, RELOAD fails to repair model utility despite succesful forgetting. We hypothesise that this is due to limits on the size of $\mathcal{D}_{repair}$. Due to computational restrictions, the size of $\mathcal{D}_{repair}$ was restricted (maximum 195 samples) which greatly limited the applicability of RELOAD when $|\mathcal{D}_{prompts}| > |\mathcal{D}_{repair}|$. As the 1% forgetting case is the only case in which $|\mathcal{D}_{prompts}| \leq |\mathcal{D}_{repair}|$, this suggests that this bound is a requirement for the effective application of RELOAD for entity unlearning. Similarly due to computational constraints, we reuse results and the reference implementation for experiments from prior work (Liu et al., 2024a). Further experiments are provided in Appendix B.5.

The experiment on Llama-2-7B-Chat completes in 8 minutes on a single RTX6000 GPU, using 7% of weights and <0.025% of retained data, demonstrating RELOAD 's efficiency for small-scale entity unlearning.

## 3.4 CORRECTIVE UNLEARNING

**Baselines and Evaluation.** Corrective unlearning (CU) (Goel et al., 2024) considers the case where a portion $\mathcal{D}_m$ of $\mathcal{D}$ has been adversely affected (e.g. mislabeled or poisoned). CU aims to update $\theta$ so as to approximate training on $\mathcal{D} \setminus \mathcal{D}_m$ where only a subset $\mathcal{D}_{forget} \subseteq \mathcal{D}_m$ has been identified. Existing methods degrade rapidly when $\gamma := |\mathcal{D}_{forget}| / |\mathcal{D}_m| < 0.8$ and fail under adversarial corruptions or large-scale poisoning Pawelczyk et al. (2025). To gauge performance on CU, we use the corrected accuracy $\text{Acc}_{corr}$ ($\uparrow$), measuring the performance of the unlearned model on the adversely affected data $\mathcal{D}_m$. Full details on prior work, baselines, and evaluation metrics for CU are detailed in Appendices A.5, B.2, and B.3.

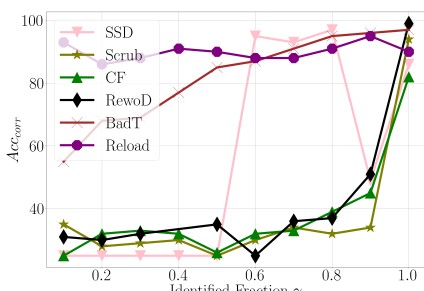

Figure 4: CU results on CIFAR-10 for dataset poisoning with $|\mathcal{D}_m| = 100$. RELOAD achieves high $\text{Acc}_{corr}$ across all $\gamma \in (0, 1.0]$ whereas baseline methods often struggle with low $\gamma$.

**RELOAD efficiently corrects trained models.** Under adverse effects of manipulations following the baselines outlined in prior work, RELOAD outperforms on $\text{Acc}_{corr}$ at low percentages of data identification (Figure 4) while observing competitive computational efficiency (Table 23) even at only $\gamma = 0.1$. Although BadT (Chundawat et al., 2022) outperforms RELOAD in CIFAR-100 poisoning experiments, it bears much greater computational cost (Table 23). We report further experiments and results in Appendix B.6.

## 3.5 ABLATION OF RELOAD COMPONENTS

To further characterize the behavior of RELOAD , we conducted an extensive suite of ablations across various architectures and data regimes. These experiments investigate the impact of gradient quantization to mitigate the storage overhead, the stability of the algorithm when applied to Vision Transformers to characterize our method across model types and architectures, performance in continual learning and fine-tuning scenarios for real-world practicality, its efficacy on human sensitive datasets for unlearning private data (Newatia et al., 2024), and how RELOAD behaves when only using subsets of $\mathcal{D}_{retain}$ for finetuning.

We defer the reader to Appendix C for detailed analyses and discussions.

## 4 RELATED WORK

**Exact and Approximate Unlearning.** Exact unlearning provides formal guarantees for information removal from model weights. Methods include naive retraining (the gold standard (Cao & Yang, 2015; Thudi et al., 2022; Shaik et al., 2024)), SISA (Bourtoule et al., 2019) for accelerated retraining via data partitioning, Certified Data Removal (Guo et al., 2019) using reverse Newton updates, and Certified Graph Unlearning (Chien et al., 2022) leveraging graph topology. Approximate unlearning methods like RELOAD recover retain-set behavior without theoretical guarantees. Approximate unlearning algorithms can be classified into gradient-based and weight-saliency approaches. *(i)* Gradient-based approximate unlearning methods perform optimization using both forget and retain sets. Simple approaches apply gradient ascent on forget loss to undo weight updates (Graves et al., 2021; Thudi et al., 2022). Teacher-student methods include Bad Teacher (Chundawat et al., 2022), which distills from models trained on retain data ("good teacher") and randomly initialized on forget data ("bad teacher"), and SCRUB (Kurmanji et al., 2023), where students learn to disobey teachers by maximizing forget loss. Representation-based methods include DUCK (Cotogni et al., 2023), driving forget representations toward incorrect centroids, and Boundary Unlearning (Chen et al., 2023) for class-level decision boundary shifts. *(ii)* Weight saliency-based approximate unlearning methods target specific weights based on neural modularity (Pfeiffer et al., 2023) and sparsity (Frankle & Carbin, 2018; Chen et al., 2024). SalUn (Fan et al., 2023) uses gradient thresholds to identify

forget-sensitive weights, while SSD (Foster et al., 2023) scales weights using Fisher Information Matrix importance scores without gradient steps.

**Partially-Blind Unlearning.** Related to Zero-Shot Unlearning Chundawat et al. (2023) (restricted to class unlearning), this setting is more realistic and applicable. Methods include Finetuning (FT) (Warnecke et al., 2023) on retain sets, Catastrophically forgetting last $k$ layers (CF-$k$), Exact-unlearning last $k$ (EU-$k$) (Goel et al., 2022), and Fisher Forgetting (Golatkar et al., 2020). Both FT and CF-$k$ provide no strong theoretical indication of unlearning while RELOAD provides stronger theoretical indication by selectively reinitialising weights bearing the most knowledge on $\mathcal{D}_{forget}$.

**Unlearning for LMs.** Most methods use optimization to balance forgetting undesirable sequences while retaining useful ones (Yao et al., 2024; Liu et al., 2024b; He et al., 2025). Some identify and edit sparse weight subsets (Wu et al., 2023b; Ilharco et al., 2023; Belrose et al., 2025) but require large retention datasets or full-model updates (Eldan & Russinovich, 2023). Optimization-free methods have been explored (Liu et al., 2024a). RELOAD offers a lightweight alternative requiring minimal data, few updates, and fast convergence.

## 5 DISCUSSION

**RELOAD effectively unlearns arbitrary samples.** Despite operating in the partially-blind setting, RELOAD outperforms MU algorithms that enjoy direct access to $\mathcal{D}_{forget}$. However, RELOAD trades off runtime and performance in unlearning arbitrary samples of data, requiring the caching of summed gradients over $\mathcal{D}$ from the final step of training, a non-trivial spatial cost. When $\mathcal{D}_{forget}$ is available, RELOAD does not require this caching as the gradients can be computed at runtime. In both settings, RELOAD 's method proves to be an empirically effective approximate unlearning method.

**RELOAD causes LMs to forget entities.** In applications to LMs, RELOAD is able to quickly and cheaply remove knowledge of entities when the number of target entities is $\leq$ the size of the subset of the retained data used for computing knowledge values. Applications of our algorithm also leads to an overall increase in model performance. When this condition isn't met, RELOAD fails to achieve both model utility and forget quality. In addition, RELOAD achieves this without needing to modify the inference pipeline of these models (as opposed to baselines such as (Liu et al., 2024a)).

**RELOAD corrects data aberrations.** In corrective unlearning, RELOAD remains an efficient, performant method in this regime, outperforming existing baselines and serving as a viable approach for all currently explored forms of corrective unlearning. RELOAD demonstrates its ability to unlearn manipulations when $< 80\%$ of the manipulated data is identified in all corrective cases, presenting a step up from prior results (Goel et al., 2024). This holds practical value when $\gamma$ is not known in deployment and suggests that our work may contain generalizable insights about about learning to fit arbitrary downstream transformations of data.

**RELOAD discards batch statistics.** Caution is required for applying RELOAD to models using Batch Normalization (e.g., ResNet-18). Since removing the forget set alters the batch statistics (mean/variance) for the retained set, the gradients are technically coupled. However, in practice, we observe that this "approximation mismatch" is negligible. As detailed in Section 3.2, RELOAD achieves competitive performance on ResNet-18, suggesting the method is robustly handles the minor gradient drift introduced by batch-dependent statistics.

## 6 CONCLUSION AND LIMITATIONS

Further work on partially-blind algorithms, in particular RELOAD , could focus on behaviour in contrastively trained models. Investigating the effect of a RELOAD -style unlearning on latent representations of a contrastively-trained model is an exciting future direction. Crucially, the focus in such a setting shifts from measuring traditional performance gain or drop with respect to the retrain baseline, to assessing the identifiability and similarity of the learned representations between the unlearned model and the retrain baseline. For instance, future work could focus on metrics that quantify how closely the representation space of the unlearned model aligns with that of the retrained model, such as centered kernel alignment (CKA). It is further important to study whether RELOAD performs well on models with self-supervised pretrining and subsequent finetuning using a task-based

decomposable loss such as cross-entropy or if decomposable losses can serve as a surrogate for RELOAD unlearning on models which are not trained with such a loss.

By enabling unlearning without direct access to the forget set, RELOAD addresses the fundamental privacy paradox in machine unlearning: that the very data subjects wish to remove must be retained during the unlearning process. This capability allows organisations to immediately delete requested data upon receipt of deletion requests, effectively stopping the cumulation of database-related privacy risks rather than perpetuating them through batched processing delays. In doing so, RELOAD represents a meaningful step toward aligning technological capabilities with regulatory mandates for deletion "without undue delay" and the privacy protection goals underlying right-to-be-forgotten legislation. While our work demonstrates substantial empirical improvements across multiple unlearning scenarios, future research may explore theoretical bounds for partially-blind unlearning, broader model classes, and further reducing auxiliary data requirements.

ACKNOWLEDGMENTS

The authors would like to thank Anvith Thudi, Patrik Reizinger, Siddharth Arya, and Alexander Capstick for constructive conversations on unlearning. The authors would also like to thank Younwoo (Ethan) Choi for insights on contextual fine-tuning for large language models alongside Asic Chen and Chenika Bukes for discussions on neural network initialization techniques, unlearning, and weight sparsity. The authors would also like to thank the RGK lab members for general discussions, support, and feedback.
RGK is supported by a Canada CIFAR AI Chair and a Canada Research Chair Tier II in Computational Medicine (CRC-2022-00049). This research was supported by an NFRF Special Call NFRFR2022- 00526. Resources used in preparing this research were provided, in part, by the Province of Ontario, the Government of Canada through CIFAR, and companies sponsoring the Vector Institute www.vectorinstitute.ai/partnerships/. This research was enabled in part by support provided by Compute Ontario (https://www.computeontario.ca/) and the Digital Research Alliance of Canada (https://www.alliancecan.ca/en).

REPRODUCIBILITY STATEMENT

In efforts to promote reproducibility the authors have provided the source code used to run the experiments.

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

## A  Formal Treatment and Gradient Derivations

### A.1  The Reload Algorithm

---

**Algorithm 1** The RELOAD Algorithm for Partially-Blind Unlearning

---

1: **Input:** $M_{\theta^*}$, cached $\nabla_\theta \mathcal{L}(\mathcal{D})$, $\mathcal{D}_{retain}$
2: **weights:** $\eta_p$: priming step learning rate, $\epsilon$: noise weight, $\alpha$: reset proportion
3: **Output:** Trained model approximating $M_{\theta^\sim}$
4:
5: **procedure** RELOAD($M_{\theta^*}, \nabla_\theta \mathcal{L}(\mathcal{D}; M_{(\theta^*)}), \mathcal{D}_{retain}$)
6:     $\theta' \leftarrow \theta^* + \eta_p(\nabla_\theta \mathcal{L}(\mathcal{D}) - \nabla_\theta \mathcal{L}(\mathcal{D}_{retain}))$             ▷ Step **(2 − 3)** *(Fig. 2)*
7:     $\text{KV} \leftarrow \left\{ \frac{|\nabla_{\theta_k} \mathcal{L}(\mathcal{D}) - \nabla_{\theta_k} \mathcal{L}(\mathcal{D}_{retain})| + \epsilon}{|\nabla_{\theta_k} \mathcal{L}(\mathcal{D})| + \epsilon} \right\}_{\theta_k \in \theta}$     ▷ Step **(3)** *(Fig. 2)*
8:     **for** $\theta_k \in \theta'$ **do**
9:         **if** $\text{QUANTILE}_{KV}(KV_{\theta_k}) \leq \alpha$ **then**
10:             $\theta'_k \leftarrow \text{INITIALIZE}(\cdot)$                        ▷ Step **(4)** *(Fig. 2)*
11:         **end if**
12:     **end for**
13:     Train $M_{(\theta')}$ to convergence on $\mathcal{D}_{retain}$                ▷ Step **(5)** *(Fig. 2)*
14: **end procedure**

---

Our RELOAD algorithm (Fig. 2) contains the following steps based on the intuition presented in Section 2.3.

**(1)** Cache the gradients $\nabla_\theta \mathcal{L}(\mathcal{D})$ at the end of training.

**(2)** Compute $\nabla_\theta \mathcal{L}(\mathcal{D}_{retain})$.

**(3)** Perform *one* step of gradient *ascent* in the direction of $\nabla_\theta \mathcal{L}(\mathcal{D}) - \nabla_\theta \mathcal{L}(\mathcal{D}_{retain})$.

**(4)** Reinitialize all weights $\theta_k$ that are smaller than the $\alpha$-QUANTILE of knowledge values.

**(5)** fine-tune until convergence on $\mathcal{L}(\mathcal{D}_{retain})$.

A formal description of this algorithm is shown in Algorithm 1.

For entity unlearning in language models, the RELOAD algorithm requires modifications. The design of RELOAD for LMs is outlined in Appendix A.3.

A.2 GRADIENT INFORMATION AND DERIVATION

**Information contained in gradients.** RELOAD relies on information about $\mathcal{D}$ contained within the cached gradients, raising the question of how it behaves in the modern setting where networks are trained to convergence. We first observe that $\|\nabla\mathcal{L}(\mathcal{D})\|_{\theta_k} \to 0$ does not imply $\|\nabla\mathcal{L}(\mathcal{S})\|_{\theta_k} \to 0$ for $\mathcal{S} \subset \mathcal{D}$. This means that even if the summed-cached gradients are all approximately zero, Step (3) may still induce non-trivial weight updates. For the same reason, the numerator (from Equation 1) in Step (4) is non-zero.

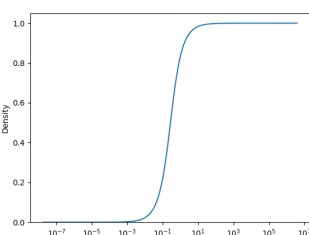

Figure 5: (Smoothed) empirical distribution of knowledge values computed over the weights of ResNet-18 trained on SVHN.

However, as $\|\nabla\mathcal{L}(\mathcal{D})\|_{\theta_k} \to 0$, the denominator in Equation 1 approaches $\epsilon$. This does not influence the behaviour of RELOAD, because this scaling by $\frac{1}{\epsilon}$ applies uniformly to all knowledge values, and $\alpha$ is a quantile of the empirical distribution of the knowledge values. As such, a constant scaling factor applied to the knowledge values will not affect which weights are re-initialized. Figure 5 shows a smoothed empirical distribution of knowledge values demonstrating the spread of values computed for different weights across a ResNet-18 network.

RELOAD relies on equating $\nabla_\theta\mathcal{L}(\mathcal{D}_{forget})$ to $\nabla_\theta\mathcal{L}(\mathcal{D}) - \nabla_\theta\mathcal{L}(\mathcal{D}_{retain})$. We provide a full derivation below following from the linearity of differentiation and is used to justify the RELOAD algorithm in the partially-blind setting of classical machine unlearning and corrective machine unlearning. For the purpose of corrective unlearning (with replacement), additional arguments are required and presented in Appendix A.5.

**Gradient Derivation.** Recall that $\mathcal{D} = \{z_i\}_{i=1}^N$ with $z_i \in \mathcal{Z}$. Let $z_i = (x_i, y_i)$ represent an input-output pair in the dataset. Then, let $\hat{y}_i = M_\theta(x_i)$ represent the model's prediction on $x_i$. Subsequently,

$$\nabla_\theta\mathcal{L}(\mathcal{D}_{forget}) = \sum_{(x_i,y_i)\in\mathcal{D}_{forget}} \nabla_\theta\mathcal{L}((x_i,y_i),\hat{y}_i) = \sum_{(x_i,y_i)\in\mathcal{D}\backslash\mathcal{D}_{retain}} \nabla_\theta\mathcal{L}((x_i,y_i),\hat{y}_i) \quad (2)$$

where the second equality follows from $\mathcal{D}_{retain} = \mathcal{D}\backslash\mathcal{D}_{forget}$. Equivalently,

$$= \sum_{(x_i,y_i)\in\mathcal{D}} \nabla_\theta\mathcal{L}((x_i,y_i),\hat{y}_i) - \mathbb{1}_{(x_i,y_i)\in\mathcal{D}_{retain}}[\nabla_\theta\mathcal{L}((x_i,y_i),\hat{y}_i)] \quad (3)$$

$$= \sum_{(x_i,y_i)\in\mathcal{D}} \nabla_\theta\mathcal{L}((x_i,y_i),\hat{y}_i) - \sum_{(x_i,y_i)\in\mathcal{D}_{retain}} \nabla_\theta\mathcal{L}((x_i,y_i),\hat{y}_i) \quad (4)$$

$$= \nabla_\theta\mathcal{L}(\mathcal{D}) - \nabla_\theta\mathcal{L}(\mathcal{D}_{retain}). \quad (5)$$

A.3 THE RELOAD ALGORITHM FOR LANGUAGE MODELS

RELOAD on LMs leverages insights from investigations into language models (Choi et al., 2025) to produce richer gradients. This is necessary as LMs are prohibitively-large for the standard RELOAD procedure which requires $\mathcal{D}_{retain}$ and cached gradients. By reducing the requirements of RELOAD for LMs to $\mathcal{D}_{prompts}$, we enable practical partially-blind unlearning on LMs without modifying the inference pipeline.

In this setting, we assume access to a model trained on a dataset $\mathcal{D}$. Additionally, we assume access to a dataset of prompts inquiring about $\mathcal{D}_{forget}$, $\mathcal{D}_{prompts}$ (eg. 'Who is Harry Potter?' if $\mathcal{D}_{forget}$ contained texts on Harry Potter). Finally, we assume access to a small sample of the retain set $\mathcal{D}_{repair} \subseteq \mathcal{D}_{retain}$. We also select a subset of model layers to unlearn from, $\mathcal{D}_{layers}$, with total parameters $\theta_{selected}$. All gradients and computations are performed solely on these layers, enabling parameter-efficient optimization and unlearning. Given that modern language models are trained on very large datasets and contain billions of parameters, it is crucial to be able to perform unlearning with small datasets in a parameter-efficient manner (operating on a small subset of model weights).

This procedure contains the following steps.

**(1)** Collect the outputs of the model on $\mathcal{D}_{prompts}$ in $\mathcal{D}_{outputs}$.

**(2)** Embed each element of $\mathcal{D}_{outputs}$ in contextual fine tuning (Choi et al., 2025) to create $\mathcal{D}_{embedded}$ (see Appendix A.4).

**(3)** Collect the outputs of the model on $\mathcal{D}_{embedded}$ in $\mathcal{D}_{embedded\_outputs}$.

**(4)** Collect language-modelling gradients $\nabla_\theta \mathcal{L}(\mathcal{D}_{repair})$, but do not update parameters.

**(5)** Collect language-modelling gradients $-\nabla_\theta \mathcal{L}(\mathcal{D}_{embedded\_outputs})$, and update parameters (one step of gradient *ascent*).

**(6)** Reinitialize all parameters $\theta_k \subseteq \theta_{selected}$ that are smaller than the $\alpha$-QUANTILE of knowledge values.

**(7)** fine-tune until convergence on $\mathcal{L}(\mathcal{D}_{repair})$.

A formal description is also shown (Algorithm 2).

It is important to note that in this setting, RELOAD does not require storing gradients, reducing space utilization and overhead. However, due to the requirement of prompts in $\mathcal{D}_{prompts}$, it offers weaker privacy guarantees (ie. knowing which concept is being unlearned).

---

**Algorithm 2** The RELOAD Algorithm for LMs

---

1: **Input:** $M_{\theta^*}, \mathcal{D}_{prompts}, \mathcal{D}_{repair}$
2: **Parameters:** $\eta_p$: priming step learning rate, $\epsilon$: noise parameter, $\alpha$: reset proportion
3: **Output:** Trained model approximating $M_{\theta\sim}$
4:
5: **procedure** RELOAD-LM($M_{\theta^*}, \mathcal{D}_{prompts}, \mathcal{D}_{repair}$)
6:      $\mathcal{D}_{outputs} \leftarrow M_{(\theta^*)}(\mathcal{D}_{prompts})$
7:      $\mathcal{D}_{embedded} \leftarrow \text{CFT}(\mathcal{D}_{outputs})$
8:      $\mathcal{D}_{embedded\_outputs} \leftarrow M_{(\theta^*)}(\mathcal{D}_{embedded})$
9:      $KV \leftarrow \left\{ \frac{|\nabla_{\theta_k} \mathcal{L}(\mathcal{D}_{embedded\_outputs})| + \epsilon}{|\nabla_{\theta_k} \mathcal{L}(\mathcal{D}_{repair})| + \epsilon} \right\}_{\theta_k \in \theta_{selected}}$        ▷ Step (3) *(Fig. 2)*
10:      $\theta'_{selected} \leftarrow \theta^*_{selected} + \eta_p \nabla_\theta (\mathcal{L}(\mathcal{D}_{embedded\_outputs})))$        ▷ Step (2 − 3) *(Fig. 2)*
11:      **for** $\theta_k \in \theta'_{selected}$ **do**
12:          **if** $\text{QUANTILE}_{KV}(KV_{\theta_k}) \leq \alpha$ **then**
13:              $\theta'_k \leftarrow \text{INITIALIZE}(\cdot)$        ▷ Step (4) *(Fig. 2)*
14:          **end if**
15:      **end for**
16:      Train $M_{(\theta')}$ to convergence on $\mathcal{D}_{repair}$        ▷ Step (5) *(Fig. 2)*
17: **end procedure**

---

## A.4   CONTEXTUAL FINE-TUNING FOR UNLEARNING FROM LANGUAGE MODELS

Choi et al. (2025) outline that embedding a prompt with their method improves the model's ability to capture underlying functional relationships. In addition, this method of fine-tuning LM's has been shown to lead to improved learning. Therefore, parameter updates informed through this method are more informed - suggesting that the gradients produced may be more informed about parameter knowledge themselves.

We find that using this method of contextual embedding to produce gradients for RELOAD leads to better identification of knowledgeable parameters about $\mathcal{D}_{prompts}$. This wholly informs the functionality of our unlearning algorithm, leading to quick and effective forgetting from LMs.

Given a prompt $x$ and a model $M$, we extract the knowledge contained within the LM by performing a forward pass on the prompt $x$ to obtain $M(x)$. $M(x)$ is a textual representation of what the LM knows about the concept prompted in $x$. We then insert $M(x)$ into the {content} field of the prompt presented below which we pass to a black-box LM (in our experiments we use the Anthropic API to access `claude-3-5-haiku-20241022`). This yields a contextual fine-tuning prompt, $y$, aimed at extracting the most knowledge about a topic from an LM. We then perform a forward pass on $y$ to yield $M(y)$ on which we collect language modeling gradients.

The following is the fine-tuning prompt used for black-box LMs for unlearning:

```
"""
Based on the TARGET CONCEPT:
Generate a concise "contextual prompt" that will enhance learning effectiveness and draw out all
relevant knowledge.
The prompt should:
1. Follow the style of [select one learning theory approach: In-Depth Exploration/Reflective
Thinking/Summarization and Synthesis/Focus on Key Concepts/Contextual Understanding/Critical
Analysis/Question-Based Learning]
2. Explicitly identify:
    • The fundamental concepts that must be understood
    • Key relationships between important elements
    • Critical facts that require focus for mastery
    • How these elements connect to and are relevant for reasoning or application
3. Be formatted as a directive that encourages active engagement with the material (approximately
3-5 sentences)
4. Frame the learning in a way that facilitates long-term retention, practical application, and
maximizes extracting knowledge from the learner.
TARGET CONCEPT: {content}
Your contextual prompt should help the learner not just memorize information but develop a deeper,
more applicable understanding of the concept.
"""
```

## A.5 Corrective Unlearning and Gradient Derivation

### A.5.1 Corrective Unlearning

Goel et al. (2024) introduce the setting of *corrective unlearning* in which a subset of the training data $\mathcal{D}_m \subset \mathcal{D}$ is adversely manipulated. This could include mislabeling, backdoors, and data poisoning. The corrective unlearning case studies the ability of unlearning algorithms to unlearn the adverse effects produced by the presence of these abberations in the training dataset when a sample of them are identified ($\mathcal{D}_{forget} \subset \mathcal{D}_m$). This extends machine unlearning beyond privacy-related deletions. Interestingly, in this setting, a naively-retrained model is *not* the gold-standard, as manipulated data may remain in $\mathcal{D}_{retain}$ unknown to the practitioner. Existing methods degrade rapidly when $\gamma = \frac{|\mathcal{D}_{forget}|}{|\mathcal{D}_m|} < 0.8$, and fail under adversarial corruptions or large-scale poisoning (Pawelczyk et al., 2025).

Following prior work, let $\mathcal{D}_m \subseteq \mathcal{D}$ denote the subset of training points that are adversely affected (mislabeled, corrupted, or poisoned). Only a portion of these may be identified as the forget set. Corrective unlearning aims to obtain $M_{\tilde{\theta}} \approx M_{\theta^\sim}$. Therefore, we write

$$\mathcal{D}_{forget} \subseteq \mathcal{D}_m, \qquad \gamma \triangleq \frac{|\mathcal{D}_{forget}|}{|\mathcal{D}_m|} \in [0,1], \qquad \theta^\sim \triangleq \arg\min_{\theta \in \Theta} \mathcal{L}(\theta; \mathcal{D} \setminus \mathcal{D}_m),$$

so $\gamma$ is the fraction of adverse instances that are identified. The difficulty of corrective unlearning increases as $\gamma$ decreases because $\mathcal{D}_{retain} = \mathcal{D} \setminus \mathcal{D}_{forget}$ not $\mathcal{D} \setminus \mathcal{D}_m$.

### A.5.2 Corrective Unlearning (with replacement)

In addition to studying the corrective unlearning case, we provide a weaker extension of the corrective setting. Our expansion to the corrective unlearning setting is applicable when the identified samples in $\mathcal{D}_{forget}$ are additionally *transformed*, *corrected*, or *amended* and *included* in $\mathcal{D}_{retain}$. In this setting, let $f : \mathcal{Z} \to \mathcal{Z}$ denote a transformation, and write $z'_i = f(z_i)$. Then, $\mathcal{D}_{retain}$ represents the result of applying $f$ item-wise to $K$ elements of $\mathcal{D}$, and applying the identity transform to the remaining $N - K$ elements, as

$$\mathcal{D}_{retain} = \{z'_i\}_{i=1,\dots,K} \cup \{z_i\}_{i=K+1,\dots,N}. \tag{6}$$

This is an extension of the corrective unlearning problem, as we wish to "unlearn" the influence of $\{z_i\}_{i=1,\dots,K}$ on our original model, and "relearn" the influence of $\{z'_i\}_{i=1,\dots,K}$. This additional setting encompasses the following data transformations, among others:

1. *Covariate Correction*: $\mathcal{D}_{retain} = \{z'_i = (x'_i, y_i)\}_{i=1,\dots,K} \cup \{z_i\}_{i=K+1,\dots,N}$, where $x'_i$ represents a corrected version of the features $x_i$, and indices $K + 1, \dots, N$ correspond to those with erroneous covariates (e.g., data was corrupted during collection/pre-processing).

2. *Label Correction*: $\mathcal{D}_{retain} = \{z'_i = (x_i, y'_i)\}_{i=1,...,K} \cup \{z_i\}_{i=K+1,...,N}$, where $y'_i$ represents a corrected version of the label $y_i$, and indices $K+1, ..., N$ correspond to those that were originally mis-labelled during annotation.

3. *Backdoor Removal*: $\mathcal{D}_{retain} = \{z'_i = (x'_i, y_i)\}_{i=1,...,K} \cup \{(x_i, y_i)\}_{i=K+1,...,N}$, where $x'_i$ represents a version of the features $x_i$ lacking the injected backdoor pattern, and indices $K+1, ..., N$ correspond to those that were originally transformed with a backdoor during processing. Models trained with backdoors in the training set learn shortcuts (Geirhos et al., 2020), which can be exploited to induce misclassification.

For simplicity, we use $\mathcal{S}^c$ to denote the complement of the set or dataset $\mathcal{S}$.

In classical and regular corrective unlearning, our goal is to obtain a gradient in the direction of $\mathcal{D}_{forget}$ for RELOAD. The corrective unlearning (with replacement) case is more general: the goal is to obtain $\nabla_\theta \mathcal{L}(\mathcal{D} \cap \mathcal{D}^c_{retain})$, a gradient pointing towards the empirical minimum of the loss on elements that are uniquely contained in $\mathcal{D}$ and not in $\mathcal{D}_{retain}$, and $-\nabla_\theta \mathcal{L}(\mathcal{D}^c \cap \mathcal{D}_{retain})$, a gradient pointing *away* from the empirical minimum of the loss on elements uniquely contained in $\mathcal{D}_{retain}$ but not in $\mathcal{D}$. This is a general abstraction of the difference in gradients between a dataset and a subset of that dataset, to the difference in gradients between two datasets. Unlearning represents the special case of this framework in which $\mathcal{D} \cap \mathcal{D}^c_{retain} = \mathcal{D}_{forget}$ and $\mathcal{D}^c \cap \mathcal{D}_{retain} = \emptyset$. In the corrective unlearning (with replacement) setting, the desired gradient is also $\nabla_\theta \mathcal{L}(\mathcal{D}) - \nabla_\theta \mathcal{L}(\mathcal{D}_{retain})$; for which we provide justification and a derivation below. This validates Step (2 - 3) in RELOAD for this problem.

We now outline the derivation of the gradients informing the RELOAD algorithm in the context of corrective unlearning (with replacement). Recall that in this case, $\mathcal{D}_{retain} \neq \mathcal{D} - \mathcal{D}_{forget}$, which invalidates the justification outlined in Section 2.3. Below, we provide a derivation which justifies the same choice of gradient for the corrective unlearning (with replacement) setting.

In the setting of corrective unlearning (with replacement), we construct these sets.

$$\mathcal{D} \cap \mathcal{D}^c_{retain} = \{z_i = (x_i, y_i)\}_{i=K+1...N}, \mathcal{D}^c \cap \mathcal{D}_{retain} = \{z'_i = (x'_i, y'_i)\}_{i=K+1...N}, \text{ and} \tag{7}$$

$$\mathcal{D} \cap \mathcal{D}_{retain} = \{z_i = (x_i, y_i)\}_{i=1...K} \tag{8}$$

The gradient then formulates as

$$\sum_{\substack{(x_i,y_i)\in \\ \{(x_i,y_i)\}_{i=K+1...N}}} \nabla_\theta \mathcal{L}((x_i,y_i), \hat{y}_i) - \sum_{\substack{(x_i,y_i)\in \\ \{(x'_i,y'_i)\}_{i=K+1...N}}} \nabla_\theta \mathcal{L}((x'_i,y'_i), \hat{y}_i) = \sum_{(x_i,y_i)\in\mathcal{D}} \nabla_\theta \mathcal{L}((x_i,y_i), \hat{y}_i) \tag{9}$$

$$- \mathbb{1}_{\substack{(x_i,y_i)\in \\ \mathcal{D}\cap\mathcal{D}_{retain}}} [\nabla_\theta \mathcal{L}((x_i,y_i), \hat{y}_i)] - \left(\sum_{(x_i,y_i)\in\mathcal{D}_{retain}} \nabla_\theta \mathcal{L}((x_i,y_i), \hat{y}_i) - \mathbb{1}_{\substack{(x_i,y_i)\in \\ \mathcal{D}\cap\mathcal{D}_{retain}}} [\nabla_\theta \mathcal{L}((x_i,y_i), \hat{y}_i)]\right) \tag{10}$$

$$= \sum_{(x_i,y_i)\in\mathcal{D}} \nabla_\theta \mathcal{L}((x_i,y_i), \hat{y}_i) - \sum_{(x_i,y_i)\in\mathcal{D}\cap\mathcal{D}_{retain}} \nabla_\theta \mathcal{L}((x_i,y_i), \hat{y}_i) - \sum_{(x_i,y_i)\in\mathcal{D}_{retain}} \nabla_\theta \mathcal{L}((x_i,y_i), \hat{y}_i) + \sum_{(x_i,y_i)\in\mathcal{D}\cap\mathcal{D}_{retain}} \nabla_\theta \mathcal{L}((x_i,y_i), \hat{y}_i) \tag{11}$$

$$= \sum_{(x_i,y_i)\in\mathcal{D}} \nabla_\theta \mathcal{L}((x_i,y_i), \hat{y}_i) - \sum_{(x_i,y_i)\in\mathcal{D}_{retain}} \nabla_\theta \mathcal{L}((x_i,y_i), \hat{y}_i) \tag{12}$$

$$= \nabla_\theta \mathcal{L}(\mathcal{D}) - \nabla_\theta \mathcal{L}(\mathcal{D}_{retain}) \tag{13}$$

In this work, we consider the Corrective Unlearning (with replacement) cases that correspond to the classical corrective unlearning scenarios outlined by Goel et al. (2024). We recreate the experimental settings exactly, except the samples in $\mathcal{D}_{forget}$ are *corrected*, and included in $\mathcal{D}_{retain}$.

# B  EXPERIMENTAL DETAILS AND RESULTS

## B.1  BASELINES FOR CLASSICAL UNLEARNING

**Gradient ascent (GA) (Thudi et al., 2022).** GA operates by taking several steps of *gradient ascent* on $\mathcal{D}_{forget}$ thereby removing the trained model from a loss minimum on $\mathcal{D}_{forget}$. This approach is not partially-blind.

**Fine-Tuning (FT) (Warnecke et al., 2023).** FT leverages the concept of catastrophically-forgetting (McCloskey & Cohen, 1989) to unlearn $\mathcal{D}_{forget}$ by fine-tuning on $\mathcal{D}_{retain}$. This approach is partially-blind.

**Selective Synaptic Dampening (SSD) (Foster et al., 2023).** SSD studies the amount of information about $\mathcal{D}_{forget}$ contained within weights using an approximation of the Fisher Information Matrix. Proportional to each weight's 'importance', SSD scales the weight value to induce forgetting. This approach is not partially-blind.

**Scalable Remembering and Unlearning Bound (SCRUB) (Kurmanji et al., 2023).** SCRUB alternates optimising between distilling away from the original model on $\mathcal{D}_{forget}$ and towards the original model on $\mathcal{D}_{retain}$. Notably, the second distillation loss is combined with a task-specific loss (eg. cross-entropy for classification).

**Catastrophically Forgetting the last $k$-layers (CF-$k$) (Goel et al., 2022).** CF-$k$ leverages the concept of catastrophically-forgetting (McCloskey & Cohen, 1989) by freezing all but the last $k$ layers of the model and performing fine-tuning on $\mathcal{D}_{retain}$. This approach is partially-blind.

**Exact Unlearning the last $k$-layers (EU-$k$) (Goel et al., 2022).** EU-$k$ reinitialises the weights of the last $k$ layers, freezes the rest, and fine-tunes on $\mathcal{D}_{retain}$. This approach is partially-blind.

**Salience Unlearning (SalUn) (Fan et al., 2023).** SalUn is a framework in which important weights to $\mathcal{D}_{forget}$ are identified and all but those are frozen for optimisation updates. Authors report the greatest improvement when combined with Random Labelling (RL) (Golatkar et al., 2020). RL assigns random labels to instances in $\mathcal{D}_{forget}$ and then fine-tunes on this data. This approach is not partially-blind.

**Fisher Forgetting (Fisher) (Golatkar et al., 2020).** Fisher leverages the Fisher Information Matrix over $\mathcal{D}_{forget}$ to perform a Fisher-regularised weight update to the model. This approach is not partially-blind.

### B.2 BASELINES FOR CORRECTIVE UNLEARNING

Baselines are taken directly from Goel et al. (2024). We repeat their details below.

**Retrain without Deletion Set (RewoD).** RewoD represents a naively retrained model on $\mathcal{D}_{retain} = \mathcal{D} \setminus \mathcal{D}_{forget}$. Notably in this setting, some affected samples of $\mathcal{D}_m$ may still be in $\mathcal{D}_{retain}$.

**Catastrophically Forgetting all layers/Finetuning (CF) (Goel et al., 2022; Warnecke et al., 2023).** CF leverages the concept of catastrophically-forgetting (McCloskey & Cohen, 1989) by performing fine-tuning on $\mathcal{D}_{retain}$.

**Selective Synaptic Dampening (SSD) (Foster et al., 2023).** SSD studies the amount of information about $\mathcal{D}_{forget}$ contained within weights using an approximation of the Fisher Information Matrix. Proportional to each weight's 'importance', SSD scales the weight value to induce forgetting.

**Knowledge Distillation from a Bad Teacher (BadT) (Chundawat et al., 2022).** BadT uses a combined distillation approach by learning from a randomly initialised network on $\mathcal{D}_{forget}$ and the original model on $\mathcal{D}_{retain}$.

**Scalable Remembering and Unlearning Bound (SCRUB) (Kurmanji et al., 2023).** SCRUB alternates optimising between distilling away from the original model on $\mathcal{D}_{forget}$ and towards the original model on $\mathcal{D}_{retain}$. Notably, the second distillation loss is combined with a task-specific loss (eg. cross-entropy for classification).

### B.3 EVALUATION METRICS FOR MACHINE UNLEARNING

We present the evaluation metrics used in our experiments comparing the RELOAD algorithm with baselines. An up arrow ↑ indicates that the higher the better, while a down arrow ↓ indicates that the lower the better.

One of the goals of unlearning is to produce a model that is a close approximation of the naively retrained one. FSKL and RSKL are evaluation metrics which quantify the dissimilarity between the outputs of the unlearned model and the retrained model on the same data. $\Delta$FA, $\Delta$FE, $\Delta$FMIA

| Statistic | Abbr. | Description |
|---|---|---|
| Accuracy on $\mathcal{D}_{retain}$ ($\uparrow$) | RA | Model accuracy on the $\mathcal{D}_{retain}$. In unlearning, a higher accuracy indicates that the unlearning process has not negatively impacted the model's performance on the retained data. |
| Diff. in Accuracy on $\mathcal{D}_{forget}$ ($\downarrow$) | $\Delta$FA | The change in accuracy on the forget set between the current model and $\mathcal{M}^{\theta^{\sim}}$. A smaller difference, approaching the accuracy of the retrained model, indicates that the unlearning method has been more effective in "forgetting" the forget set. |
| Diff. in Error on $\mathcal{D}_{forget}$ ($\downarrow$) | $\Delta$FE | The reduction in error on the forget set between the current model and $\mathcal{M}^{\theta^{\sim}}$. A smaller difference, approaching the error of the retrained model, signifies that the unlearning method has been more effective at "forgetting" the forget set. |
| Diff. in MIA Success Rate on $\mathcal{D}_{forget}$ ($\downarrow$) | $\Delta$FMIA | Difference in success rate of a membership inference attack (MIA) on the forget set between the current model and $\mathcal{M}^{\theta^{\sim}}$. In this work, we use the attack from (Shokri et al., 2017) implemented in the repository for (Kurmanji et al., 2023). A success rate approaching that of the retrained model implies the forgotten data is indistinguishable to an MIA on in-distribution data that the model was not trained on. |
| Symmetric KL-Divergence on $\mathcal{D}_{retain}$ ($\downarrow$) | RSKL | Symmetric KL-Divergence between the logits of the current model and those of $\mathcal{M}^{\theta^{\sim}}$. This metric is averaged over all instances in the $\mathcal{D}_{retain}$. A lower symmetric KL divergence indicates an unlearning method that behaves similarly on the $\mathcal{D}_{retain}$ to a model retrained from scratch without the forget set. |
| Symmetric KL-Divergence on $\mathcal{D}_{forget}$ ($\downarrow$) | FSKL | The Symmetric KL-Divergence between the logits of the current model and those of $\mathcal{M}^{\theta^{\sim}}$. This metric is averaged over all instances in the $\mathcal{D}_{forget}$. A lower symmetric KL divergence indicates that the unlearning method that behaves similarly on the $\mathcal{D}_{forget}$ to a model retrained from scratch without the forget set. |
| Cost ($\downarrow$) | Cost | Ratio of the runtime of the unlearning method to the runtime of retraining a baseline model from scratch without the forget set. A lower cost indicates a more computationally efficient method. |

Table 2: **Evaluation Statistics for Unlearning.**

are comparison metrics to benchmark the difference in performance between the unlearned model and the retrained model on the same data. Similar behaviour on $\mathcal{D}_{forget}$ implies that the unlearned model is indistinguishable from a retrained one. Unlearning is only a useful procedure if it is cheap and yields a useful model. RA measures the utility of the unlearned model, and Cost measures how expensive the unlearning procedure is, relative to retraining from scratch.

**Corrective Unlearning (with and without replacement) Evaluation Metrics**

$Acc_{retain}$ and $Acc_{corr}$ are metrics introduced in Goel et al. (2024) to measure the effectiveness of a corrective unlearning algorithm. Cost is the same as above, to compare the unlearning algorithm to the expense of full training.

### B.4 CLASSICAL UNLEARNING RESULTS

We present a full set of experiments exploring the effectiveness of the RELOAD algorithm on the classical unlearning task of *item unlearning*. This suite of experiments include unlearning randomly-selected samples and correlated samples. Across these categories we select 10% and 30% of training data samples for random-sample unlearning, and 100 data points from a single class for correlated-sample unlearning. These cases are explored across 3 datasets (CIFAR10, CIFAR100, and SVHN) and 2 models (ResNet-18 and VGG16-BN).

As previously discussed, RELOAD operates in a partially-blind setting. This means that within the results presented below, RELOAD performs this unlearning without access to $\mathcal{D}_{forget}$.

| Statistic | Abbr. | Description |
|---|---|---|
| Retain Accuracy ($\uparrow$) | Acc$_{\text{retain}}$ | Model accuracy on a held out test set ($\mathcal{D}_{retain}^{(test)}$) of the same distribution as $\mathcal{D}_{retain}$. In corrective unlearning, a higher accuracy indicates that the corrective unlearning process has correctly adapted the model to its new training set. |
| Corrected Accuracy ($\uparrow$) | Acc$_{\text{corr}}$ | Model accuracy on the adversely affected data $\mathcal{D}_m$. In the case of backdoor attacks or noisy corrective unlearning, a higher value indicates the relearned model correctly has lost its reliance on the backdoor pattern. In label correction setting, the desirable value is the percentage of samples that did not have their labels flipped (in our experiments, 90%). |
| Cost ($\downarrow$) | Cost | Ratio of the runtime of the corrective unlearning method to the runtime of retraining a baseline model from scratch without the forget set. A lower cost indicates a more computationally efficient method. |

Table 3: **Evaluation Statistics for Corrective Unlearning.**

**RELOAD unlearns randomly-selected samples.** We randomly assign 10% of CIFAR-100 training samples to $\mathcal{D}_{forget}$, to showcase how well each method can unlearn arbitrary training samples from a ResNet-18 and report our results in Table 8. RELOAD achieves the highest RA, while maintaining the lowest $\Delta$FA, $\Delta$FE, $\Delta$FMIA, and FSKL of all approaches. This suggests that RELOAD successfully approximates $M_{\theta\sim}$ better than the baselines. That FT achieves a lower RSKL than RELOAD is hardly surprising, as RSKL measures dissimilarity in logits on $\mathcal{D}_{retain}$, and FT adjusts a converged model $M_{\theta*}$ to fit a subset of its original task. Similarly, the computational cost of RELOAD , though similar to many baselines, is considerably greater than either SSD or GA.

**RELOAD efficiently unlearns correlated samples.** We randomly assign 100 samples from a single class of the training data to $\mathcal{D}_{forget}$, to evaluate how well each method can unlearn arbitrary but related training samples and report our results in Table 15. RELOAD achieves the lowest $\Delta$FMIA and FSKL of all approaches and very close to the lowest $\Delta$FA, $\Delta$FE, and RSKL of all approaches, suggesting that RELOAD closely approximates $M_{\theta\sim}$ in this setting. RELOAD is marginally outperformed by Fisher in these settings, but is far more feasible, as Fisher requires over twice as much time as retraining. Although RELOAD achieves an RA competitive with that of most baselines, naive gradient ascent, CF-$k$, and EU-$k$ yield a marginally higher RA. This can be attributed to the small number of unlearning samples; optimizing to maximize the loss on these samples does not provide a strong enough gradient update. CF-$k$ and EU-$k$ both make few weight updates to $M_{\theta*}$, which leads to a high RA but poor performance on unlearning metrics like $\Delta$FA and $\Delta$FE.

Further experimental results on random 10% and random 30% forgetting are provided in the tables below (Appendix B.4.1, B.4.2). Results on 100 correlated-sample unlearning are provided in Appendix B.4.3.

## B.4.1 RANDOM 10% FORGETTING

| Method | RA ($\uparrow$) | $\Delta$FA ($\downarrow$) | $\Delta$FE ($\downarrow$) | $\Delta$FMIA ($\downarrow$) | Cost ($\downarrow$) | RSKL ($\downarrow$) | FSKL ($\downarrow$) |
|---|---|---|---|---|---|---|---|
| GA | $98.38_{\pm 0.21}$ | $3.86_{\pm 0.66}$ | $0.21_{\pm 0.07}$ | $0.04_{\pm 0.02}$ | $\mathbf{0.00_{\pm 0.00}}$ | $0.06_{\pm 0.02}$ | $0.66_{\pm 0.06}$ |
| FT | $98.24_{\pm 0.21}$ | $\mathbf{1.45_{\pm 0.53}}$ | $0.16_{\pm 0.03}$ | $0.03_{\pm 0.01}$ | $0.27_{\pm 0.00}$ | $\mathbf{0.05_{\pm 0.01}}$ | $\mathbf{0.48_{\pm 0.04}}$ |
| SSD | $20.02_{\pm 29.99}$ | $75.65_{\pm 26.45}$ | $1.88_{\pm 0.62}$ | $0.01_{\pm 0.02}$ | $0.01_{\pm 0.00}$ | $8.30_{\pm 3.11}$ | $7.83_{\pm 2.70}$ |
| SCRUB | $98.41_{\pm 0.20}$ | $3.89_{\pm 0.70}$ | $0.21_{\pm 0.07}$ | $0.04_{\pm 0.02}$ | $0.02_{\pm 0.00}$ | $0.06_{\pm 0.02}$ | $0.65_{\pm 0.04}$ |
| CF-$k$ | $98.28_{\pm 0.23}$ | $3.81_{\pm 0.71}$ | $0.21_{\pm 0.07}$ | $0.05_{\pm 0.02}$ | $0.21_{\pm 0.00}$ | $0.06_{\pm 0.02}$ | $0.55_{\pm 0.04}$ |
| EU-$k$ | $98.31_{\pm 0.21}$ | $3.83_{\pm 0.71}$ | $0.21_{\pm 0.07}$ | $0.05_{\pm 0.01}$ | $0.21_{\pm 0.00}$ | $0.07_{\pm 0.02}$ | $0.56_{\pm 0.04}$ |
| SalUn | $99.78_{\pm 0.05}$ | $3.68_{\pm 0.48}$ | $0.26_{\pm 0.02}$ | $0.01_{\pm 0.01}$ | $0.16_{\pm 0.01}$ | $0.06_{\pm 0.02}$ | $0.55_{\pm 0.04}$ |
| Fisher | $99.51_{\pm 0.17}$ | $3.83_{\pm 0.44}$ | $0.07_{\pm 0.01}$ | $0.02_{\pm 0.00}$ | $1.83_{\pm 0.06}$ | $0.07_{\pm 0.02}$ | $0.56_{\pm 0.04}$ |
| RELOAD (OURS) | $\mathbf{99.49_{\pm 0.10}}$ | $1.83_{\pm 0.83}$ | $\mathbf{0.05_{\pm 0.04}}$ | $\mathbf{0.00_{\pm 0.00}}$ | $0.26_{\pm 0.09}$ | $0.12_{\pm 0.01}$ | $0.53_{\pm 0.07}$ |
| Retrained (Baseline) | $99.99_{\pm 0.01}$ | $94.40_{\pm 0.72}$ | $0.23_{\pm 0.08}$ | $0.50_{\pm 0.01}$ | - | - | - |

Table 4: **10% Random Forgetting on CIFAR-10 (ResNet-18)**
$\uparrow$: the goal is to have as high of a value as possible, $\Delta^{\downarrow}$: the value in the table is the difference between the result of the unlearning method and retraining (bottom row) on the metric and the goal is to have a low difference, $\downarrow$: the goal is to have as low of a value as possible. The bottom row presents the absolute value of $M_{(\theta\sim)}$ on each metric. For any metric with $\Delta$, the raw value is instead reported. Rows for $\Delta$FA ($\downarrow$), $\Delta$FE ($\downarrow$), and $\Delta$FMIA ($\downarrow$) present the absolute difference in the value of the corresponding method on this metric to the value of $M_{(\theta\sim)}$ on the metric. These results show that RELOAD outperforms all the baselines on RA, $\Delta$FE, $\Delta$FMIA by large margins. RELOAD performs competitively on the $\Delta$FA, FSKL, and RSKL metrics, but is outperformed by FT. RELOAD incurs a higher computational cost than other baselines other than FT.

| Method | RA ($\uparrow$) | $\Delta$FA ($\downarrow$) | $\Delta$FE ($\downarrow$) | $\Delta$FMIA ($\downarrow$) | Cost ($\downarrow$) | RSKL ($\downarrow$) | FSKL ($\downarrow$) |
|---|---|---|---|---|---|---|---|
| GA | $98.38_{\pm 0.21}$ | $4.40_{\pm 0.41}$ | $0.18_{\pm 0.02}$ | $0.05_{\pm 0.01}$ | $\mathbf{0.00_{\pm 0.00}}$ | $0.06_{\pm 0.02}$ | $0.66_{\pm 0.06}$ |
| FT | $98.24_{\pm 0.21}$ | $4.33_{\pm 0.37}$ | $0.18_{\pm 0.02}$ | $0.04_{\pm 0.01}$ | $0.26_{\pm 0.02}$ | $\mathbf{0.05_{\pm 0.01}}$ | $0.48_{\pm 0.04}$ |
| SSD | $20.02_{\pm 29.99}$ | $75.41_{\pm 26.74}$ | $1.89_{\pm 0.62}$ | $0.02_{\pm 0.03}$ | $0.01_{\pm 0.00}$ | $8.30_{\pm 3.11}$ | $7.83_{\pm 2.70}$ |
| SCRUB | $98.41_{\pm 0.20}$ | $4.47_{\pm 0.40}$ | $0.19_{\pm 0.02}$ | $0.05_{\pm 0.01}$ | $0.02_{\pm 0.00}$ | $0.06_{\pm 0.02}$ | $0.65_{\pm 0.04}$ |
| CF-$k$ | $98.28_{\pm 0.23}$ | $4.47_{\pm 0.39}$ | $0.19_{\pm 0.02}$ | $0.05_{\pm 0.01}$ | $0.17_{\pm 0.01}$ | $0.06_{\pm 0.02}$ | $0.55_{\pm 0.04}$ |
| EU-$k$ | $98.31_{\pm 0.21}$ | $4.48_{\pm 0.40}$ | $0.19_{\pm 0.02}$ | $0.06_{\pm 0.01}$ | $0.17_{\pm 0.01}$ | $0.07_{\pm 0.02}$ | $0.56_{\pm 0.04}$ |
| SalUn | $\mathbf{99.86_{\pm 0.04}}$ | $1.98_{\pm 0.48}$ | $0.09_{\pm 0.02}$ | $0.04_{\pm 0.01}$ | $0.17_{\pm 0.00}$ | $0.06_{\pm 0.02}$ | $0.55_{\pm 0.04}$ |
| Fisher | $99.61_{\pm 0.14}$ | $0.15_{\pm 0.06}$ | $\mathbf{0.00_{\pm 0.00}}$ | $0.01_{\pm 0.01}$ | $2.17_{\pm 0.04}$ | $0.07_{\pm 0.02}$ | $0.56_{\pm 0.04}$ |
| RELOAD (OURS) | $99.76_{\pm 0.16}$ | $\mathbf{0.08_{\pm 0.08}}$ | $0.01_{\pm 0.00}$ | $\mathbf{0.00_{\pm 0.00}}$ | $0.12_{\pm 0.01}$ | $\mathbf{0.05_{\pm 0.03}}$ | $\mathbf{0.19_{\pm 0.02}}$ |
| Retrained (Baseline) | $99.99_{\pm 0.00}$ | $95.16_{\pm 0.30}$ | $0.20_{\pm 0.02}$ | $0.50_{\pm 0.00}$ | - | - | - |

Table 5: **10% Random Forgetting on SVHN (ResNet-18)**
$\uparrow$: the goal is to have as high of a value as possible, $\Delta^{\downarrow}$: the value in the table is the difference between the result of the unlearning method and retraining (bottom row) on the metric and the goal is to have a low difference, $\downarrow$: the goal is to have as low of a value as possible. The bottom row presents the absolute value of $M_{(\theta\sim)}$ on each metric. For any metric with $\Delta$, the raw value is instead reported. Rows for $\Delta$FA ($\downarrow$), $\Delta$FE ($\downarrow$), and $\Delta$FMIA ($\downarrow$) present the absolute difference in the value of the corresponding method on this metric to the value of $M_{(\theta\sim)}$ on the metric. These results show that RELOAD outperforms all the baselines on RA, $\Delta$FA, $\Delta$FE, $\Delta$FMIA, FSKL, and RSKL by large margins. RELOAD performs competitively on the Cost, but incurs a higher computational cost than other baselines other than FT, CF-$k$, EU-$k$.

| Method | RA ($\uparrow$) | $\Delta$FA ($\downarrow$) | $\Delta$FE ($\downarrow$) | $\Delta$FMIA ($\downarrow$) | Cost ($\downarrow$) | RSKL ($\downarrow$) | FSKL ($\downarrow$) |
|---|---|---|---|---|---|---|---|
| GA | $98.40_{\pm 0.23}$ | $4.43_{\pm 0.44}$ | $0.21_{\pm 0.02}$ | $0.03_{\pm 0.01}$ | $\mathbf{0.00}_{\pm 0.00}$ | $0.06_{\pm 0.02}$ | $0.65_{\pm 0.06}$ |
| FT | $98.30_{\pm 0.18}$ | $4.49_{\pm 0.43}$ | $0.22_{\pm 0.02}$ | $0.03_{\pm 0.01}$ | $0.24_{\pm 0.03}$ | $\mathbf{0.05}_{\pm 0.01}$ | $\mathbf{0.49}_{\pm 0.04}$ |
| SSD | $22.88_{\pm 34.01}$ | $70.45_{\pm 29.04}$ | $1.80_{\pm 0.69}$ | $0.01_{\pm 0.01}$ | $\mathbf{0.00}_{\pm 0.00}$ | $7.99_{\pm 3.52}$ | $7.56_{\pm 3.06}$ |
| SCRUB | $98.43_{\pm 0.22}$ | $4.50_{\pm 0.41}$ | $0.22_{\pm 0.02}$ | $0.03_{\pm 0.01}$ | $0.02_{\pm 0.00}$ | $0.06_{\pm 0.02}$ | $0.66_{\pm 0.04}$ |
| CF-$k$ | $98.34_{\pm 0.24}$ | $4.51_{\pm 0.42}$ | $0.22_{\pm 0.02}$ | $0.04_{\pm 0.01}$ | $0.21_{\pm 0.03}$ | $0.06_{\pm 0.02}$ | $0.55_{\pm 0.05}$ |
| EU-$k$ | $98.34_{\pm 0.23}$ | $4.51_{\pm 0.42}$ | $0.22_{\pm 0.02}$ | $0.04_{\pm 0.01}$ | $0.21_{\pm 0.03}$ | $0.06_{\pm 0.02}$ | $0.56_{\pm 0.05}$ |
| SalUn | $\mathbf{99.94}_{\pm 0.02}$ | $3.88_{\pm 0.62}$ | $0.13_{\pm 0.01}$ | $0.04_{\pm 0.01}$ | $0.15_{\pm 0.00}$ | $0.06_{\pm 0.02}$ | $0.55_{\pm 0.05}$ |
| Fisher | $99.55_{\pm 0.18}$ | $\mathbf{0.04}_{\pm 0.04}$ | $\mathbf{0.00}_{\pm 0.00}$ | $\mathbf{0.00}_{\pm 0.00}$ | $1.46_{\pm 0.03}$ | $0.06_{\pm 0.02}$ | $0.56_{\pm 0.05}$ |
| RELOAD (OURS) | $99.50_{\pm 0.11}$ | $0.65_{\pm 0.72}$ | $0.04_{\pm 0.04}$ | $\mathbf{0.00}_{\pm 0.00}$ | $0.26_{\pm 0.10}$ | $0.12_{\pm 0.01}$ | $0.53_{\pm 0.08}$ |
| Retrained (Baseline) | $99.99_{\pm 0.00}$ | $95.08_{\pm 0.31}$ | $0.24_{\pm 0.02}$ | $0.50_{\pm 0.00}$ | - | - | - |

Table 6: **10% Random Forgetting on SVHN (VGG16-BN)**
$\uparrow$: the goal is to have as high of a value as possible, $\Delta^{\downarrow}$: the value in the table is the difference between the result of the unlearning method and retraining (bottom row) on the metric and the goal is to have a low difference, $\downarrow$: the goal is to have as low of a value as possible. The bottom row presents the absolute value of $M_{(\theta \sim)}$ on each metric. For any metric with $\Delta$, the raw value is instead reported. Rows for $\Delta$FA ($\downarrow$), $\Delta$FE ($\downarrow$), and $\Delta$FMIA ($\downarrow$) present the absolute difference in the value of the corresponding method on this metric to the value of $M_{(\theta \sim)}$ on the metric. These results show that RELOAD outperforms all the baselines on RA, $\Delta$FA, $\Delta$FE, and $\Delta$FMIA, by large margins aside from Fisher. However, Fisher incurs a substantially higher Cost, making it far less efficient than retraining from scratch. Therefore, Fisher is impractical, and RELOAD demonstrates the best practicality as an unlearning mechanism. RELOAD performs competitively on RSKL and FSKL but is outperformed by FT. RELOAD also incurs a higher computational cost than the other baselines.

| Method | RA ($\uparrow$) | $\Delta$FA ($\downarrow$) | $\Delta$FE ($\downarrow$) | $\Delta$FMIA ($\downarrow$) | Cost ($\downarrow$) | RSKL ($\downarrow$) | FSKL ($\downarrow$) |
|---|---|---|---|---|---|---|---|
| GA | $98.41_{\pm 0.25}$ | $26.40_{\pm 1.18}$ | $1.64_{\pm 0.07}$ | $0.14_{\pm 0.03}$ | $\mathbf{0.00}_{\pm 0.00}$ | $\mathbf{0.06}_{\pm 0.03}$ | $0.66_{\pm 0.06}$ |
| FT | $98.27_{\pm 0.20}$ | $12.65_{\pm 1.81}$ | $1.16_{\pm 0.07}$ | $0.08_{\pm 0.02}$ | $0.25_{\pm 0.03}$ | $\mathbf{0.06}_{\pm 0.01}$ | $\mathbf{0.50}_{\pm 0.03}$ |
| SSD | $22.86_{\pm 34.01}$ | $61.38_{\pm 15.72}$ | $2.57_{\pm 0.55}$ | $0.02_{\pm 0.05}$ | $\mathbf{0.00}_{\pm 0.00}$ | $8.01_{\pm 3.53}$ | $7.57_{\pm 3.07}$ |
| SCRUB | $98.43_{\pm 0.23}$ | $26.62_{\pm 1.10}$ | $1.66_{\pm 0.06}$ | $0.14_{\pm 0.03}$ | $0.02_{\pm 0.00}$ | $\mathbf{0.06}_{\pm 0.02}$ | $0.66_{\pm 0.04}$ |
| CF-$k$ | $98.30_{\pm 0.27}$ | $26.26_{\pm 1.25}$ | $1.68_{\pm 0.06}$ | $0.15_{\pm 0.02}$ | $0.27_{\pm 0.04}$ | $\mathbf{0.06}_{\pm 0.02}$ | $0.56_{\pm 0.04}$ |
| EU-$k$ | $98.35_{\pm 0.25}$ | $26.16_{\pm 1.26}$ | $1.67_{\pm 0.06}$ | $0.15_{\pm 0.02}$ | $0.27_{\pm 0.04}$ | $\mathbf{0.06}_{\pm 0.02}$ | $0.57_{\pm 0.04}$ |
| RELOAD (OURS) | $\mathbf{99.51}_{\pm 0.09}$ | $\mathbf{3.37}_{\pm 1.55}$ | $\mathbf{0.40}_{\pm 0.07}$ | $\mathbf{0.02}_{\pm 0.01}$ | $0.24_{\pm 0.11}$ | $0.11_{\pm 0.01}$ | $0.51_{\pm 0.03}$ |
| Retrained (Baseline) | $97.80_{\pm 0.33}$ | $68.25_{\pm 0.49}$ | $1.82_{\pm 0.06}$ | $0.50_{\pm 0.01}$ | - | - | - |

Table 7: **10% Random Forgetting on CIFAR-100(VGG16-BN)**
$\uparrow$: the goal is to have as high of a value as possible, $\Delta^{\downarrow}$: the value in the table is the difference between the result of the unlearning method and retraining (bottom row) on the metric and the goal is to have a low difference, $\downarrow$: the goal is to have as low of a value as possible. The bottom row presents the absolute value of $M_{(\theta \sim)}$ on each metric. For any metric with $\Delta$, the raw value is instead reported. Rows for $\Delta$FA ($\downarrow$), $\Delta$FE ($\downarrow$), and $\Delta$FMIA ($\downarrow$) present the absolute difference in the value of the corresponding method on this metric to the value of $M_{(\theta \sim)}$ on the metric. These results show that RELOAD outperforms all the baselines on RA, $\Delta$FA, $\Delta$FE, and $\Delta$FMIA, by large margins. RELOAD performs competitively on RSKL and FSKL but is outperformed by FT. RELOAD also incurs a higher computational cost than other baselines other than FT, CF-$k$, and EU-$k$.

| Method | RA ($\uparrow$) | $\Delta$FA ($\downarrow$) | $\Delta$FE ($\downarrow$) | $\Delta$FMIA ($\downarrow$) | Cost ($\downarrow$) | RSKL ($\downarrow$) | FSKL ($\downarrow$) |
|---|---|---|---|---|---|---|---|
| GA | $93.81_{\pm 0.75}$ | $18.77_{\pm 2.43}$ | $0.95_{\pm 0.14}$ | $0.21_{\pm 0.06}$ | $\mathbf{0.00}_{\pm \mathbf{0.00}}$ | $0.29_{\pm 0.09}$ | $2.62_{\pm 0.05}$ |
| FT | $96.00_{\pm 0.12}$ | $16.46_{\pm 2.47}$ | $0.89_{\pm 0.14}$ | $0.19_{\pm 0.08}$ | $0.27_{\pm 0.00}$ | $\mathbf{0.03}_{\pm \mathbf{0.01}}$ | $2.11_{\pm 0.06}$ |
| SSD | $1.01_{\pm 0.02}$ | $74.17_{\pm 2.04}$ | $4.19_{\pm 0.59}$ | $0.15_{\pm 0.21}$ | $0.01_{\pm 0.00}$ | $14.90_{\pm 1.24}$ | $11.81_{\pm 1.24}$ |
| SCRUB | $93.76_{\pm 0.74}$ | $18.85_{\pm 2.39}$ | $0.95_{\pm 0.14}$ | $0.20_{\pm 0.06}$ | $0.02_{\pm 0.00}$ | $0.29_{\pm 0.09}$ | $2.63_{\pm 0.06}$ |
| CF-$k$ | $94.75_{\pm 0.41}$ | $18.01_{\pm 2.60}$ | $0.94_{\pm 0.14}$ | $0.20_{\pm 0.06}$ | $0.21_{\pm 0.00}$ | $0.14_{\pm 0.03}$ | $2.47_{\pm 0.07}$ |
| EU-$k$ | $94.32_{\pm 0.49}$ | $17.93_{\pm 2.55}$ | $0.94_{\pm 0.14}$ | $0.20_{\pm 0.06}$ | $0.21_{\pm 0.00}$ | $0.19_{\pm 0.05}$ | $2.33_{\pm 0.05}$ |
| SalUn | $99.06_{\pm 0.22}$ | $13.14_{\pm 2.53}$ | $0.11_{\pm 0.09}$ | $7.39_{\pm 2.60}$ | $0.16_{\pm 0.00}$ | $0.06_{\pm 0.02}$ | $\mathbf{0.55}_{\pm \mathbf{0.04}}$ |
| Fisher | $97.76_{\pm 0.78}$ | $22.99_{\pm 2.30}$ | $0.95_{\pm 0.14}$ | $7.27_{\pm 2.48}$ | $1.78_{\pm 0.04}$ | $0.07_{\pm 0.02}$ | $0.56_{\pm 0.04}$ |
| RELOAD (OURS) | $\mathbf{99.56}_{\pm \mathbf{0.11}}$ | $\mathbf{0.30}_{\pm \mathbf{0.50}}$ | $\mathbf{0.04}_{\pm \mathbf{0.02}}$ | $\mathbf{0.01}_{\pm \mathbf{0.01}}$ | $0.12_{\pm 0.01}$ | $0.15_{\pm 0.03}$ | $1.23_{\pm 0.11}$ |
| Retrained (Baseline) | $99.98_{\pm 0.01}$ | $74.89_{\pm 2.03}$ | $1.06_{\pm 0.13}$ | $0.63_{\pm 0.20}$ | - | - | - |

Table 8: **10% Random Forgetting on CIFAR-100 (ResNet-18).** The bottom row presents the absolute value of $M_{(\theta \sim)}$ on each metric. For any metric with $\Delta$, the raw value is instead reported. Rows for $\Delta$FA ($\downarrow$), $\Delta$FE ($\downarrow$), and $\Delta$FMIA ($\downarrow$) present the absolute difference in the value of the corresponding method on this metric to the value of $M_{(\theta \sim)}$ on the metric. These results show that RELOAD outperforms all the baselines on RA, $\Delta$FA, $\Delta$FE, $\Delta$FMIA, and FSKL by large margins. RELOAD performs competitively on the RSKL metric, outperformed by FT and CF-$k$. RELOAD incurs a higher computational cost than most baselines, but is cheaper than FT, CF-$k$, and EU-$k$.

### B.4.2 RANDOM 30% FORGETTING

| Method | RA ($\uparrow$) | $\Delta$FA ($\downarrow$) | $\Delta$FE ($\downarrow$) | $\Delta$FMIA ($\downarrow$) | $\Delta$AUC ($\downarrow$) | Cost ($\downarrow$) | RSKL ($\downarrow$) | FSKL ($\downarrow$) |
|---|---|---|---|---|---|---|---|---|
| GA | $17.20_{\pm 30.17}$ | $77.46_{\pm 26.25}$ | $8.86_{\pm 6.48}$ | $0.02_{\pm 0.02}$ | $0.01_{\pm 0.02}$ | $\mathbf{0.01}_{\pm \mathbf{0.00}}$ | $0.06_{\pm 0.02}$ | $0.66_{\pm 0.06}$ |
| FT | $\mathbf{99.69}_{\pm \mathbf{0.24}}$ | $3.92_{\pm 0.53}$ | $0.19_{\pm 0.02}$ | $0.02_{\pm 0.01}$ | $0.02_{\pm 0.00}$ | $0.28_{\pm 0.01}$ | $\mathbf{0.05}_{\pm \mathbf{0.01}}$ | $\mathbf{0.48}_{\pm \mathbf{0.04}}$ |
| SSD | $19.85_{\pm 29.65}$ | $74.50_{\pm 25.90}$ | $1.82_{\pm 0.58}$ | $0.01_{\pm 0.02}$ | $0.01_{\pm 0.02}$ | $\mathbf{0.01}_{\pm \mathbf{0.00}}$ | $8.30_{\pm 3.11}$ | $7.83_{\pm 2.70}$ |
| SCRUB | $82.59_{\pm 1.39}$ | $12.72_{\pm 1.51}$ | $0.31_{\pm 0.04}$ | $\mathbf{0.00}_{\pm \mathbf{0.00}}$ | $\mathbf{0.00}_{\pm \mathbf{0.00}}$ | $0.07_{\pm 0.00}$ | $0.06_{\pm 0.02}$ | $0.65_{\pm 0.04}$ |
| CF-k | $99.58_{\pm 0.11}$ | $6.28_{\pm 0.19}$ | $0.27_{\pm 0.01}$ | $0.05_{\pm 0.00}$ | $0.05_{\pm 0.00}$ | $0.11_{\pm 0.00}$ | $0.06_{\pm 0.02}$ | $0.55_{\pm 0.04}$ |
| EU-k | $99.59_{\pm 0.15}$ | $6.28_{\pm 0.22}$ | $0.27_{\pm 0.01}$ | $0.05_{\pm 0.01}$ | $0.05_{\pm 0.01}$ | $0.22_{\pm 0.01}$ | $0.07_{\pm 0.02}$ | $0.56_{\pm 0.04}$ |
| SalUn | $99.63_{\pm 0.08}$ | $2.97_{\pm 0.50}$ | $0.37_{\pm 0.02}$ | $0.02_{\pm 0.02}$ | $0.02_{\pm 0.02}$ | $0.20_{\pm 0.00}$ | $0.06_{\pm 0.02}$ | $0.55_{\pm 0.04}$ |
| Fisher | $99.50_{\pm 0.18}$ | $2.37_{\pm 0.47}$ | $0.08_{\pm 0.01}$ | $0.02_{\pm 0.00}$ | $0.02_{\pm 0.01}$ | $1.79_{\pm 0.03}$ | $0.07_{\pm 0.02}$ | $0.56_{\pm 0.04}$ |
| RELOAD (OURS) | $99.51_{\pm 0.15}$ | $\mathbf{1.35}_{\pm \mathbf{0.83}}$ | $\mathbf{0.05}_{\pm \mathbf{0.02}}$ | $\mathbf{0.00}_{\pm \mathbf{0.00}}$ | $\mathbf{0.00}_{\pm \mathbf{0.00}}$ | $0.30_{\pm 0.10}$ | $0.12_{\pm 0.01}$ | $0.53_{\pm 0.07}$ |
| Retrained (Baseline) | $99.99_{\pm 0.01}$ | $94.40_{\pm 0.72}$ | $0.23_{\pm 0.08}$ | $0.50_{\pm 0.01}$ | $0.50_{\pm 0.00}$ | - | - | - |

Table 9: **30% Random Forgetting on CIFAR-10(ResNet-18)**
$\uparrow$: the goal is to have as high of a value as possible, $\Delta^{\downarrow}$: the value in the table is the difference between the result of the unlearning method and retraining (bottom row) on the metric and the goal is to have a low difference, $\downarrow$: the goal is to have as low of a value as possible. The bottom row presents the absolute value of $M_{(\theta \sim)}$ on each metric. For any metric with $\Delta$, the raw value is instead reported. Rows for $\Delta$FA ($\downarrow$), $\Delta$FE ($\downarrow$), and $\Delta$FMIA ($\downarrow$) present the absolute difference in the value of the corresponding method on this metric to the value of $M_{(\theta \sim)}$ on the metric. These results show that RELOAD outperforms all the baselines on RA, $\Delta$FA, $\Delta$FE, and $\Delta$FMIA, by large margins. RELOAD performs competitively on RSKL and FSKL but is outperformed by FT. RELOAD also incurs a higher computational cost than other baselines other than FT, CF-$k$, and EU-$k$.

| Method | RA ($\uparrow$) | $\Delta$FA ($\downarrow$) | $\Delta$FE ($\downarrow$) | $\Delta$FMIA ($\downarrow$) | $\Delta$AUC ($\downarrow$) | Cost ($\downarrow$) | RSKL ($\downarrow$) | FSKL ($\downarrow$) |
|---|---|---|---|---|---|---|---|---|
| GA | $18.97_{\pm28.44}$ | $73.93_{\pm23.07}$ | $0.43_{\pm0.01}$ | $0.05_{\pm0.03}$ | $0.01_{\pm0.01}$ | $0.01_{\pm0.00}$ | $0.06_{\pm0.02}$ | $0.66_{\pm0.06}$ |
| FT | $99.37_{\pm0.21}$ | $4.41_{\pm0.53}$ | $0.27_{\pm0.02}$ | $0.02_{\pm0.01}$ | $0.02_{\pm0.00}$ | $0.27_{\pm0.01}$ | $0.05_{\pm0.01}$ | $0.48_{\pm0.04}$ |
| SSD | $22.73_{\pm29.27}$ | $70.55_{\pm23.73}$ | $1.67_{\pm0.60}$ | $0.01_{\pm0.02}$ | $0.01_{\pm0.02}$ | $0.01_{\pm0.00}$ | $8.30_{\pm3.11}$ | $7.83_{\pm2.70}$ |
| SCRUB | $14.29_{\pm5.02}$ | $77.10_{\pm5.08}$ | $2.02_{\pm0.57}$ | $0.01_{\pm0.01}$ | $0.01_{\pm0.00}$ | $0.08_{\pm0.00}$ | $0.06_{\pm0.02}$ | $0.65_{\pm0.04}$ |
| CF-k | $99.46_{\pm0.19}$ | $8.16_{\pm0.27}$ | $0.40_{\pm0.02}$ | $0.05_{\pm0.01}$ | $0.05_{\pm0.00}$ | $0.15_{\pm0.00}$ | $0.06_{\pm0.02}$ | $0.55_{\pm0.04}$ |
| EU-k | $99.47_{\pm0.19}$ | $8.17_{\pm0.27}$ | $0.40_{\pm0.02}$ | $0.05_{\pm0.01}$ | $0.05_{\pm0.00}$ | $0.30_{\pm0.01}$ | $0.07_{\pm0.02}$ | $0.56_{\pm0.04}$ |
| SalUn | $99.73_{\pm0.06}$ | $0.90_{\pm0.25}$ | $0.25_{\pm0.01}$ | $0.01_{\pm0.01}$ | $0.01_{\pm0.00}$ | $0.18_{\pm0.00}$ | $0.06_{\pm0.02}$ | $0.55_{\pm0.04}$ |
| Fisher | $99.37_{\pm0.21}$ | $3.66_{\pm0.30}$ | $0.13_{\pm0.01}$ | $0.02_{\pm0.01}$ | $0.02_{\pm0.01}$ | $1.07_{\pm0.02}$ | $0.07_{\pm0.02}$ | $0.56_{\pm0.04}$ |
| RELOAD (OURS) | $98.43_{\pm1.49}$ | $2.46_{\pm1.63}$ | $0.07_{\pm0.05}$ | $0.00_{\pm0.00}$ | $0.00_{\pm0.00}$ | $0.57_{\pm0.13}$ | $0.12_{\pm0.01}$ | $0.53_{\pm0.07}$ |
| Retrained (Baseline) | $99.93_{\pm0.02}$ | $94.40_{\pm0.72}$ | $0.23_{\pm0.08}$ | $0.50_{\pm0.01}$ | $0.50_{\pm0.00}$ | - | - | - |

Table 10: **30% Random Forgetting on CIFAR-10(VGG16-BN)**
$\uparrow$: the goal is to have as high of a value as possible, $\Delta^{\downarrow}$: the value in the table is the difference between the result of the unlearning method and retraining (bottom row) on the metric and the goal is to have a low difference, $\downarrow$: the goal is to have as low of a value as possible. The bottom row presents the absolute value of $M_{(\theta\sim)}$ on each metric. For any metric with $\Delta$, the raw value is instead reported. Rows for $\Delta$FA ($\downarrow$), $\Delta$FE ($\downarrow$), and $\Delta$FMIA ($\downarrow$) present the absolute difference in the value of the corresponding method on this metric to the value of $M_{(\theta\sim)}$ on the metric. These results show that RELOAD outperforms all the baselines on RA, $\Delta$FA, $\Delta$FE, and $\Delta$FMIA, by large margins. RELOAD performs competitively on RSKL and FSKL but is outperformed by FT. RELOAD also incurs a higher computational cost than other baselines other than FT, CF-$k$, and EU-$k$.

| Method | RA ($\uparrow$) | $\Delta$FA ($\downarrow$) | $\Delta$FE ($\downarrow$) | $\Delta$FMIA ($\downarrow$) | $\Delta$AUC ($\downarrow$) | Cost ($\downarrow$) | RSKL ($\downarrow$) | FSKL ($\downarrow$) |
|---|---|---|---|---|---|---|---|---|
| GA | $36.61_{\pm42.78}$ | $45.98_{\pm28.62}$ | $4.71_{\pm3.85}$ | $0.06_{\pm0.08}$ | $0.06_{\pm0.07}$ | $0.01_{\pm0.00}$ | $0.06_{\pm0.02}$ | $0.66_{\pm0.06}$ |
| FT | $99.96_{\pm0.02}$ | $24.94_{\pm0.90}$ | $1.02_{\pm0.04}$ | $0.13_{\pm0.01}$ | $0.13_{\pm0.01}$ | $0.27_{\pm0.02}$ | $0.05_{\pm0.01}$ | $0.48_{\pm0.04}$ |
| SSD | $11.89_{\pm32.69}$ | $65.53_{\pm14.26}$ | $3.15_{\pm0.75}$ | $0.03_{\pm0.07}$ | $0.02_{\pm0.06}$ | $0.01_{\pm0.00}$ | $8.30_{\pm3.11}$ | $7.83_{\pm2.70}$ |
| SCRUB | $23.96_{\pm2.23}$ | $48.86_{\pm2.24}$ | $1.97_{\pm0.13}$ | $0.01_{\pm0.01}$ | $0.01_{\pm0.00}$ | $0.07_{\pm0.00}$ | $0.06_{\pm0.02}$ | $0.65_{\pm0.04}$ |
| CF-k | $98.85_{\pm0.40}$ | $21.38_{\pm1.27}$ | $0.92_{\pm0.04}$ | $0.12_{\pm0.01}$ | $0.11_{\pm0.01}$ | $0.10_{\pm0.01}$ | $0.06_{\pm0.02}$ | $0.55_{\pm0.04}$ |
| EU-k | $98.30_{\pm0.55}$ | $20.18_{\pm0.68}$ | $0.89_{\pm0.04}$ | $0.11_{\pm0.01}$ | $0.11_{\pm0.01}$ | $0.21_{\pm0.02}$ | $0.07_{\pm0.02}$ | $0.56_{\pm0.04}$ |
| SalUn | $97.33_{\pm0.30}$ | $40.31_{\pm3.78}$ | $1.20_{\pm0.04}$ | $0.10_{\pm0.01}$ | $0.10_{\pm0.01}$ | $0.20_{\pm0.00}$ | $0.06_{\pm0.02}$ | $0.55_{\pm0.04}$ |
| Fisher | $97.76_{\pm0.78}$ | $1.54_{\pm0.27}$ | $0.08_{\pm0.01}$ | $0.03_{\pm0.01}$ | $0.03_{\pm0.01}$ | $1.77_{\pm0.03}$ | $0.07_{\pm0.02}$ | $0.56_{\pm0.04}$ |
| RELOAD (OURS) | $99.56_{\pm0.06}$ | $1.47_{\pm1.05}$ | $0.08_{\pm0.05}$ | $0.01_{\pm0.01}$ | $0.00_{\pm0.00}$ | $0.32_{\pm0.04}$ | $0.12_{\pm0.01}$ | $0.53_{\pm0.07}$ |
| Retrained (Baseline) | $99.98_{\pm0.01}$ | $94.40_{\pm0.72}$ | $0.23_{\pm0.08}$ | $0.50_{\pm0.01}$ | $0.50_{\pm0.00}$ | - | - | - |

Table 11: **30% Random Forgetting on CIFAR-100(ResNet-18)**
$\uparrow$: the goal is to have as high of a value as possible, $\Delta^{\downarrow}$: the value in the table is the difference between the result of the unlearning method and retraining (bottom row) on the metric and the goal is to have a low difference, $\downarrow$: the goal is to have as low of a value as possible. The bottom row presents the absolute value of $M_{(\theta\sim)}$ on each metric. For any metric with $\Delta$, the raw value is instead reported. Rows for $\Delta$FA ($\downarrow$), $\Delta$FE ($\downarrow$), and $\Delta$FMIA ($\downarrow$) present the absolute difference in the value of the corresponding method on this metric to the value of $M_{(\theta\sim)}$ on the metric. These results show that RELOAD outperforms all the baselines on RA, $\Delta$FA, $\Delta$FE, and $\Delta$FMIA, by large margins. RELOAD performs competitively on RSKL and FSKL but is outperformed by FT. RELOAD also incurs a higher computational cost than other baselines other than FT, CF-$k$, and EU-$k$.

| Method | RA (↑) | ΔFA (↓) | ΔFE (↓) | ΔFMIA (↓) | ΔAUC (↓) | Cost (↓) | RSKL (↓) | FSKL (↓) |
|---|---|---|---|---|---|---|---|---|
| GA | $10.75_{\pm 30.87}$ | $62.76_{\pm 11.35}$ | $2.10_{\pm 0.05}$ | $0.15_{\pm 0.09}$ | $0.02_{\pm 0.05}$ | $0.01_{\pm 0.00}$ | $0.06_{\pm 0.02}$ | $0.66_{\pm 0.06}$ |
| FT | $98.30_{\pm 0.53}$ | $15.86_{\pm 1.34}$ | $1.40_{\pm 0.06}$ | $0.06_{\pm 0.01}$ | $0.06_{\pm 0.01}$ | $0.28_{\pm 0.01}$ | $0.05_{\pm 0.01}$ | $0.48_{\pm 0.04}$ |
| SSD | $11.72_{\pm 32.14}$ | $62.23_{\pm 12.36}$ | $2.43_{\pm 0.19}$ | $0.02_{\pm 0.04}$ | $0.02_{\pm 0.05}$ | $0.01_{\pm 0.00}$ | $8.30_{\pm 3.11}$ | $7.83_{\pm 2.70}$ |
| SCRUB | $1.60_{\pm 0.66}$ | $65.78_{\pm 0.92}$ | $2.39_{\pm 0.10}$ | $0.01_{\pm 0.00}$ | $0.01_{\pm 0.00}$ | $0.08_{\pm 0.00}$ | $0.06_{\pm 0.02}$ | $0.65_{\pm 0.04}$ |
| CF-k | $97.61_{\pm 0.61}$ | $29.83_{\pm 0.65}$ | $1.95_{\pm 0.04}$ | $0.14_{\pm 0.01}$ | $0.14_{\pm 0.01}$ | $0.15_{\pm 0.00}$ | $0.06_{\pm 0.02}$ | $0.55_{\pm 0.04}$ |
| EU-k | $97.71_{\pm 0.78}$ | $29.84_{\pm 0.84}$ | $1.95_{\pm 0.04}$ | $0.14_{\pm 0.01}$ | $0.14_{\pm 0.01}$ | $0.30_{\pm 0.01}$ | $0.07_{\pm 0.02}$ | $0.56_{\pm 0.04}$ |
| SalUn | $98.86_{\pm 0.27}$ | $3.28_{\pm 1.23}$ | $0.42_{\pm 0.04}$ | $0.00_{\pm 0.00}$ | $0.00_{\pm 0.00}$ | $0.18_{\pm 0.00}$ | $0.06_{\pm 0.02}$ | $0.55_{\pm 0.04}$ |
| Fisher | $97.39_{\pm 0.91}$ | $14.19_{\pm 0.81}$ | $0.56_{\pm 0.02}$ | $0.07_{\pm 0.02}$ | $0.07_{\pm 0.01}$ | $1.06_{\pm 0.02}$ | $0.07_{\pm 0.02}$ | $0.56_{\pm 0.04}$ |
| RELOAD (OURS) | $88.95_{\pm 9.23}$ | $8.94_{\pm 5.71}$ | $0.18_{\pm 0.09}$ | $0.00_{\pm 0.00}$ | $0.00_{\pm 0.00}$ | $0.60_{\pm 0.02}$ | $0.12_{\pm 0.01}$ | $0.53_{\pm 0.07}$ |
| Retrained (Baseline) | $99.85_{\pm 0.02}$ | $94.40_{\pm 0.72}$ | $0.23_{\pm 0.08}$ | $0.50_{\pm 0.01}$ | $0.50_{\pm 0.00}$ | - | - | - |

Table 12: **30% Random Forgetting on CIFAR-100(VGG16-BN)**
↑: the goal is to have as high of a value as possible, $\Delta^{\downarrow}$: the value in the table is the difference between the result of the unlearning method and retraining (bottom row) on the metric and the goal is to have a low difference, ↓: the goal is to have as low of a value as possible. The bottom row presents the absolute value of $M_{(\theta \sim)}$ on each metric. For any metric with $\Delta$, the raw value is instead reported. Rows for $\Delta$FA (↓), $\Delta$FE (↓), and $\Delta$FMIA (↓) present the absolute difference in the value of the corresponding method on this metric to the value of $M_{(\theta \sim)}$ on the metric. These results show that RELOAD outperforms all the baselines on RA, $\Delta$FA, $\Delta$FE, and $\Delta$FMIA, by large margins. RELOAD performs competitively on RSKL and FSKL but is outperformed by FT. RELOAD also incurs a higher computational cost than other baselines other than FT, CF-$k$, and EU-$k$.

| Method | RA (↑) | ΔFA (↓) | ΔFE (↓) | ΔFMIA (↓) | ΔAUC (↓) | Cost (↓) | RSKL (↓) | FSKL (↓) |
|---|---|---|---|---|---|---|---|---|
| GA | $36.70_{\pm 41.55}$ | $59.35_{\pm 39.81}$ | $6.22_{\pm 5.55}$ | $0.02_{\pm 0.03}$ | $0.02_{\pm 0.03}$ | $0.01_{\pm 0.00}$ | $0.06_{\pm 0.02}$ | $0.66_{\pm 0.06}$ |
| FT | $100.00_{\pm 0.00}$ | $4.73_{\pm 0.20}$ | $0.19_{\pm 0.01}$ | $0.04_{\pm 0.01}$ | $0.04_{\pm 0.01}$ | $0.28_{\pm 0.01}$ | $0.05_{\pm 0.01}$ | $0.48_{\pm 0.04}$ |
| SSD | $20.64_{\pm 29.80}$ | $75.32_{\pm 26.33}$ | $1.89_{\pm 0.63}$ | $0.01_{\pm 0.03}$ | $0.01_{\pm 0.03}$ | $0.01_{\pm 0.00}$ | $8.30_{\pm 3.11}$ | $7.83_{\pm 2.70}$ |
| SCRUB | $97.23_{\pm 0.29}$ | $0.49_{\pm 0.21}$ | $0.02_{\pm 0.01}$ | $0.00_{\pm 0.00}$ | $0.00_{\pm 0.00}$ | $0.08_{\pm 0.00}$ | $0.06_{\pm 0.02}$ | $0.65_{\pm 0.04}$ |
| CF-k | $100.00_{\pm 0.01}$ | $4.79_{\pm 0.22}$ | $0.19_{\pm 0.01}$ | $0.05_{\pm 0.01}$ | $0.05_{\pm 0.01}$ | $0.10_{\pm 0.00}$ | $0.06_{\pm 0.02}$ | $0.55_{\pm 0.04}$ |
| EU-k | $99.98_{\pm 0.05}$ | $4.76_{\pm 0.25}$ | $0.18_{\pm 0.01}$ | $0.05_{\pm 0.01}$ | $0.05_{\pm 0.01}$ | $0.19_{\pm 0.00}$ | $0.07_{\pm 0.02}$ | $0.56_{\pm 0.04}$ |
| SalUn | $99.65_{\pm 0.09}$ | $1.84_{\pm 0.31}$ | $0.09_{\pm 0.01}$ | $0.02_{\pm 0.01}$ | $0.02_{\pm 0.01}$ | $0.22_{\pm 0.00}$ | $0.06_{\pm 0.02}$ | $0.55_{\pm 0.04}$ |
| Fisher | $99.62_{\pm 0.14}$ | $0.09_{\pm 0.02}$ | $0.00_{\pm 0.00}$ | $0.01_{\pm 0.01}$ | $0.01_{\pm 0.01}$ | $2.12_{\pm 0.03}$ | $0.07_{\pm 0.02}$ | $0.56_{\pm 0.04}$ |
| RELOAD (OURS) | $99.58_{\pm 0.30}$ | $0.08_{\pm 0.06}$ | $0.01_{\pm 0.01}$ | $0.00_{\pm 0.01}$ | $0.00_{\pm 0.01}$ | $0.11_{\pm 0.05}$ | $0.12_{\pm 0.01}$ | $0.53_{\pm 0.07}$ |
| Retrained (Baseline) | $100.00_{\pm 0.00}$ | $94.72_{\pm 0.12}$ | $0.25_{\pm 0.01}$ | $0.50_{\pm 0.00}$ | $0.50_{\pm 0.00}$ | - | - | - |

Table 13: **30% Random Forgetting on SVHN(ResNet-18)**
↑: the goal is to have as high of a value as possible, $\Delta^{\downarrow}$: the value in the table is the difference between the result of the unlearning method and retraining (bottom row) on the metric and the goal is to have a low difference, ↓: the goal is to have as low of a value as possible. The bottom row presents the absolute value of $M_{(\theta \sim)}$ on each metric. For any metric with $\Delta$, the raw value is instead reported. Rows for $\Delta$FA (↓), $\Delta$FE (↓), and $\Delta$FMIA (↓) present the absolute difference in the value of the corresponding method on this metric to the value of $M_{(\theta \sim)}$ on the metric. These results show that RELOAD outperforms all the baselines on RA, $\Delta$FA, $\Delta$FE, and $\Delta$FMIA, by large margins. RELOAD performs competitively on RSKL and FSKL but is outperformed by FT. RELOAD also incurs a higher computational cost than other baselines other than FT, CF-$k$, and EU-$k$.

| Method | RA ($\uparrow$) | $\Delta$FA ($\downarrow$) | $\Delta$FE ($\downarrow$) | $\Delta$FMIA ($\downarrow$) | $\Delta$AUC ($\downarrow$) | Cost ($\downarrow$) | RSKL ($\downarrow$) | FSKL ($\downarrow$) |
|---|---|---|---|---|---|---|---|---|
| GA | $16.05_{\pm29.50}$ | $79.62_{\pm26.16}$ | $0.25_{\pm0.01}$ | $0.05_{\pm0.03}$ | $0.01_{\pm0.02}$ | $0.01_{\pm0.00}$ | $0.06_{\pm0.02}$ | $0.66_{\pm0.06}$ |
| FT | $100.00_{\pm0.00}$ | $4.84_{\pm0.16}$ | $0.23_{\pm0.01}$ | $0.03_{\pm0.01}$ | $0.03_{\pm0.01}$ | $0.28_{\pm0.00}$ | $0.05_{\pm0.01}$ | $0.48_{\pm0.04}$ |
| SSD | $24.17_{\pm28.49}$ | $71.83_{\pm25.07}$ | $1.85_{\pm0.60}$ | $0.01_{\pm0.02}$ | $0.01_{\pm0.02}$ | $0.01_{\pm0.00}$ | $8.30_{\pm3.11}$ | $7.83_{\pm2.70}$ |
| SCRUB | $24.26_{\pm14.07}$ | $70.72_{\pm13.55}$ | $1.85_{\pm0.38}$ | $0.01_{\pm0.00}$ | $0.01_{\pm0.00}$ | $0.08_{\pm0.00}$ | $0.06_{\pm0.02}$ | $0.65_{\pm0.04}$ |
| CF-k | $99.60_{\pm0.14}$ | $4.84_{\pm0.16}$ | $0.23_{\pm0.01}$ | $0.04_{\pm0.01}$ | $0.04_{\pm0.01}$ | $0.12_{\pm0.00}$ | $0.06_{\pm0.02}$ | $0.55_{\pm0.04}$ |
| EU-k | $99.60_{\pm0.14}$ | $4.85_{\pm0.16}$ | $0.23_{\pm0.01}$ | $0.04_{\pm0.01}$ | $0.04_{\pm0.01}$ | $0.25_{\pm0.00}$ | $0.07_{\pm0.02}$ | $0.56_{\pm0.04}$ |
| SalUn | $99.91_{\pm0.04}$ | $0.81_{\pm0.12}$ | $0.04_{\pm0.01}$ | $0.00_{\pm0.00}$ | $0.00_{\pm0.00}$ | $0.19_{\pm0.00}$ | $0.06_{\pm0.02}$ | $0.55_{\pm0.04}$ |
| Fisher | $99.53_{\pm0.16}$ | $0.04_{\pm0.03}$ | $0.00_{\pm0.00}$ | $0.00_{\pm0.00}$ | $0.00_{\pm0.00}$ | $1.43_{\pm0.01}$ | $0.07_{\pm0.02}$ | $0.56_{\pm0.04}$ |
| RELOAD (OURS) | $99.37_{\pm0.15}$ | $0.10_{\pm0.09}$ | $0.02_{\pm0.01}$ | $0.00_{\pm0.00}$ | $0.00_{\pm0.00}$ | $0.15_{\pm0.01}$ | $0.12_{\pm0.01}$ | $0.53_{\pm0.07}$ |
| Retrained (Baseline) | $100.00_{\pm0.00}$ | $94.40_{\pm0.72}$ | $0.23_{\pm0.08}$ | $0.50_{\pm0.01}$ | $0.50_{\pm0.00}$ | - | - | - |

Table 14: **30% Random Forgetting on SVHN(VGG16-BN)**
$\uparrow$: the goal is to have as high of a value as possible, $\Delta^{\downarrow}$: the value in the table is the difference between the result of the unlearning method and retraining (bottom row) on the metric and the goal is to have a low difference, $\downarrow$: the goal is to have as low of a value as possible. The bottom row presents the absolute value of $M_{(\theta\sim)}$ on each metric. For any metric with $\Delta$, the raw value is instead reported. Rows for $\Delta$FA ($\downarrow$), $\Delta$FE ($\downarrow$), and $\Delta$FMIA ($\downarrow$) present the absolute difference in the value of the corresponding method on this metric to the value of $M_{(\theta\sim)}$ on the metric. These results show that RELOAD outperforms all the baselines on RA, $\Delta$FA, $\Delta$FE, and $\Delta$FMIA, by large margins. RELOAD performs competitively on RSKL and FSKL but is outperformed by FT. RELOAD also incurs a higher computational cost than other baselines other than FT, CF-$k$, and EU-$k$.

### B.4.3 RANDOM 100 IN CLASS FORGETTING - ADDITIONAL EXPERIMENTS

| Method | RA ($\uparrow$) | FA ($\Delta^{\downarrow}$) | FE ($\Delta^{\downarrow}$) | FMIA ($\Delta^{\downarrow}$) | Cost ($\downarrow$) | RSKL ($\downarrow$) | FSKL ($\downarrow$) |
|---|---|---|---|---|---|---|---|
| GA | $99.57_{\pm0.02}$ | $4.37_{\pm0.25}$ | $0.17_{\pm0.01}$ | $0.05_{\pm0.01}$ | $\mathbf{0.00}_{\pm\mathbf{0.00}}$ | $0.05_{\pm0.00}$ | $0.52_{\pm0.02}$ |
| FT | $\mathbf{99.99}_{\pm\mathbf{0.00}}$ | $4.33_{\pm0.22}$ | $0.17_{\pm0.01}$ | $0.04_{\pm0.01}$ | $0.27_{\pm0.00}$ | $\mathbf{0.00}_{\pm\mathbf{0.00}}$ | $0.43_{\pm0.02}$ |
| SSD | $12.75_{\pm4.69}$ | $82.52_{\pm4.73}$ | $2.12_{\pm0.06}$ | $0.01_{\pm0.01}$ | $0.01_{\pm0.00}$ | $8.55_{\pm0.13}$ | $7.88_{\pm0.12}$ |
| SCRUB | $99.79_{\pm0.01}$ | $4.44_{\pm0.26}$ | $0.18_{\pm0.01}$ | $0.05_{\pm0.01}$ | $0.03_{\pm0.00}$ | $0.03_{\pm0.00}$ | $0.50_{\pm0.02}$ |
| CF-$k$ | $99.76_{\pm0.01}$ | $4.47_{\pm0.24}$ | $0.18_{\pm0.01}$ | $0.05_{\pm0.01}$ | $0.23_{\pm0.02}$ | $0.03_{\pm0.00}$ | $0.50_{\pm0.02}$ |
| EU-$k$ | $99.63_{\pm0.01}$ | $4.46_{\pm0.25}$ | $0.18_{\pm0.01}$ | $0.05_{\pm0.01}$ | $0.23_{\pm0.02}$ | $0.05_{\pm0.00}$ | $0.47_{\pm0.02}$ |
| SalUn | $99.90_{\pm0.04}$ | $3.14_{\pm1.00}$ | $0.13_{\pm0.03}$ | $0.04_{\pm0.02}$ | $0.17_{\pm0.00}$ | $0.03_{\pm0.00}$ | $0.50_{\pm0.02}$ |
| Fisher | $99.57_{\pm0.02}$ | $\mathbf{0.09}_{\pm\mathbf{0.05}}$ | $\mathbf{0.00}_{\pm\mathbf{0.00}}$ | $0.01_{\pm0.00}$ | $2.17_{\pm0.04}$ | $0.05_{\pm0.00}$ | $0.47_{\pm0.02}$ |
| RELOAD (OURS) | $99.68_{\pm0.17}$ | $0.25_{\pm0.21}$ | $0.01_{\pm0.01}$ | $\mathbf{0.00}_{\pm\mathbf{0.00}}$ | $0.12_{\pm0.01}$ | $0.06_{\pm0.02}$ | $\mathbf{0.21}_{\pm\mathbf{0.02}}$ |
| Retrained (Baseline) | $99.99_{\pm0.00}$ | $95.12_{\pm0.23}$ | $0.20_{\pm0.01}$ | $0.50_{\pm0.00}$ | - | - | - |

Table 15: **100 In Class Random Forgetting on SVHN (ResNet-18)**
$\uparrow$: the goal is to have as high of a value as possible, $\Delta^{\downarrow}$: the value in the table is the difference between the result of the unlearning method and retraining (bottom row) on the metric and the goal is to have a low difference, $\downarrow$: the goal is to have as low of a value as possible. The bottom row presents the absolute value of $M_{(\theta\sim)}$ on each metric. For any metric with $\Delta$, the raw value is instead reported. Rows for $\Delta$FA ($\downarrow$), $\Delta$FE ($\downarrow$), and $\Delta$FMIA ($\downarrow$) present the absolute difference in the value of the corresponding method on this metric to the value of $M_{(\theta\sim)}$ on the metric. These results show that RELOAD outperforms all the baselines on $\Delta$FA, $\Delta$FE, $\Delta$FMIA, and RSKL by large margins. RELOAD performs competitively on RA and FSKL but is outperformed by FT. RELOAD also incurs a higher computational cost than the other baselines.

| Method | RA (↑) | $\Delta$FA (↓) | $\Delta$FE (↓) | $\Delta$FMIA (↓) | Cost (↓) | RSKL (↓) | FSKL (↓) |
|---|---|---|---|---|---|---|---|
| GA | $98.30_{\pm0.04}$ | $5.43_{\pm0.55}$ | $0.21_{\pm0.01}$ | $0.04_{\pm0.00}$ | $\mathbf{0.00}_{\pm\mathbf{0.00}}$ | $0.07_{\pm0.00}$ | $0.63_{\pm0.04}$ |
| FT | $\mathbf{98.38}_{\pm\mathbf{0.15}}$ | $\mathbf{3.19}_{\pm\mathbf{0.41}}$ | $0.15_{\pm0.02}$ | $0.02_{\pm0.00}$ | $0.27_{\pm0.00}$ | $\mathbf{0.05}_{\pm\mathbf{0.01}}$ | $\mathbf{0.46}_{\pm\mathbf{0.03}}$ |
| SSD | $10.04_{\pm0.06}$ | $83.15_{\pm0.87}$ | $2.07_{\pm0.02}$ | $\mathbf{0.00}_{\pm\mathbf{0.00}}$ | $0.01_{\pm0.00}$ | $9.39_{\pm0.08}$ | $8.80_{\pm0.05}$ |
| SCRUB | $98.33_{\pm0.04}$ | $6.70_{\pm0.55}$ | $0.22_{\pm0.01}$ | $0.05_{\pm0.00}$ | $0.02_{\pm0.00}$ | $0.07_{\pm0.00}$ | $0.63_{\pm0.03}$ |
| CF-$k$ | $98.27_{\pm0.06}$ | $5.23_{\pm0.55}$ | $0.21_{\pm0.01}$ | $0.05_{\pm0.01}$ | $0.23_{\pm0.03}$ | $0.07_{\pm0.00}$ | $0.54_{\pm0.02}$ |
| EU-$k$ | $98.28_{\pm0.07}$ | $5.25_{\pm0.54}$ | $0.22_{\pm0.01}$ | $0.05_{\pm0.01}$ | $0.23_{\pm0.03}$ | $0.07_{\pm0.00}$ | $0.52_{\pm0.03}$ |
| SalUn | $99.74_{\pm0.04}$ | $4.11_{\pm0.45}$ | $0.27_{\pm0.02}$ | $0.01_{\pm0.01}$ | $0.16_{\pm0.00}$ | $0.07_{\pm0.00}$ | $0.54_{\pm0.02}$ |
| Fisher | $99.45_{\pm0.02}$ | $3.60_{\pm0.21}$ | $0.06_{\pm0.01}$ | $0.02_{\pm0.00}$ | $1.78_{\pm0.03}$ | $0.07_{\pm0.00}$ | $0.52_{\pm0.03}$ |
| RELOAD (OURS) | $97.00_{\pm1.09}$ | $3.46_{\pm0.86}$ | $\mathbf{0.08}_{\pm\mathbf{0.02}}$ | $0.01_{\pm0.01}$ | $0.31_{\pm0.09}$ | $0.11_{\pm0.03}$ | $0.52_{\pm0.09}$ |
| Retrained (Baseline) | $98.99_{\pm0.25}$ | $92.81_{\pm0.52}$ | $0.24_{\pm0.01}$ | $0.50_{\pm0.00}$ | - | - | - |

Table 16: **100 In Class Random Forgetting on CIFAR-10(ResNet-18)**
↑: the goal is to have as high of a value as possible, $\Delta^{\downarrow}$: the value in the table is the difference between the result of the unlearning method and retraining (bottom row) on the metric and the goal is to have a low difference, ↓: the goal is to have as low of a value as possible. The bottom row presents the absolute value of $M_{(\theta\sim)}$ on each metric. For any metric with $\Delta$, the raw value is instead reported. Rows for $\Delta$FA (↓), $\Delta$FE (↓), and $\Delta$FMIA (↓) present the absolute difference in the value of the corresponding method on this metric to the value of $M_{(\theta\sim)}$ on the metric. These results show that RELOAD outperforms all the baselines on $\Delta$FE. RELOAD performs competitively on RA, $\Delta$FA, $\Delta$FMIA, RSKL, and FSKL but is outperformed. FT which performs well, empirically makes little adjustment to the actual FA value. RELOAD also incurs a higher computational cost than the other baselines.

| Method | RA (↑) | $\Delta$FA (↓) | $\Delta$FE (↓) | $\Delta$FMIA (↓) | Cost (↓) | RSKL (↓) | FSKL (↓) |
|---|---|---|---|---|---|---|---|
| GA | $99.02_{\pm0.05}$ | $6.94_{\pm0.32}$ | $0.33_{\pm0.01}$ | $0.04_{\pm0.01}$ | $\mathbf{0.00}_{\pm\mathbf{0.00}}$ | $0.10_{\pm0.00}$ | $0.91_{\pm0.03}$ |
| FT | $98.72_{\pm0.30}$ | $3.50_{\pm0.37}$ | $0.23_{\pm0.01}$ | $0.02_{\pm0.01}$ | $0.27_{\pm0.01}$ | $\mathbf{0.08}_{\pm\mathbf{0.01}}$ | $0.65_{\pm0.03}$ |
| SSD | $9.99_{\pm0.04}$ | $81.88_{\pm0.50}$ | $2.12_{\pm0.30}$ | $0.01_{\pm0.01}$ | $0.01_{\pm0.00}$ | $10.88_{\pm0.79}$ | $10.25_{\pm0.83}$ |
| SCRUB | $97.31_{\pm3.57}$ | $5.79_{\pm2.28}$ | $0.14_{\pm0.08}$ | $0.04_{\pm0.01}$ | $0.03_{\pm0.00}$ | $1.37_{\pm0.45}$ | $1.75_{\pm0.45}$ |
| CF-$k$ | $\mathbf{99.03}_{\pm\mathbf{0.05}}$ | $6.95_{\pm0.33}$ | $0.33_{\pm0.01}$ | $0.05_{\pm0.01}$ | $0.37_{\pm0.08}$ | $0.10_{\pm0.01}$ | $0.79_{\pm0.02}$ |
| EU-$k$ | $99.02_{\pm0.05}$ | $6.96_{\pm0.35}$ | $0.33_{\pm0.01}$ | $0.05_{\pm0.01}$ | $0.37_{\pm0.08}$ | $0.10_{\pm0.00}$ | $0.78_{\pm0.04}$ |
| SalUn | $99.80_{\pm0.02}$ | $\mathbf{0.33}_{\pm\mathbf{0.36}}$ | $\mathbf{0.12}_{\pm\mathbf{0.01}}$ | $0.01_{\pm0.00}$ | $0.14_{\pm0.00}$ | $0.10_{\pm0.01}$ | $0.79_{\pm0.02}$ |
| Fisher | $99.32_{\pm0.03}$ | $3.81_{\pm0.46}$ | $0.10_{\pm0.01}$ | $0.02_{\pm0.00}$ | $1.07_{\pm0.03}$ | $0.10_{\pm0.00}$ | $0.78_{\pm0.04}$ |
| RELOAD (OURS) | $98.57_{\pm0.24}$ | $1.88_{\pm1.62}$ | $0.14_{\pm0.09}$ | $\mathbf{0.01}_{\pm\mathbf{0.01}}$ | $0.15_{\pm0.07}$ | $0.10_{\pm0.01}$ | $\mathbf{0.57}_{\pm\mathbf{0.09}}$ |
| Retrained (Baseline) | $99.56_{\pm0.08}$ | $92.02_{\pm0.32}$ | $0.37_{\pm0.01}$ | $0.50_{\pm0.01}$ | - | - | - |

Table 17: **100 In Class Random Forgetting on CIFAR-10 (VGG16-BN).** The bottom row presents the absolute value of $M_{(\theta\sim)}$ on each metric. For any metric with $\Delta$, the raw value is instead reported. Rows for $\Delta$FA (↓), $\Delta$FE (↓), and $\Delta$FMIA (↓) present the absolute difference in the value of the corresponding method on this metric to the value of $M_{(\theta\sim)}$ on the metric. These results show that RELOAD outperforms all baselines on $\Delta$FA, $\Delta$FE, $\Delta$FMIA, FSKL indicating it behaves the closes to $M_{(\theta\sim)}$ on $\mathcal{D}_{forget}$. RELOAD performs competitively on RA and RSKL, falling behind of the leading method by 0.46 for RA and 0.02 for RSKL. RELOAD incurs a higher computational cost than most baselines, but is cheaper than FT, CF-$k$, and EU-$k$. Other experimental settings are presented in Appendix B.4.3

.

| Method | RA ($\uparrow$) | FA ($\Delta^{\downarrow}$) | FE ($\Delta^{\downarrow}$) | FMIA ($\Delta^{\downarrow}$) | Cost ($\downarrow$) | RSKL ($\downarrow$) | FSKL ($\downarrow$) |
|---|---|---|---|---|---|---|---|
| GA | $98.32_{\pm0.03}$ | $23.33_{\pm1.06}$ | $1.00_{\pm0.06}$ | $0.07_{\pm0.06}$ | $\mathbf{0.00}_{\pm\mathbf{0.00}}$ | $0.07_{\pm0.00}$ | $0.65_{\pm0.06}$ |
| FT | $98.22_{\pm0.23}$ | $16.84_{\pm1.08}$ | $0.82_{\pm0.06}$ | $0.05_{\pm0.03}$ | $0.27_{\pm0.00}$ | $\mathbf{0.05}_{\pm\mathbf{0.01}}$ | $\mathbf{0.48}_{\pm\mathbf{0.04}}$ |
| SSD | $10.01_{\pm0.05}$ | $68.67_{\pm1.97}$ | $5.75_{\pm0.99}$ | $0.38_{\pm0.14}$ | $\mathbf{0.00}_{\pm\mathbf{0.00}}$ | $9.33_{\pm0.06}$ | $8.72_{\pm0.04}$ |
| SCRUB | $98.35_{\pm0.03}$ | $27.55_{\pm1.43}$ | $1.02_{\pm0.06}$ | $0.07_{\pm0.06}$ | $0.02_{\pm0.00}$ | $0.07_{\pm0.00}$ | $0.65_{\pm0.04}$ |
| CF-$k$ | $98.22_{\pm0.11}$ | $21.84_{\pm0.88}$ | $0.99_{\pm0.05}$ | $0.07_{\pm0.06}$ | $0.21_{\pm0.01}$ | $0.07_{\pm0.00}$ | $0.54_{\pm0.04}$ |
| EU-$k$ | $98.24_{\pm0.03}$ | $21.95_{\pm0.78}$ | $0.99_{\pm0.05}$ | $0.07_{\pm0.06}$ | $0.21_{\pm0.01}$ | $0.07_{\pm0.00}$ | $0.55_{\pm0.04}$ |
| SalUn | $99.57_{\pm0.02}$ | $12.08_{\pm3.13}$ | $0.48_{\pm0.07}$ | $0.02_{\pm0.02}$ | $0.14_{\pm0.00}$ | $0.07_{\pm0.00}$ | $0.54_{\pm0.04}$ |
| Fisher | $97.50_{\pm0.06}$ | $10.72_{\pm1.98}$ | $0.19_{\pm0.04}$ | $0.03_{\pm0.04}$ | $1.81_{\pm0.04}$ | $0.07_{\pm0.00}$ | $0.55_{\pm0.04}$ |
| RELOAD (OURS) | $\mathbf{99.47}_{\pm\mathbf{0.09}}$ | $\mathbf{3.44}_{\pm\mathbf{1.46}}$ | $\mathbf{0.20}_{\pm\mathbf{0.16}}$ | $\mathbf{0.02}_{\pm\mathbf{0.02}}$ | $0.26_{\pm0.11}$ | $0.12_{\pm0.01}$ | $0.53_{\pm0.08}$ |
| Retrained (Baseline) | $95.50_{\pm0.24}$ | $70.05_{\pm1.99}$ | $1.13_{\pm0.07}$ | $0.83_{\pm0.20}$ | - | - | - |

Table 18: **100 In Class Random Forgetting on CIFAR-100(ResNet-18)**
$\uparrow$: the goal is to have as high of a value as possible, $\Delta^{\downarrow}$: the value in the table is the difference between the result of the unlearning method and retraining (bottom row) on the metric and the goal is to have a low difference, $\downarrow$: the goal is to have as low of a value as possible. The bottom row presents the absolute value of $M_{(\theta\sim)}$ on each metric. For any metric with $\Delta$, the raw value is instead reported. Rows for $\Delta$FA ($\downarrow$), $\Delta$FE ($\downarrow$), and $\Delta$FMIA ($\downarrow$) present the absolute difference in the value of the corresponding method on this metric to the value of $M_{(\theta\sim)}$ on the metric. These results show that RELOAD outperforms all the baselines on RA, $\Delta$FA, $\Delta$FE, and $\Delta$FMIA, by large margins. RELOAD performs competitively on RSKL and FSKL but is outperformed by FT. RELOAD also incurs a higher computational cost than the other baselines.

| Method | RA ($\uparrow$) | FA ($\Delta^{\downarrow}$) | FE ($\Delta^{\downarrow}$) | FMIA ($\Delta^{\downarrow}$) | Cost ($\downarrow$) | RSKL ($\downarrow$) | FSKL ($\downarrow$) |
|---|---|---|---|---|---|---|---|
| GA | $98.31_{\pm0.03}$ | $28.55_{\pm2.02}$ | $1.70_{\pm0.04}$ | $0.03_{\pm0.02}$ | $\mathbf{0.00}_{\pm\mathbf{0.00}}$ | $0.07_{\pm0.00}$ | $0.65_{\pm0.04}$ |
| FT | $98.14_{\pm0.25}$ | $11.44_{\pm1.77}$ | $1.07_{\pm0.09}$ | $\mathbf{0.01}_{\pm\mathbf{0.01}}$ | $0.28_{\pm0.01}$ | $\mathbf{0.06}_{\pm\mathbf{0.01}}$ | $\mathbf{0.47}_{\pm\mathbf{0.03}}$ |
| SSD | $10.00_{\pm0.03}$ | $63.86_{\pm2.12}$ | $2.70_{\pm0.13}$ | $0.45_{\pm0.04}$ | $\mathbf{0.00}_{\pm\mathbf{0.00}}$ | $9.36_{\pm0.05}$ | $8.75_{\pm0.04}$ |
| SCRUB | $98.33_{\pm0.02}$ | $30.59_{\pm1.25}$ | $1.76_{\pm0.05}$ | $0.04_{\pm0.01}$ | $0.02_{\pm0.00}$ | $0.07_{\pm0.00}$ | $0.63_{\pm0.03}$ |
| CF-$k$ | $98.15_{\pm0.12}$ | $26.86_{\pm2.16}$ | $1.75_{\pm0.07}$ | $0.04_{\pm0.01}$ | $0.34_{\pm0.07}$ | $0.07_{\pm0.00}$ | $0.54_{\pm0.03}$ |
| EU-$k$ | $98.22_{\pm0.04}$ | $25.37_{\pm1.35}$ | $1.68_{\pm0.06}$ | $0.03_{\pm0.02}$ | $0.33_{\pm0.07}$ | $0.07_{\pm0.00}$ | $0.55_{\pm0.03}$ |
| SalUn | $99.40_{\pm0.04}$ | $7.56_{\pm0.47}$ | $0.31_{\pm0.16}$ | $0.00_{\pm0.00}$ | $0.13_{\pm0.00}$ | $0.07_{\pm0.00}$ | $0.54_{\pm0.03}$ |
| Fisher | $97.16_{\pm0.03}$ | $19.55_{\pm0.59}$ | $0.67_{\pm0.05}$ | $0.03_{\pm0.00}$ | $1.05_{\pm0.04}$ | $0.07_{\pm0.00}$ | $0.55_{\pm0.03}$ |
| RELOAD (OURS) | $\mathbf{99.47}_{\pm\mathbf{0.04}}$ | $\mathbf{1.84}_{\pm\mathbf{1.26}}$ | $\mathbf{0.14}_{\pm\mathbf{0.04}}$ | $0.03_{\pm0.02}$ | $0.29_{\pm0.01}$ | $0.12_{\pm0.01}$ | $0.51_{\pm0.02}$ |
| Retrained (Baseline) | $93.85_{\pm1.04}$ | $65.26_{\pm2.16}$ | $1.95_{\pm0.10}$ | $0.93_{\pm0.02}$ | - | - | - |

Table 19: **100 In Class Random Forgetting on CIFAR-100(VGG16-BN)**
$\uparrow$: the goal is to have as high of a value as possible, $\Delta^{\downarrow}$: the value in the table is the difference between the result of the unlearning method and retraining (bottom row) on the metric and the goal is to have a low difference, $\downarrow$: the goal is to have as low of a value as possible. The bottom row presents the absolute value of $M_{(\theta\sim)}$ on each metric. For any metric with $\Delta$, the raw value is instead reported. Rows for $\Delta$FA ($\downarrow$), $\Delta$FE ($\downarrow$), and $\Delta$FMIA ($\downarrow$) present the absolute difference in the value of the corresponding method on this metric to the value of $M_{(\theta\sim)}$ on the metric. These results show that RELOAD outperforms all the baselines on RA, $\Delta$FA, and $\Delta$FE by large margins. RELOAD performs competitively on $\Delta$FMIA, RSKL and FSKL but is outperformed by FT. RELOAD also incurs a higher computational cost than the other baselines.

| Method | RA ($\uparrow$) | FA ($\Delta^{\downarrow}$) | FE ($\Delta^{\downarrow}$) | FMIA ($\Delta^{\downarrow}$) | Cost ($\downarrow$) | RSKL ($\downarrow$) | FSKL ($\downarrow$) |
|---|---|---|---|---|---|---|---|
| GA | $99.57_{\pm 0.02}$ | $4.46_{\pm 0.24}$ | $0.22_{\pm 0.01}$ | $0.03_{\pm 0.01}$ | $\mathbf{0.00_{\pm 0.00}}$ | $0.05_{\pm 0.00}$ | $0.51_{\pm 0.02}$ |
| FT | $\mathbf{99.99_{\pm 0.001}}$ | $4.47_{\pm 0.23}$ | $0.22_{\pm 0.01}$ | $0.03_{\pm 0.01}$ | $0.27_{\pm 0.00}$ | $\mathbf{0.00_{\pm 0.00}}$ | $0.43_{\pm 0.02}$ |
| SSD | $14.55_{\pm 3.93}$ | $84.19_{\pm 1.55}$ | $2.05_{\pm 0.01}$ | $\mathbf{0.00_{\pm 0.00}}$ | $0.01_{\pm 0.00}$ | $8.51_{\pm 0.03}$ | $7.84_{\pm 0.02}$ |
| SCRUB | $99.79_{\pm 0.01}$ | $9.55_{\pm 9.76}$ | $0.36_{\pm 0.34}$ | $0.03_{\pm 0.01}$ | $0.02_{\pm 0.00}$ | $0.03_{\pm 0.00}$ | $0.50_{\pm 0.03}$ |
| CF-$k$ | $99.76_{\pm 0.01}$ | $4.53_{\pm 0.25}$ | $0.23_{\pm 0.01}$ | $0.04_{\pm 0.01}$ | $0.24_{\pm 0.02}$ | $0.03_{\pm 0.00}$ | $0.50_{\pm 0.02}$ |
| EU-$k$ | $99.63_{\pm 0.02}$ | $4.54_{\pm 0.23}$ | $0.23_{\pm 0.01}$ | $0.04_{\pm 0.01}$ | $0.24_{\pm 0.02}$ | $0.05_{\pm 0.00}$ | $0.47_{\pm 0.02}$ |
| SalUn | $99.94_{\pm 0.01}$ | $5.04_{\pm 1.37}$ | $0.16_{\pm 0.04}$ | $0.03_{\pm 0.01}$ | $0.14_{\pm 0.01}$ | $0.03_{\pm 0.00}$ | $0.50_{\pm 0.02}$ |
| Fisher | $99.48_{\pm 0.02}$ | $0.09_{\pm 0.06}$ | $0.00_{\pm 0.00}$ | $0.00_{\pm 0.00}$ | $1.42_{\pm 0.14}$ | $0.05_{\pm 0.00}$ | $0.47_{\pm 0.02}$ |
| RELOAD (OURS) | $99.67_{\pm 0.14}$ | $\mathbf{0.93_{\pm 1.21}}$ | $\mathbf{0.05_{\pm 0.06}}$ | $0.01_{\pm 0.01}$ | $0.14_{\pm 0.08}$ | $0.06_{\pm 0.02}$ | $\mathbf{0.21_{\pm 0.02}}$ |
| Retrained (Baseline) | $99.999_{\pm 0.001}$ | $95.09_{\pm 0.19}$ | $0.20_{\pm 0.01}$ | $0.50_{\pm 0.00}$ | - | - | - |

Table 20: **100 In Class Random Forgetting on SVHN (VGG16-BN)**
$\uparrow$: the goal is to have as high of a value as possible, $\Delta^{\downarrow}$: the value in the table is the difference between the result of the unlearning method and retraining (bottom row) on the metric and the goal is to have a low difference, $\downarrow$: the goal is to have as low of a value as possible. The bottom row presents the absolute value of $M_{(\theta\sim)}$ on each metric. For any metric with $\Delta$, the raw value is instead reported. Rows for $\Delta$FA ($\downarrow$), $\Delta$FE ($\downarrow$), and $\Delta$FMIA ($\downarrow$) present the absolute difference in the value of the corresponding method on this metric to the value of $M_{(\theta\sim)}$ on the metric. These results show that RELOAD outperforms all the baselines on $\Delta$FA, $\Delta$FE, and FSKL, by large margins. RELOAD performs competitively on RA, $\Delta$FMIA, and RSKL but is outperformed by FT. RELOAD also incurs a higher computational cost than the other baselines.

## B.5 LANGUAGE MODEL ENTITY UNLEARNING RESULTS

**Unlearning for language models (LMs).** When $\mathcal{D}$ is a corpus of texts, we express forgetting via a set of prompts $\mathcal{D}_{prompts}$ that target concepts or entities in $\mathcal{D}_{forget}$, and possibly a small repair set $\mathcal{D}_{repair} \subseteq \mathcal{D}_{retain}$.

The results presented below are taken from prior work (Liu et al., 2024a) with results for RELOAD appended to the bottom due to computational constraints. In these result tables, the gold-standard retrained model is denoted 'Retain'.

Due to computational constraints and the lack of open-source retrained models for Phi-1.5 in the 1% and 5% forgetting case, our results for Phi-1.5 are limited to the 10% forgetting case.

| Split | Method | Change in Model Utility from Original | Forget Quality |
|---|---|---|---|
| | Original | +0.0000 | 0.0030 |
| | Retain | -0.0131 | 1.0000 |
| | Grad Ascent | -0.0233 | 0.0068 |
| | Grad Diff | -0.0198 | 0.0143 |
| 1% | KL Min | -0.0221 | 0.0068 |
| | Pref Opt | -0.0021 | 0.0971 |
| | Prompt | -0.0628 | 0.0068 |
| | NPO | -0.1725 | 0.7659 |
| | NPO-KL | -0.1703 | 0.4046 |
| | NPO-RT | -0.1361 | 0.5786 |
| | ECO (Rand Noise) | +0.0000 | 0.9188 |
| | ECO (Zero-Out) | +0.0000 | 0.9900 |
| | ECO (Sign-Flip) | +0.0000 | 0.0002 |
| | RELOAD (OURS) | +0.0748 | 0.4046 |
| | Original | +0.0000 | 0.0000 |
| | Retain | -0.0229 | 1.0000 |
| | Grad Ascent | -0.6257 | 0.0118 |
| | Grad Diff | -0.3013 | 0.0000 |
| 5% | KL Min | -0.6257 | 0.0163 |
| | Pref Opt | -0.1472 | 0.0000 |
| | Prompt | -0.1063 | 0.0000 |
| | NPO | -0.4512 | 0.7934 |
| | NPO-KL | -0.2203 | 0.7934 |
| | NPO-RT | -0.0838 | 0.6284 |
| | ECO (Rand Noise) | +0.0000 | 0.9647 |
| | ECO (Zero-Out) | -0.0009 | 0.9647 |
| | ECO (Sign-Flip) | +0.0000 | 0.0000 |
| | RELOAD (OURS) | -0.2870 | 0.5453 |
| | Original | 0.0000 | 0.0000 |
| | Retain | -0.0160 | 1.0000 |
| | Grad Ascent | -0.6257 | 0.0000 |
| | Grad Diff | -0.0434 | 0.0000 |
| 10% | KL Min | -0.6257 | 0.1810 |
| | Pref Opt | -0.0862 | 0.0000 |
| | Prompt | -0.1380 | 0.0000 |
| | NPO | -0.4556 | 0.0126 |
| | NPO-KL | -0.2634 | 0.0158 |
| | NPO-RT | -0.1260 | 0.0783 |
| | ECO (Rand Noise) | -0.0028 | 0.5812 |
| | ECO (Zero-Out) | -0.0014 | 0.9674 |
| | ECO (Sign-Flip) | -0.0022 | 0.0000 |
| | RELOAD (OURS) | -0.3384 | 0.7000 |

Table 21: Change in Model Utility and Forget Quality of different unlearning methods on unlearning entities from the TOFU dataset on

| Split | Method | Change in Model Utility from Original | Forget Quality |
|---|---|---|---|
| | Original | 0.0000 | 0.0000 |
| | Retain | +0.0053 | 1.0000 |
| | Grad Ascent | -0.5518 | 0.2107 |
| | Grad Diff | -0.1999 | 0.0000 |
| 10% | KL Min | -0.5518 | 0.4158 |
| | Pref Opt | -0.0379 | 0.0000 |
| | Prompt | -0.0363 | 0.0000 |
| | NPO | -0.3669 | 0.0013 |
| | NPO-KL | -0.2520 | 0.0049 |
| | NPO-RT | -0.1144 | 0.7000 |
| | ECO (Rand Noise) | -0.0003 | 0.8635 |
| | ECO (Zero-Out) | -0.0031 | 0.9844 |
| | ECO (Sign-Flip) | -0.0001 | 0.0446 |
| | RELOAD (OURS) | -0.3384 | 0.4680 |

Table 22: Change in Model Utility and Forget Quality of different unlearning methods on unlearning entities from the TOFU dataset on Phi-1.5

## B.6 CORRECTIVE UNLEARNING RESULTS

**Baselines.** The corrective unlearning setting admits different baselines than the unlearning setting based on prior work (Goel et al., 2024). RewoD represents a baseline model trained directly on $\mathcal{D}_{retain}$.

**Evaluation.** We evaluate corrective unlearning following Goel et al. (2024). The corrected accuracy $Acc_{corr}$ measures the performance of the unlearned model on the adversely affected data, $\mathcal{D}_m$. The retain accuracy $Acc_{retain}$ measures unlearned model performance on a held-out validation sample of $\mathcal{D}_{retain}$, $\mathcal{D}_{retain}^{(test)}$. Cost measures the runtime of the algorithm in comparison to retraining (Table 2).

**Reload efficiently corrects trained models.** We evaluate RELOAD 's ability to unlearn adverse effects of manipulations following the baselines outlined in prior work (Goel et al., 2024). We present results for RELOAD and unlearning baselines on the two conventional corrective unlearning tasks, Poisoning and Interclass Confusion (IC), as well as the corrective unlearning (with replacement) settings introduced in Appendix A.5. The results of this experiment over different settings are presented in Table 24 and Figures 9, 10, 11, and 12 in Appendix B.6 for consistency with prior work. RELOAD outperforms baselines on $Acc_{corr}$ at low percentages of data identification (Figures 8a, 8b) while observing competitive computational efficiency (Table 23), even at only 10% data identification ($\gamma$ = 0.1). Across 10 values of $\gamma$ from the corrective setting (Goel et al., 2024), RELOAD consistently outperforms all other baselines in most experiments. Although BadT (Chundawat et al., 2022) outperforms RELOAD in CIFAR100 Poisoning experiments, it bears much greater computational cost (Table 23).

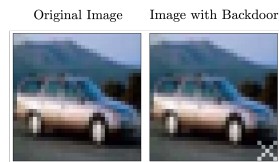

Original Image    Image with Backdoor

Figure 6: Data poisoning inserts the patterns (*right*) in all selected images in $\mathcal{D}$.

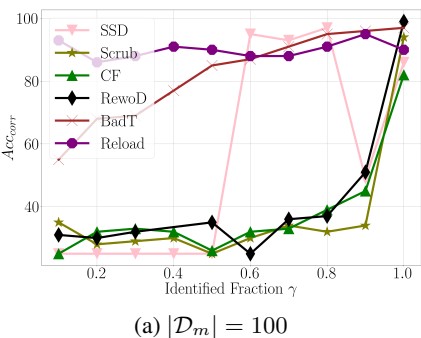 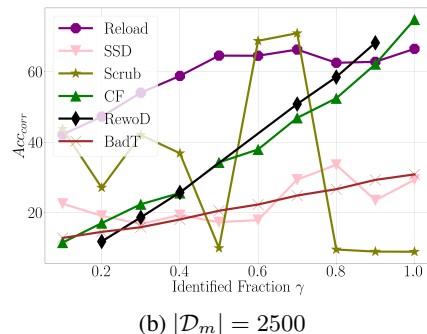

(a) $|\mathcal{D}_m| = 100$        (b) $|\mathcal{D}_m| = 2500$

Figure 7: Corrective Accuracy ($\text{Acc}_{\text{corr}}$) after applying different unlearning methods. This measures the performance of the unlearned model on the domain representing the adversely affected data, $\mathcal{D}_m$. $\gamma$ measures the proportion of the adversely affected data which was identified and collected within $\mathcal{D}_m$. We note that at small $\gamma$, RELOAD achieves consistently higher $\text{Acc}_{\text{corr}}$ than existing baselines and performs across $\gamma$ values.

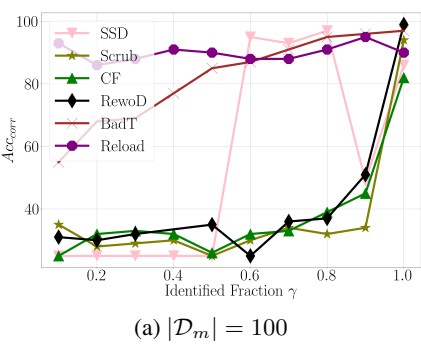 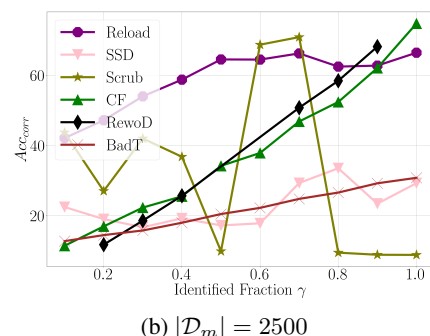

(a) $|\mathcal{D}_m| = 100$        (b) $|\mathcal{D}_m| = 2500$

Figure 8: Corrective Accuracy ($\text{Acc}_{\text{corr}}$) after applying different unlearning methods. This measures the performance of the unlearned model on the domain representing the adversely affected data, $\mathcal{D}_m$. $\gamma$ measures the proportion of the adversely affected data which was identified and collected within $\mathcal{D}_{forget}$. We note that at small $\gamma$, RELOAD achieves consistently higher $\text{Acc}_{\text{corr}}$ than existing baselines and performs across $\gamma$ values.

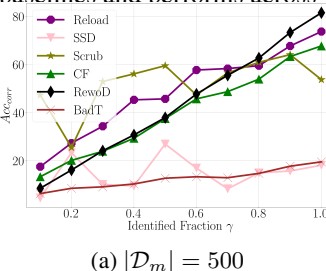 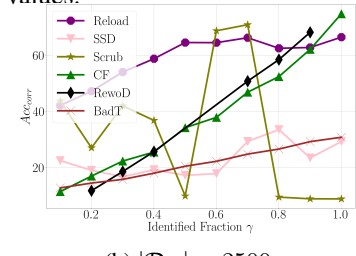 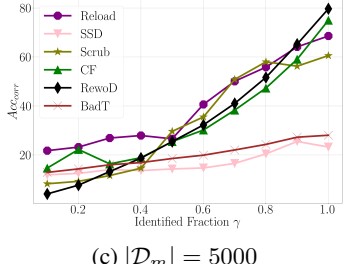

(a) $|\mathcal{D}_m| = 500$     (b) $|\mathcal{D}_m| = 2500$     (c) $|\mathcal{D}_m| = 5000$

Figure 9: CIFAR10 Interclass Confusion

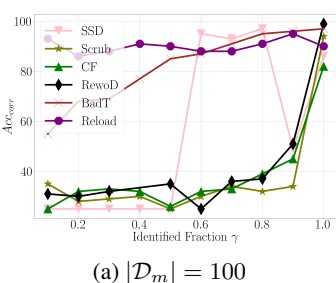 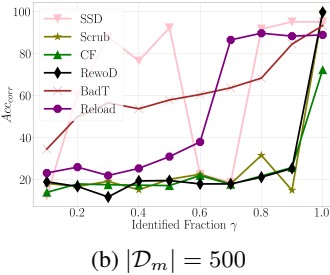 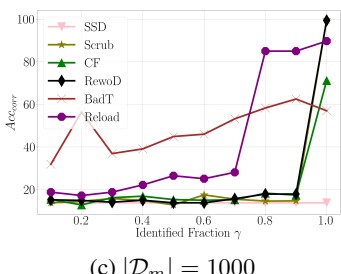

(a) $|\mathcal{D}_m| = 100$     (b) $|\mathcal{D}_m| = 500$     (c) $|\mathcal{D}_m| = 1000$

Figure 10: CIFAR10 Poisoning

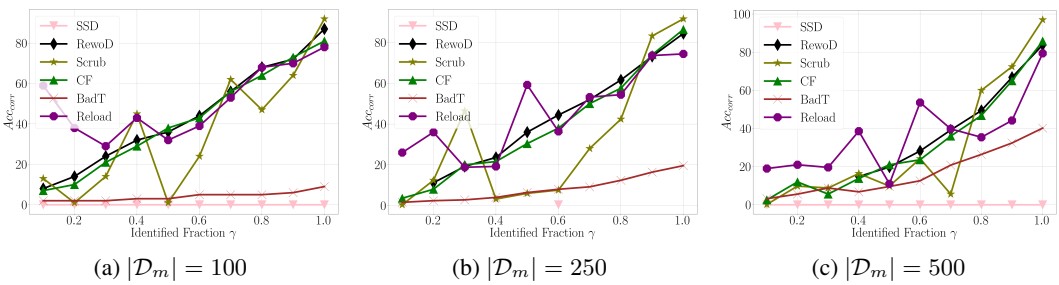

Figure 11: CIFAR100 Interclass Confusion

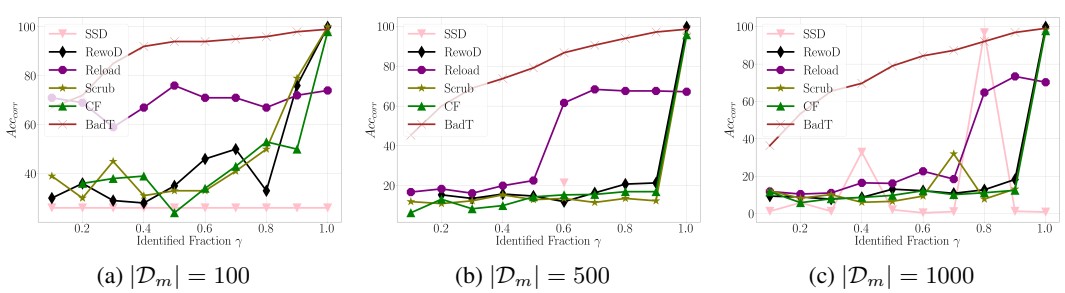

Figure 12: CIFAR100 Poisoning

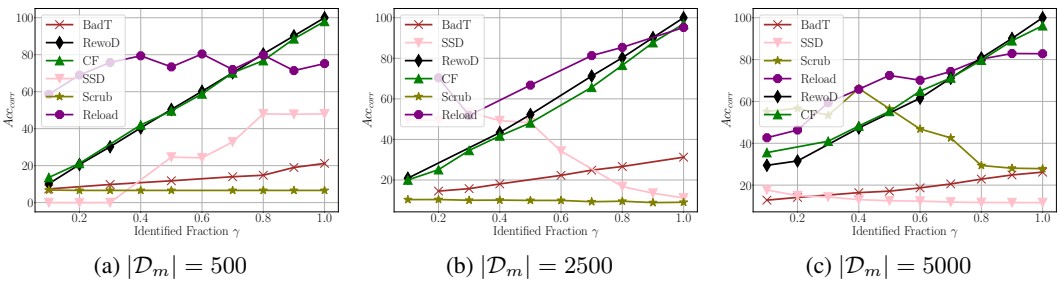

Figure 13: CIFAR10 Interclass Confusion (with replacement)

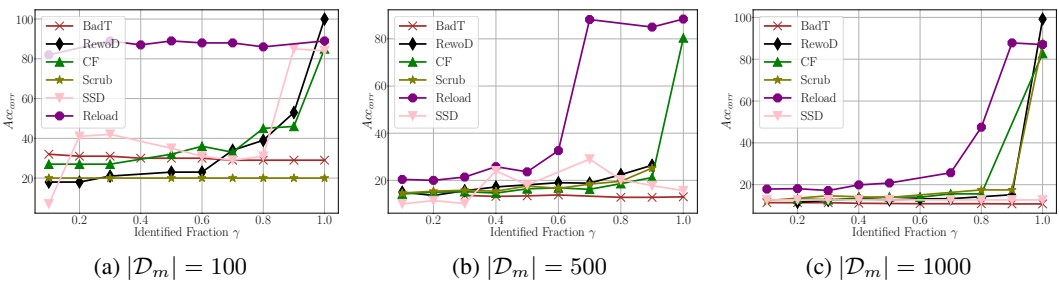

Figure 14: CIFAR10 Poisoning (with replacement)

| Cost (↓) | CIFAR10 | CIFAR100 |
|---|---|---|
| **Poisoning** | | |
| BadT | $0.44_{\pm 0.00}$ | $0.68_{\pm 0.00}$ |
| CF | $0.26_{\pm 0.00}$ | $0.25_{\pm 0.00}$ |
| SSD | $0.04_{\pm 0.00}$ | $0.06_{\pm 0.00}$ |
| Scrub | $0.26_{\pm 0.00}$ | $0.31_{\pm 0.00}$ |
| RewoD | $1.00_{\pm 0.00}$ | $1.00_{\pm 0.00}$ |
| RELOAD | $0.37_{\pm 0.02}$ | $0.24_{\pm 0.00}$ |
| **Interclass Confusion (IC)** | | |
| BadT | $0.42_{\pm 0.01}$ | $0.68_{\pm 0.02}$ |
| CF | $0.23_{\pm 0.00}$ | $0.26_{\pm 0.01}$ |
| SSD | $0.04_{\pm 0.00}$ | $0.06_{\pm 0.00}$ |
| Scrub | $0.24_{\pm 0.00}$ | $0.31_{\pm 0.01}$ |
| RewoD | $1.00_{\pm 0.00}$ | $1.00_{\pm 0.00}$ |
| RELOAD | $0.28_{\pm 0.01}$ | $0.14_{\pm 0.00}$ |
| **Num. Corrupted Samples** | | |
| CIFAR10 | Poison | 100 |
| CIFAR100 | Poison | 100 |
| CIFAR10 | IC | 500 |
| CIFAR100 | IC | 100 |

| Cost (↓) | CIFAR10 | CIFAR100 |
|---|---|---|
| **Poisoning** | | |
| BadT | $0.46_{\pm 0.01}$ | $0.68_{\pm 0.00}$ |
| CF | $0.27_{\pm 0.00}$ | $0.25_{\pm 0.00}$ |
| SSD | $0.04_{\pm 0.00}$ | $0.06_{\pm 0.00}$ |
| Scrub | $0.27_{\pm 0.00}$ | $0.31_{\pm 0.00}$ |
| RewoD | $1.00_{\pm 0.00}$ | $1.00_{\pm 0.00}$ |
| RELOAD | $0.29_{\pm 0.01}$ | $0.25_{\pm 0.00}$ |
| **Interclass Confusion (IC)** | | |
| BadT | $0.46_{\pm 0.00}$ | $0.68_{\pm 0.01}$ |
| CF | $0.27_{\pm 0.00}$ | $0.25_{\pm 0.00}$ |
| SSD | $0.04_{\pm 0.00}$ | $0.06_{\pm 0.00}$ |
| Scrub | $0.27_{\pm 0.01}$ | $0.31_{\pm 0.00}$ |
| RewoD | $1.00_{\pm 0.00}$ | $1.00_{\pm 0.00}$ |
| RELOAD | $0.40_{\pm 0.00}$ | $0.15_{\pm 0.00}$ |
| **Num. Corrupted Samples** | | |
| CIFAR10 | Poison | 500 |
| CIFAR100 | Poison | 500 |
| CIFAR10 | IC | 2500 |
| CIFAR100 | IC | 250 |

| Cost (↓) | CIFAR10 | CIFAR100 |
|---|---|---|
| **Poisoning** | | |
| BadT | $0.46_{\pm 0.01}$ | $0.68_{\pm 0.00}$ |
| CF | $0.27_{\pm 0.00}$ | $0.25_{\pm 0.00}$ |
| SSD | $0.04_{\pm 0.00}$ | $0.06_{\pm 0.00}$ |
| Scrub | $0.27_{\pm 0.00}$ | $0.31_{\pm 0.00}$ |
| RewoD | $1.00_{\pm 0.00}$ | $1.00_{\pm 0.00}$ |
| RELOAD | $0.33_{\pm 0.02}$ | $0.25_{\pm 0.00}$ |
| **Interclass Confusion (IC)** | | |
| BadT | $0.43_{\pm 0.02}$ | $0.68_{\pm 0.00}$ |
| CF | $0.27_{\pm 0.01}$ | $0.25_{\pm 0.00}$ |
| SSD | $0.04_{\pm 0.00}$ | $0.06_{\pm 0.00}$ |
| Scrub | $0.27_{\pm 0.01}$ | $0.31_{\pm 0.00}$ |
| RewoD | $1.00_{\pm 0.00}$ | $1.00_{\pm 0.00}$ |
| RELOAD | $0.14_{\pm 0.00}$ | $0.14_{\pm 0.00}$ |
| **Num. Corrupted Samples** | | |
| CIFAR10 | Poison | 1000 |
| CIFAR100 | Poison | 1000 |
| CIFAR10 | IC | 5000 |
| CIFAR100 | IC | 500 |

Table 23: Cost (↓) values across different sizes of $\mathcal{D}_{forget}$ (Corrective Unlearning). Results are reported as mean$_{\pm\text{stddev}}$ over 10 values of $\gamma$.

| $Acc_{retain}$ (↑) | CIFAR10 | CIFAR100 |
|---|---|---|
| **Poisoning** | | |
| None | 91.35 | 74.05 |
| BadT | $0.13_{\pm 0.04}$ | $0.13_{\pm 0.05}$ |
| CF | $0.09_{\pm 0.12}$ | $0.54_{\pm 0.17}$ |
| SSD | $-3.05_{\pm 4.56}$ | $-2.08_{\pm 0.00}$ |
| Scrub | $0.01_{\pm 0.11}$ | $0.25_{\pm 0.23}$ |
| RewoD | $0.86_{\pm 0.00}$ | $1.14_{\pm 0.00}$ |
| RELOAD | $-7.83_{\pm 0.20}$ | $-13.20_{\pm 0.69}$ |
| **Interclass Confusion (IC)** | | |
| None | 93.01 | 73.82 |
| BadT | $0.39_{\pm 0.09}$ | $0.22_{\pm 0.04}$ |
| CF | $0.16_{\pm 0.16}$ | $0.63_{\pm 0.14}$ |
| SSD | $-1.45_{\pm 4.44}$ | $0.14_{\pm 0.00}$ |
| Scrub | $0.19_{\pm 0.19}$ | $0.17_{\pm 0.06}$ |
| RewoD | $0.82_{\pm 0.00}$ | $1.29_{\pm 0.00}$ |
| RELOAD | $-0.20_{\pm 0.22}$ | $-4.45_{\pm 0.87}$ |
| **Num. Corrupted Samples** | | |
| CIFAR10 | Poison | 100 |
| CIFAR100 | Poison | 100 |
| CIFAR10 | IC | 500 |
| CIFAR100 | IC | 100 |

| $Acc_{retain}$ (↑) | CIFAR10 | CIFAR100 |
|---|---|---|
| **Poisoning** | | |
| None | 90.97 | 74.20 |
| BadT | $-0.05_{\pm 0.17}$ | $-0.32_{\pm 0.11}$ |
| CF | $0.35_{\pm 0.14}$ | $0.32_{\pm 0.24}$ |
| SSD | $-14.03_{\pm 22.86}$ | $0.00_{\pm 0.00}$ |
| Scrub | $0.43_{\pm 0.04}$ | $0.20_{\pm 0.19}$ |
| RewoD | $1.25_{\pm 0.00}$ | $0.71_{\pm 0.00}$ |
| RELOAD | $-8.14_{\pm 0.22}$ | $-13.22_{\pm 0.54}$ |
| **Interclass Confusion (IC)** | | |
| None | 92.22 | 74.06 |
| BadT | $0.81_{\pm 0.12}$ | $0.05_{\pm 0.07}$ |
| CF | $0.59_{\pm 0.30}$ | $0.46_{\pm 0.18}$ |
| SSD | $0.73_{\pm 0.17}$ | $0.00_{\pm 0.00}$ |
| Scrub | $0.63_{\pm 0.56}$ | $-0.06_{\pm 0.14}$ |
| RewoD | $1.14_{\pm 0.00}$ | $0.99_{\pm 0.00}$ |
| RELOAD | $-2.18_{\pm 1.43}$ | $-5.21_{\pm 0.76}$ |
| **Num. Corrupted Samples** | | |
| CIFAR10 | Poison | 500 |
| CIFAR100 | Poison | 500 |
| CIFAR10 | IC | 2500 |
| CIFAR100 | IC | 250 |

| $Acc_{retain}$ (↑) | CIFAR10 | CIFAR100 |
|---|---|---|
| **Poisoning** | | |
| None | 90.84 | 74.34 |
| BadT | $0.01_{\pm 0.07}$ | $-0.24_{\pm 0.22}$ |
| CF | $0.27_{\pm 0.10}$ | $0.39_{\pm 0.19}$ |
| SSD | $-0.66_{\pm 1.47}$ | $-0.61_{\pm 0.60}$ |
| Scrub | $0.27_{\pm 0.05}$ | $0.00_{\pm 0.11}$ |
| RewoD | $0.92_{\pm 0.00}$ | $1.18_{\pm 0.00}$ |
| RELOAD | $-7.55_{\pm 0.24}$ | $-13.63_{\pm 0.61}$ |
| **Interclass Confusion (IC)** | | |
| None | 92.81 | 73.81 |
| BadT | $0.52_{\pm 0.11}$ | $-0.02_{\pm 0.13}$ |
| CF | $0.49_{\pm 0.31}$ | $0.53_{\pm 0.17}$ |
| SSD | $0.72_{\pm 0.31}$ | $0.00_{\pm 0.00}$ |
| Scrub | $-1.10_{\pm 1.85}$ | $-0.05_{\pm 0.18}$ |
| RewoD | $0.95_{\pm 0.00}$ | $1.13_{\pm 0.00}$ |
| RELOAD | $-1.14_{\pm 0.54}$ | $-4.93_{\pm 0.81}$ |
| **Num. Corrupted Samples** | | |
| CIFAR10 | Poison | 1000 |
| CIFAR100 | Poison | 1000 |
| CIFAR10 | IC | 5000 |
| CIFAR100 | IC | 500 |

Table 24: Acc$_{\text{retain}}$ (↑) values across different sizes of $\mathcal{D}_{forget}$ (Corrective Unlearning). Results are reported as mean$_{\pm\text{stddev}}$ over 10 values of $\gamma$.

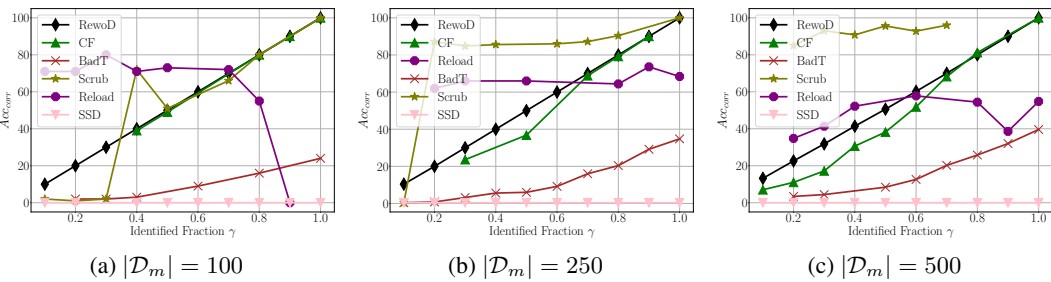

(a) $|\mathcal{D}_m| = 100$    (b) $|\mathcal{D}_m| = 250$    (c) $|\mathcal{D}_m| = 500$

Figure 15: CIFAR100 Interclass Confusion (with replacement)

| Cost (↓) | CIFAR10 | CIFAR100 |
|---|---|---|
| **Poisoning** | | |
| BadT | 0.63 ±0.01 | 0.69 ±0.00 |
| CF | 0.26 ±0.01 | 0.25 ±0.00 |
| SSD | 0.06 ±0.01 | 0.06 ±0.00 |
| Scrub | 0.27 ±0.01 | 0.30 ±0.00 |
| RewoD | 1.00 ±0.04 | 1.00 ±0.01 |
| RELOAD | 0.44 ±0.03 | 0.31 ±0.00 |
| **Interclass Confusion (IC)** | | |
| BadT | 0.62 ±0.01 | 0.68 ±0.01 |
| CF | 0.25 ±0.01 | 0.25 ±0.00 |
| SSD | 0.06 ±0.01 | 0.06 ±0.00 |
| Scrub | 0.27 ±0.01 | 0.30 ±0.00 |
| RewoD | 1.00 ±0.04 | 1.00 ±0.04 |
| RELOAD | 0.35 ±0.01 | 0.20 ±0.00 |
| **Num. Corrupted Samples** | | |
| CIFAR10 | Poison | 100 |
| CIFAR100 | Poison | 100 |
| CIFAR10 | IC | 500 |
| CIFAR100 | IC | 100 |

| Cost (↓) | CIFAR10 | CIFAR100 |
|---|---|---|
| **Poisoning** | | |
| BadT | 0.64 ±0.01 | 0.70 ±0.00 |
| CF | 0.26 ±0.01 | 0.25 ±0.00 |
| SSD | 0.06 ±0.01 | 0.06 ±0.00 |
| Scrub | 0.28 ±0.01 | 0.31 ±0.00 |
| RewoD | 1.00 ±0.04 | 1.00 ±0.01 |
| RELOAD | 0.42 ±0.02 | 0.31 ±0.00 |
| **Interclass Confusion (IC)** | | |
| BadT | 0.63 ±0.01 | 0.67 ±0.00 |
| CF | 0.27 ±0.01 | 0.24 ±0.00 |
| SSD | 0.06 ±0.01 | 0.06 ±0.00 |
| Scrub | 0.27 ±0.01 | 0.30 ±0.00 |
| RewoD | 1.00 ±0.05 | 1.00 ±0.05 |
| RELOAD | 0.46 ±0.03 | 0.20 ±0.00 |
| **Num. Corrupted Samples** | | |
| CIFAR10 | Poison | 500 |
| CIFAR100 | Poison | 500 |
| CIFAR10 | IC | 2500 |
| CIFAR100 | IC | 250 |

| Cost (↓) | CIFAR10 | CIFAR100 |
|---|---|---|
| **Poisoning** | | |
| BadT | 0.64 ±0.01 | 0.68 ±0.01 |
| CF | 0.25 ±0.01 | 0.25 ±0.00 |
| SSD | 0.06 ±0.00 | 0.06 ±0.00 |
| Scrub | 0.28 ±0.01 | 0.30 ±0.00 |
| RewoD | 1.00 ±0.03 | 1.00 ±0.04 |
| RELOAD | 0.40 ±0.02 | 0.30 ±0.00 |
| **Interclass Confusion (IC)** | | |
| BadT | 0.64 ±0.02 | 0.68 ±0.00 |
| CF | 0.27 ±0.01 | 0.25 ±0.00 |
| SSD | 0.06 ±0.01 | 0.06 ±0.00 |
| Scrub | 0.27 ±0.01 | 0.31 ±0.00 |
| RewoD | 1.00 ±0.04 | 1.00 ±0.06 |
| RELOAD | 0.24 ±0.01 | 0.20 ±0.00 |
| **Num. Corrupted Samples** | | |
| CIFAR10 | Poison | 1000 |
| CIFAR100 | Poison | 1000 |
| CIFAR10 | IC | 5000 |
| CIFAR100 | IC | 500 |

Table 25: Cost (↓) values across different sizes of $\mathcal{D}_{forget}$ (Corrective Unlearning with replacement). Results are reported as mean$_{\pm\text{stddev}}$ over 10 values of $\gamma$.

| $Acc_{retain}$ (↑) | CIFAR10 | CIFAR100 |
|---|---|---|
| **Poisoning** | | |
| None | 91.35 | 74.05 |
| BadT | -0.02 ±0.06 | -0.13 ±0.15 |
| CF | 0.07 ±0.09 | 0.37 ±0.07 |
| SSD | -16.01 ±23.22 | 0.00 ±0.00 |
| Scrub | -0.01 ±0.01 | -0.01 ±0.06 |
| RewoD | 0.72 ±0.10 | 1.31 ±0.19 |
| RELOAD | -10.03 ±0.42 | -73.04 ±0.01 |
| **Interclass Confusion (IC)** | | |
| None | 93.01 | 73.82 |
| BadT | 0.39 ±0.16 | 0.03 ±0.11 |
| CF | -0.27 ±0.22 | 0.48 ±0.21 |
| SSD | -44.46 ±24.45 | -72.80 ±0.00 |
| Scrub | 0.38 ±0.01 | 0.12 ±0.28 |
| RewoD | 0.66 ±0.10 | 1.30 ±0.10 |
| RELOAD | -5.84 ±3.03 | -30.63 ±16.14 |
| **Num. Corrupted Samples** | | |
| CIFAR10 | Poison | 100 |
| CIFAR100 | Poison | 100 |
| CIFAR10 | IC | 500 |
| CIFAR100 | IC | 100 |

| $Acc_{retain}$ (↑) | CIFAR10 | CIFAR100 |
|---|---|---|
| **Poisoning** | | |
| None | 90.97 | 74.20 |
| BadT | -0.35 ±0.14 | -0.72 ±0.42 |
| CF | 0.34 ±0.11 | 0.22 ±0.11 |
| SSD | -40.73 ±30.47 | 0.00 ±0.00 |
| Scrub | 0.50 ±0.13 | 0.36 ±0.17 |
| RewoD | 1.31 ±0.09 | 0.94 ±0.20 |
| RELOAD | -7.92 ±0.38 | -73.20 ±0.01 |
| **Interclass Confusion (IC)** | | |
| None | 92.22 | 74.06 |
| BadT | 0.83 ±0.12 | -0.23 ±0.04 |
| CF | -0.18 ±0.18 | 0.14 ±0.26 |
| SSD | -21.52 ±22.80 | 0.00 ±0.00 |
| Scrub | 0.99 ±0.25 | -0.05 ±0.25 |
| RewoD | 0.99 ±0.24 | 1.03 ±0.11 |
| RELOAD | -0.01 ±1.12 | -22.25 ±1.63 |
| **Num. Corrupted Samples** | | |
| CIFAR10 | Poison | 500 |
| CIFAR100 | Poison | 500 |
| CIFAR10 | IC | 2500 |
| CIFAR100 | IC | 250 |

| $Acc_{retain}$ (↑) | CIFAR10 | CIFAR100 |
|---|---|---|
| **Poisoning** | | |
| None | 90.84 | 74.34 |
| BadT | -0.19 ±0.30 | -1.37 ±0.65 |
| CF | 0.36 ±0.13 | 0.38 ±0.11 |
| SSD | -0.00 ±0.00 | -73.34 ±0.00 |
| Scrub | 0.41 ±0.12 | 0.61 ±0.16 |
| RewoD | 1.11 ±0.11 | 0.96 ±0.11 |
| RELOAD | -8.49 ±0.29 | -73.33 ±0.15 |
| **Interclass Confusion (IC)** | | |
| None | 92.81 | 73.81 |
| BadT | 0.56 ±0.13 | -0.15 ±0.12 |
| CF | -0.07 ±0.08 | 0.15 ±0.27 |
| SSD | -1.23 ±1.63 | -72.79 ±0.00 |
| Scrub | -17.88 ±20.93 | -0.07 ±0.18 |
| RewoD | 0.31 ±0.50 | 1.03 ±0.16 |
| RELOAD | -3.02 ±4.14 | -26.35 ±6.45 |
| **Num. Corrupted Samples** | | |
| CIFAR10 | Poison | 1000 |
| CIFAR100 | Poison | 1000 |
| CIFAR10 | IC | 5000 |
| CIFAR100 | IC | 500 |

Table 26: Acc$_{retain}$ (↑) values across different sizes of $\mathcal{D}_{forget}$ (Corrective Unlearning with replacement). Results are reported as mean$_{\pm\text{stddev}}$ over 10 values of $\gamma$.

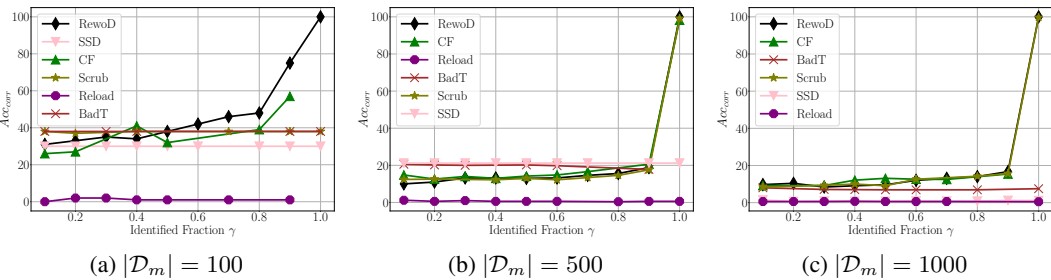

(a) $|\mathcal{D}_m| = 100$  (b) $|\mathcal{D}_m| = 500$  (c) $|\mathcal{D}_m| = 1000$

Figure 16: CIFAR100 Poisoning (with replacement)

## B.7 HYPERPARAMETER SELECTION

RELOAD admits 4 hyperparameters. Additional hyperparameters may be introduced depending on the optimisation procedure used by the practitioner for RELOAD (eg. weight decay).

1. Alpha ($\alpha$): The quantile of weights to reinitialise

2. Ascent Learning Rate: The step size for the ascent stage of RELOAD

3. Finetuning Learning Rate: The step size for the finetuning stage of RELOAD

4. Weight Reset Method: The scheme to use for reinitialising weights

### B.7.1 WEIGHT RESET/REINITIALISATION METHODS

First, we detail the different weight reinitialisation methods we explore as options for the resetting step of RELOAD. This setting is a hyperparameter of RELOAD .

*Mean.* The selected parameters are replaced with the mean value of the tensor they are part of.

*Zero.* The selected parameters are replaced with the value 0.

*Normal.* The selected parameters are replaced with a random number drawn from $\mathcal{N}(0, 1)$.

*Uniform.* The selected parameters are replaced with a random number drawn from $\mathcal{U}(-1, 1)$.

*Xavier Uniform.* The selected parameters are replaced with values obtained through Xavier Uniform initialisation (Glorot & Bengio, 2010).

*Xavier Normal.* The selected parameters are replaced with values obtained through Xavier Normal initialisation (Glorot & Bengio, 2010).

*Kaiming Uniform.* The selected parameters are replaced with values obtained through Kaiming Uniform initialisation (He et al., 2015).

*Kaiming Normal.* The selected parameters are replaced with values obtained through Kaiming Normal initialisation (He et al., 2015).

### B.7.2 HYPERPARAMETERS FOR CLASSICAL UNLEARNING

We train ResNet-18 and VGG16-BN models on CIFAR-10 (Krizhevsky, 2012), CIFAR-100 (Krizhevsky et al.), and SVHN (Netzer et al., 2011) for image classification for 182 epochs. We apply the cross-entropy loss function and a learning rate of 0.1 with a batch size of 256. We conducted these experiments over 10 random seeds to obtain average results and standard deviation measurements. The results in our tables are reported in the format $\mu_{\pm\sigma}$ where $\mu$ is the average value and $\sigma$ is the standard deviation, across the 10 seeds.

The 10 seeds we selected for unlearning experiments were seeds $\{1, 2, 3, 4, 5, 6, 7, 8, 9, 10\}$.

Hyperparameters for RELOAD were chosen through a bayesian hyperparameter sweep. The chosen hyperparameters for the unlearning tasks are presented in Table 27.

We empirically find that the cumulative distribution function of the knowledge-values for forgetting 10% of data from a ResNet-18 model trained on SVHN forms a sigmoid-like curve around $10^{-1}$. This further evidences the existence of clear differences in the knowledge-values for different parameters. Experimentally, we select the thresholding hyperparameter $\alpha$ using a hyperparameter sweep. We have included in ablation (Appendix C.8), a study with varying learning rates ($\eta$) and thresholds ($\alpha$).

**Baseline Implementations.** Implementations for baselines were taken from the reference implementations for SCRUB, SSD, EU-$k$, and CF-$k$. Implementations for FT and GA were taken from the repository for SalUn.

**Codebase Structure.** Our codebase is built on the publicly-released repository for SalUn (Fan et al., 2023).

| Experiment | Alpha ($\alpha$) | Ascent Learning Rate | Finetuning Learning Rate | Weight Reset Method |
|---|---|---|---|---|
| SVHN + ResNet-18 | 0.1 | 0.243 | 0.098 | Uniform |
| SVHN + VGG16-BN | 0.1 | 0.496 | 0.496 | Xavier Uniform |
| CIFAR-10 + ResNet-18 | 0.1 | 0.44 | 0.33 | Xavier Uniform |
| CIFAR-10 + VGG16-BN | 0.1 | 0.167 | 0.39 | Kaiming Uniform |
| CIFAR-100 + ResNet-18 | 0.1 | 0.18 | 0.33 | Xavier Normal |
| CIFAR-100 + VGG16-BN | 0.1 | 0.325 | 0.164 | Kaiming Normal |

Table 27: RELOAD Hyperparameter Settings for Unlearning

### B.7.3 HYPERPARAMETERS FOR LANGUAGE MODEL ENTITY UNLEARNING

For our base models we employ open-source model weights fine-tuned on the TOFU dataset publicly available on HuggingFace for Llama-2-7b-Chat (Touvron et al., 2023) and Phi-1.5 (Li et al., 2023). We use open-source fine-tuned models available on Hugging Face (locuslab, 2025; Unlearning, 2025a;b;c) as our gold-standard retrained models.

Hyperparameters for RELOAD were chosen through a bayesian hyperparameter sweep. The chosen hyperparameters for the unlearning tasks are presented in Table 28. All experiments were conducted using the AdamW optimizer from PyTorch.

As discussed, RELOAD for LMs is parameter-efficient, and operates on a subset of the model parameters in a language model. As such, the entire RELOAD process is performed over a subset of the layers in the language model. We selected this layer as a hyperparameter through a sweep, and report it as `Target Layers` in the below table. This structure enabled effective unlearning, and also increased the efficiency of the algorithm as gradients were only computed for certain layers. This allowed RELOAD to operate on large language models on less powerful hardware setups and in less time.

| Experiment | Alpha ($\alpha$) | Ascent Learning Rate | Finetuning Learning Rate | Weight Reset Method | Retain Sample Size | Target Layers | Repair Epochs |
|---|---|---|---|---|---|---|---|
| Llama-2-7b-Chat + Forget01 | 0.008 | 0.039 | 0.0002 | Zero | 152 | `mlp.gate_proj.weight` | 5 |
| Llama-2-7b-Chat + Forget05 | 0.017 | 0.049 | 0.0002 | Uniform | 193 | `self_attn.k_proj.weight` | 5 |
| Llama-2-7b-Chat + Forget10 | 0.303 | 0.022 | 0.0488 | Xavier Uniform | 60 | `self_attn.q_proj.weight` | 28 |
| Phi-1.5 + Forget10 | 0.396 | 0.094 | 0.0445 | Uniform | 90 | `self_attn.v_proj.weight` | 40 |

Table 28: Hyperparameter Settings for LM Entity Unlearning

**Baseline Implementation and Results.** Baseline implementations and results were obtained and reused from the repository and paper of Liu et al. (2024a).

**Codebase Structure.** For LM Entity Unlearning we reuse the publicly-released repository for Large Language Model Unlearning via Embedding-Corrupted Prompts (Liu et al., 2024a) to which we add our implementation of RELOAD for LMs.

### B.7.4 HYPERPARAMETERS FOR CORRECTIVE UNLEARNING

We train ResNet-9 models on CIFAR-10 (Krizhevsky, 2012) for 4000 pretraining iterations. We train ResNet-28x10 models on CIFAR-100 (Krizhevsky et al.) for 6000 pretraining iterations. We apply the cross-entropy loss function, a batch-size of 512, and a learning rate of 0.025. The results in our tables are reported in a manner consistent with prior work (Goel et al., 2024) across 10 selections of $\gamma$ where $\gamma$ is the proportion of the forget set identified.

Hyperparameters for RELOAD are chosen through a Bayesian Hyperparameter sweep. The chosen hyperparameters are presented in Table 29. Hyperparameters for baselines are chosen through a grid search defined in the reference implementation. For all experiments we employ the SGD (Stochastic Gradient Descent) optimizer with momentum 0.9 and weight decay 0.0005.

**Baseline Implementation.** Baseline implementations are taken from the reference implementations provided in the repository for Corrective Machine Unlearning.

**Codebase Structure.** For corrective unlearning we reuse the publicly-released repository for Corrective Machine Unlearning (Goel et al., 2024) to which we add our implementation of RELOAD. To this repository, we also add our implementation of corrective unlearning (with replacement) experiments.

| Experiment | Alpha ($\alpha$) | Ascent Learning Rate | Finetuning Learning Rate | Weight Reset Method |
|---|---|---|---|---|
| CIFAR10 + ResNet-9 + Poisoning | 0.3984 | 0.01 | 0.00381 | Xavier Normal |
| CIFAR10 + ResNet-9 + Interclass Confusion | 0.1978 | 0.01 | 0.00957 | Mean |
| CIFAR100 + ResNet-28x10 + Poisoning | 0.2401 | 0.01 | 0.00483 | Xavier Normal |
| CIFAR100 + ResNet-28x10 + Interclass Confusion | 0.0924 | 0.01 | 0.00237 | Zero |

Table 29: RELOAD Hyperparameter Settings for corrective unlearning (with and without replacement)

## B.8   HARDWARE USAGE

**Hardware for Classical Unlearning**   All experiments were run on 4 CPU cores, 20 GB of RAM, and 1 NVIDIA T4 GPU.

**Hardware for Language Model Entity Unlearning**   All experiments were run on 30 CPU cores, 60GB of RAM, and 1 NVIDIA A40 GPU. Some experiments were also conducted using 1 NVIDIA RTX6000 GPU to highlight the lightweight nature of RELOAD for LMs.

**Hardware for Corrective Unlearning**   All experiments were run on 4 CPU cores, 60 GB of RAM, and 1 NVIDIA RTX6000 GPU.

## C   FURTHER ABLATIONS

### C.1   ABLATION: CRITICAL COMPONENTS OF RELOAD

In designing the knowledge values for identifying knowledgeable parameters, we considered several other approaches in addition to the final formula (Eq. 1). This includes normalising gradients and utilising cosine similarity for the computation of knowledge values.

**Ascent Steps.** In designing the ascent step, we considered the possibility of needing multiple steps to appropriately scrub the global information from the model parameters. Theoretically, this notion violates the partially-blind nature of the unlearning setup, and was thus undesirable. Empirically, we noted that using multiple ascent steps does not improve forgetting and can lead to further performance degradation on $\mathcal{D}_{retain}$ requiring more retraining to get to a final unlearned model. As it is the only partially-blind variant, we include a study of RELOAD when the ascent step is not applied below.

### C.2   ABLATION: EMPIRICAL EVIDENCE

Below we present a short ablation study on 3 different variations of components of the RELOAD algorithm.

1. ReloadWithoutAscent: This is the same as the standard RELOAD algorithm without the ascent step

2. ReloadWithNormalisation: This variant employs gradient normalisation before the calculation of knowledge values to increase directional information and reduce scaling issues

3. ReloadWithCosineKV: This variant uses cosine similarity between gradients to compute knowledge values instead of gradient magnitudes

We demonstrate these variants against the baselines RELOAD algorithm on corrective unlearning tasks. Select results for corrective unlearning are shown in Figure 17.

In the case of corrective unlearning, we note that the variants of RELOAD perform comparatively to the base algorithm but all note significant weaknesses in comparison. ReloadWithCosineKV produces unlearned models with higher utility (higher $Acc_{retain}$, Table 18) but significantly lower corrective accuracy ($Acc_{corr}$, Fig. 17. On the other hand, ReloadWithoutAscent and ReloadWithNormalisation both exhibit better corrective accuracy performance than ReloadWithCosineKV but are still weaker than RELOAD and have lower $Acc_{retain}$.

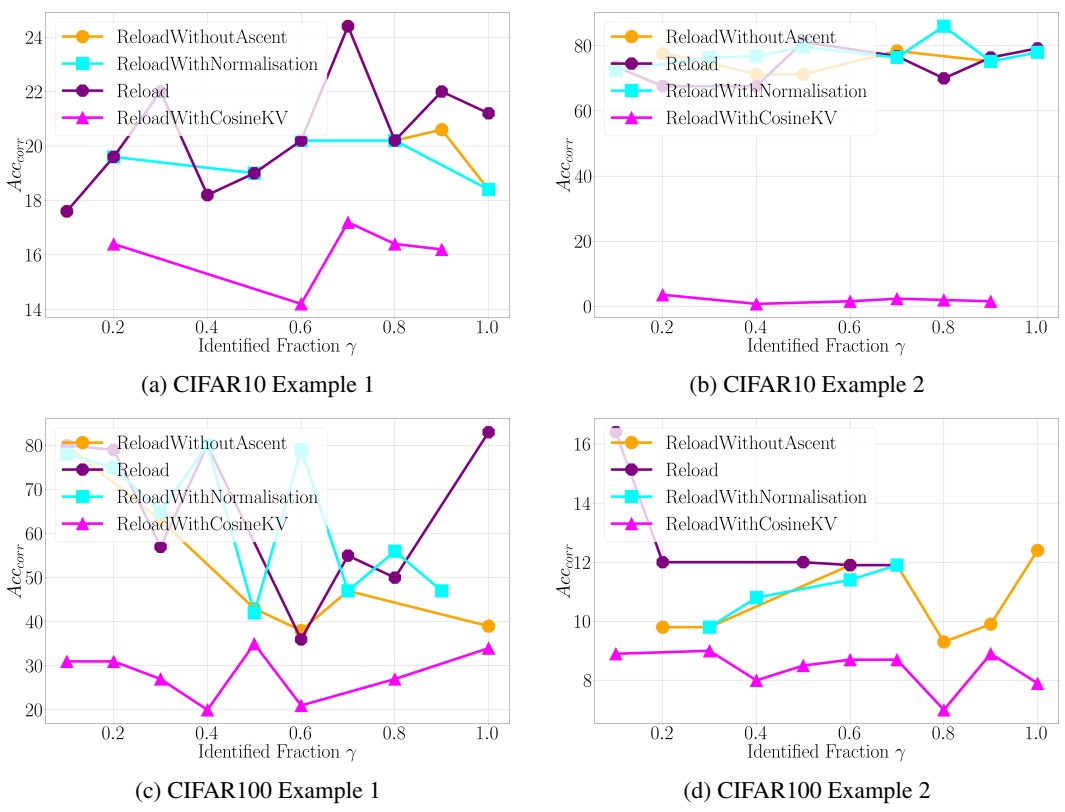

Figure 17: Corrective Accuracy ($Acc_{corr}$) across variants of RELOAD.

| $Acc_{retain}$ (↑) | CIFAR10 | CIFAR100 |
|---|---|---|
| **Poisoning** | | |
| RELOAD | -2.84 $_{\pm 2.56}$ | -5.08 $_{\pm 1.39}$ |
| ReloadWithoutAscent | -3.28 $_{\pm 2.49}$ | -5.24 $_{\pm 1.65}$ |
| ReloadWithNormalisation | -4.23 $_{\pm 2.11}$ | -5.34 $_{\pm 1.72}$ |
| ReloadWithCosineKV | 0.24 $_{\pm 0.12}$ | 0.49 $_{\pm 0.15}$ |
| **Interclass Confusion (IC)** | | |
| RELOAD | -2.70 $_{\pm 1.99}$ | -26.18 $_{\pm 29.97}$ |
| ReloadWithoutAscent | -3.66 $_{\pm 2.13}$ | -16.83 $_{\pm 23.29}$ |
| ReloadWithNormalisation | -4.27 $_{\pm 1.56}$ | -16.67 $_{\pm 21.67}$ |
| ReloadWithCosineKV | 0.01 $_{\pm 0.12}$ | 0.24 $_{\pm 0.15}$ |
| **Num. Corrupted Samples** | | |
| CIFAR10 | Poison | 100 |
| CIFAR100 | Poison | 100 |
| CIFAR10 | IC | 500 |
| CIFAR100 | IC | 100 |

Figure 18: Model Utility across variants of RELOAD

## C.3 ABLATION: RELOAD ON VISION TRANSFORMERS

We study the impact of layer normalization on the performance of RELOAD . We train a Vision Transformer (Dosovitskiy et al., 2020) on the CIFAR-10 dataset (Krizhevsky, 2012) and randomly unlearn 6000 data samples (10% of CIFAR-10).

We reuse a PyTorch implementation of a ViT (Wang et al., 2025) and train the model for 1000 epochs with learning rate 1e-4 using the Adam (Kingma & Ba, 2014) optimizer. The baseline model is trained to an accuracy of 99.94%. Results are presented in Table 30.

| Method | RA ($\uparrow$) | $\Delta$FA ($\downarrow$) | $\Delta$FE ($\downarrow$) | $\Delta$FMIA ($\downarrow$) | RSKL ($\downarrow$) | FSKL ($\downarrow$) |
|---|---|---|---|---|---|---|
| RELOAD (OURS) | $99.45_{\pm 0.12}$ | $0.53_{\pm 0.50}$ | $0.79_{\pm 0.1}$ | $0.01_{\pm 0.01}$ | $0.19_{\pm 0.03}$ | $8.7_{\pm 0.11}$ |
| Retrained (Baseline) | $99.91_{\pm 0.02}$ | $54.11_{\pm 0.50}$ | $4.31_{\pm 0.09}$ | $0.51_{\pm 0.01}$ | - | - |

Table 30: **10% Random Forgetting on CIFAR-10 (ViT)**
$\uparrow$: the goal is to have as high of a value as possible, $\Delta^{\downarrow}$: the value in the table is the difference between the result of the unlearning method and retraining (bottom row) on the metric and the goal is to have a low difference, $\downarrow$: the goal is to have as low of a value as possible. The bottom row presents the absolute value of $M_{(\theta \sim)}$ on each metric. For any metric with $\Delta$, the raw value is instead reported. Rows for $\Delta$FA ($\downarrow$), $\Delta$FE ($\downarrow$), and $\Delta$FMIA ($\downarrow$) present the absolute difference in the value of the corresponding method on this metric to the value of $M_{(\theta \sim)}$ on the metric. These results show that RELOAD performs similarly to the ground truth Retrained model on RA, $\Delta$FA, $\Delta$FMIA, and RSKL. RELOAD strays from the Retrained model in $\Delta$FE and FSKL.

## C.4 ABLATION: RELOAD WITH QUANTIZED GRADIENTS

RELOAD incurs a storage overhead when caching gradients. We explore the feasibility of quantizing the cached gradients, to reduce the footprint of the algorithm. In this experiment, we unlearn 6000 samples (10%) of CIFAR-10 from a trained ResNet-18 model and we quantize the cached gradients from `torch.float32` to `torch.float16`. Before proceeding with unlearning, we expand these gradients back to `torch.float32`. The ResNet18 model is trained for 400 epochs with a learning rate of 1e-3 using the SGD optimizer. The model has a trained accuracy of 99.76%. Results are presented in Table 31.

| Method | RA ($\uparrow$) | $\Delta$FA ($\downarrow$) | $\Delta$FE ($\downarrow$) | $\Delta$FMIA ($\downarrow$) | RSKL ($\downarrow$) | FSKL ($\downarrow$) |
|---|---|---|---|---|---|---|
| RELOAD (UN-QUANTIZED) | $99.99_{\pm 0.01}$ | $0.46_{\pm 0.57}$ | $0.76_{\pm 0.08}$ | $0.01_{\pm 0.01}$ | $0.44_{\pm 0.03}$ | $4.04_{\pm 0.1}$ |
| RELOAD (QUANTIZED) | $99.99_{\pm 0.01}$ | $0.46_{\pm 0.57}$ | $0.76_{\pm 0.08}$ | $0.01_{\pm 0.01}$ | $0.44_{\pm 0.03}$ | $4.04_{\pm 0.1}$ |
| Retrained (Baseline) | $99.52_{\pm 0.16}$ | $36.22_{\pm 0.49}$ | $2.25_{\pm 0.03}$ | $0.51_{\pm 0.01}$ | - | - |

Table 31: **10% Random Forgetting on CIFAR-10 (ResNet-18) with Quantized Cached Gradients**
$\uparrow$: the goal is to have as high of a value as possible, $\Delta^{\downarrow}$: the value in the table is the difference between the result of the unlearning method and retraining (bottom row) on the metric and the goal is to have a low difference, $\downarrow$: the goal is to have as low of a value as possible. The bottom row presents the absolute value of $M_{(\theta \sim)}$ on each metric. For any metric with $\Delta$, the raw value is instead reported. Rows for $\Delta$FA ($\downarrow$), $\Delta$FE ($\downarrow$), and $\Delta$FMIA ($\downarrow$) present the absolute difference in the value of the corresponding method on this metric to the value of $M_{(\theta \sim)}$ on the metric. These results show that RELOAD performs similarly to the ground truth Retrained model on RA, $\Delta$FA, $\Delta$FE $\Delta$FMIA, RSKL. RELOAD strays from the Retrained model in FSKL.

As observed, performance deterioration across key unlearning and utility metrics is minimal. Gradient quantization is thus a highly effective and low-cost solution for mitigating storage overhead of RELOAD in models where full gradients are required.

## C.5 ABLATION: RELOAD WITH FINETUNING ON OUT-OF-DISTRIBUTION DATA

In this section we explore the practicality of applying RELOAD after some finetuning has been performed on the model. Theoretically, significant parameter changes during finetuning could reduce the fidelity of the cached gradients relative to the current model state. As such, to operate in this setting, the cached gradients must be recomputed over the training and finetuning dataset after finetuning is completed. To explore this, we first train a ResNet18 model on CIFAR-10 for 400 epochs with a learning rate of 1e-3 to a training accuracy of 99.76%. Afterwards, we finetune the model on the out-of-distribution CIFAR-10.1 (Torralba et al., 2008; Recht et al., 2018) dataset for 100 epochs with a learning rate of 1e-3. Prior to finetuning, the model had an accuracy of 30.8% on CIFAR10.1. After finetuning, the model achieved an accuracy of 99.2% on CIFAR10.1.

We evaluate RELOAD unlearning in 3 modes. 1) We unlearn samples from the original CIFAR10 dataset after finetuning on CIFAR10.1 (Table 32, 2) We unlearn samples from CIFAR10.1 after

finetuning on CIFAR10.1 (Table 33), 3) We unlearn samples from both CIFAR10 and CIFAR10.1 after finetuning on CIFAR10.1 (Table 34).

| Method | RA ($\uparrow$) | $\Delta$FA ($\downarrow$) | $\Delta$FE ($\downarrow$) | $\Delta$FMIA ($\downarrow$) | RSKL ($\downarrow$) | FSKL ($\downarrow$) |
|---|---|---|---|---|---|---|
| RELOAD (OURS) | $98.16_{\pm 0.01}$ | $0.57_{\pm 0.46}$ | $0.06_{\pm 0.03}$ | $0.01_{\pm 0.01}$ | $0.91_{\pm 0.02}$ | $3.45_{\pm 0.06}$ |
| Retrained (Baseline) | $92.34_{\pm 0.6}$ | $34.57_{\pm 0.00}$ | $2.33_{\pm 0.00}$ | $0.51_{\pm 0.01}$ | - | - |

Table 32: **10% Random Forgetting from CIFAR-10 after finetuning on CIFAR-10.1 (ResNet-18)**

| Method | RA ($\uparrow$) | $\Delta$FA ($\downarrow$) | $\Delta$FE ($\downarrow$) | $\Delta$FMIA ($\downarrow$) | RSKL ($\downarrow$) | FSKL ($\downarrow$) |
|---|---|---|---|---|---|---|
| RELOAD (OURS) | $98.89_{\pm 0.19}$ | $2.85_{\pm 2.05}$ | $0.20_{\pm 0.13}$ | $0.05_{\pm 0.04}$ | $0.51_{\pm 0.02}$ | $3.95_{\pm 0.27}$ |
| Retrained (Baseline) | $99.73_{\pm 0.11}$ | $32.4_{\pm 2.78}$ | $2.70_{\pm 0.11}$ | $0.54_{\pm 0.05}$ | - | - |

Table 33: **10% Random Forgetting from CIFAR-10.1 after finetuning on CIFAR-10.1 (ResNet-18)**

| Method | RA ($\uparrow$) | $\Delta$FA ($\downarrow$) | $\Delta$FE ($\downarrow$) | $\Delta$FMIA ($\downarrow$) | RSKL ($\downarrow$) | FSKL ($\downarrow$) |
|---|---|---|---|---|---|---|
| RELOAD (OURS) | $99.69_{\pm 0.13}$ | $3.88_{\pm 1.52}$ | $0.27_{\pm 0.11}$ | $0.02_{\pm 0.01}$ | $1.04_{\pm 0.13}$ | $3.97_{\pm 0.15}$ |
| Retrained (Baseline) | $89.39_{\pm 2.89}$ | $35.83_{\pm 0.93}$ | $2.34_{\pm 0.05}$ | $0.52_{\pm 0.02}$ | - | - |

Table 34: **10% Random Forgetting from CIFAR-10 and CIFAR-10.1 after finetuning on CIFAR-10.1 (ResNet-18)**

We observe that across 10 seeds, RELOAD exhibits stronger performance across key metrics with respect to the retrain baseline on all three experiments. This suggests that the continual caching of gradients post finetuning proves RELOAD effective in scenarios where the base model is fine-tuned on distributionally shifted data.

An inherent requirement for this robustness is that gradients must be re-computed across the cumulative dataset (original + finetuning data) after significant model updates. While this necessitates continual storage of the training set during the model's active lifecycle, we argue this aligns with standard industry operations wherein data holders typically preserve their full datasets for versioning, debugging, and compliance until a specific deletion request is received.

### C.6 ABLATION: FINETUNING RELOAD WITH SUBSETS OF $\mathcal{D}_{retain}$

By design, Steps (1-4) of RELOAD necessitate the use of all of $\mathcal{D}_{retain}$. Finetuning, Step (5), in literature does not always require the entire dataset. Here we ablate the usage of a subset of $\mathcal{D}_{retain}$ during the finetuning step of RELOAD . We randomly select $\{10, 20, \ldots, 90, 100\}\%$ of $\mathcal{D}_{retain}$ to use for Step (5) of RELOAD . We observe that the performance of RELOAD on RA, $\Delta$FA, and RSKL is directly correlated with the portion of $\mathcal{D}_{retain}$ used for finetuning. Interestingly, lower $\Delta$FE and FSKL scores are produced when finetuning with smaller portions of $\mathcal{D}_{retain}$, despite a larger $\Delta$FA. These results are presented in Table 35.

| Method | RA ($\uparrow$) | $\Delta$FA ($\downarrow$) | $\Delta$FE ($\downarrow$) | $\Delta$FMIA ($\downarrow$) | RSKL ($\downarrow$) | FSKL ($\downarrow$) |
|---|---|---|---|---|---|---|
| RELOAD (10% OF $\mathcal{D}_{retain}$) | $32.33_{\pm 0.36}$ | $11.44_{\pm 0.52}$ | $0.19_{\pm 0.04}$ | $0.01_{\pm 0.01}$ | $4.32_{\pm 0.07}$ | $3.60_{\pm 0.1}$ |
| RELOAD (20% OF $\mathcal{D}_{retain}$) | $42.33_{\pm 0.27}$ | $8.16_{\pm 0.53}$ | $0.26_{\pm 0.04}$ | $0.01_{\pm 0.01}$ | $3.82_{\pm 0.06}$ | $3.62_{\pm 0.1}$ |
| RELOAD (30% OF $\mathcal{D}_{retain}$) | $50.37_{\pm 0.30}$ | $6.92_{\pm 0.79}$ | $0.25_{\pm 0.04}$ | $0.01_{\pm 0.01}$ | $3.35_{\pm 0.06}$ | $3.60_{\pm 0.1}$ |
| RELOAD (40% OF $\mathcal{D}_{retain}$) | $58.30_{\pm 0.32}$ | $5.53_{\pm 0.79}$ | $0.34_{\pm 0.05}$ | $0.01_{\pm 0.01}$ | $2.96_{\pm 0.06}$ | $3.75_{\pm 0.1}$ |
| RELOAD (50% OF $\mathcal{D}_{retain}$) | $65.78_{\pm 0.26}$ | $4.68_{\pm 1.04}$ | $0.43_{\pm 0.05}$ | $0.01_{\pm 0.01}$ | $2.57_{\pm 0.05}$ | $3.88_{\pm 0.1}$ |
| RELOAD (60% OF $\mathcal{D}_{retain}$) | $72.86_{\pm 0.22}$ | $4.00_{\pm 0.78}$ | $0.42_{\pm 0.05}$ | $0.01_{\pm 0.01}$ | $2.12_{\pm 0.03}$ | $3.89_{\pm 0.1}$ |
| RELOAD (70% OF $\mathcal{D}_{retain}$) | $79.91_{\pm 0.17}$ | $3.16_{\pm 0.71}$ | $0.49_{\pm 0.05}$ | $0.01_{\pm 0.01}$ | $1.72_{\pm 0.02}$ | $4.01_{\pm 0.1}$ |
| RELOAD (80% OF $\mathcal{D}_{retain}$) | $86.76_{\pm 0.14}$ | $2.61_{\pm 0.68}$ | $0.54_{\pm 0.05}$ | $0.01_{\pm 0.01}$ | $1.31_{\pm 0.02}$ | $4.13_{\pm 0.1}$ |
| RELOAD (90% OF $\mathcal{D}_{retain}$) | $93.47_{\pm 0.10}$ | $2.27_{\pm 0.75}$ | $0.55_{\pm 0.05}$ | $0.01_{\pm 0.01}$ | $0.87_{\pm 0.01}$ | $4.15_{\pm 0.1}$ |
| RELOAD (100% OF $\mathcal{D}_{retain}$) | $99.99_{\pm 0.01}$ | $1.62_{\pm 0.86}$ | $0.51_{\pm 0.05}$ | $0.01_{\pm 0.01}$ | $0.46_{\pm 0.02}$ | $4.25_{\pm 0.1}$ |
| Retrained (Baseline) | $99.6_{\pm 0.07}$ | $36.57_{\pm 0.56}$ | $2.24_{\pm 0.03}$ | $0.51_{\pm 0.01}$ | - | - |

Table 35: 10% Random Forgetting on CIFAR-10 (ResNet-18). Finetuning with different sized subsets of $\mathcal{D}_{retain}$.

## C.7 ABLATION: RELOAD OPERATES IN HUMAN SENSITIVE AND TABULAR CONTEXTS

Human-sensitive data domains are a primary use case for unlearning in the real-world. In previous work (Newatia et al., 2024), we demonstrate that RELOAD successfully unlearns random samples and entire features from a TabNet model (Arik & Pfister, 2021) trained on the Adult Census Income dataset (Kohavi, 1996; Asuncion et al., 2007), a standard benchmark involving sensitive personal attributes. Full results and discussions are presented in Newatia et al. (2024).

| Method | Forgetting a Random 30% of the Data | | | | | Forgetting a Random Feature | | | |
|---|---|---|---|---|---|---|---|---|---|
| | RA (↑) | ΔFA (↓) | ΔFE (↓) | ΔFMIA (↓) | Cost (↓) | OA (↑) | RA (↑) | TA (↑) | Cost (↓) |
| Original | N/A | N/A | N/A | N/A | N/A | $0.84_{\pm 0.00}$ | $0.73_{\pm 0.07}$ | $0.82_{\pm 0.01}$ | N/A |
| Retrain | $0.84_{\pm 0.01}$ | $0.82_{\pm 0.01}$ | $0.48_{\pm 0.00}$ | $0.51_{\pm 0.13}$ | $1.00_{\pm 0.00}$ | $0.77_{\pm 0.09}$ | $0.84_{\pm 0.01}$ | $0.75_{\pm 0.09}$ | $1.00_{\pm 0.00}$ |
| RELOAD | $\mathbf{0.84_{\pm 0.01}}$ | $0.02_{\pm 0.01}$ | $\mathbf{0.00_{\pm 0.00}}$ | $\mathbf{0.00_{\pm 0.00}}$ | $0.29_{\pm 0.18}$ | $\mathbf{0.78_{\pm 0.03}}$ | $0.81_{\pm 0.01}$ | $\mathbf{0.77_{\pm 0.03}}$ | $0.40_{\pm 0.27}$ |
| GA | $0.38_{\pm 0.22}$ | $0.54_{\pm 0.04}$ | $0.55_{\pm 0.04}$ | $0.06_{\pm 0.07}$ | $\mathbf{0.13_{\pm 0.02}}$ | $0.35_{\pm 0.16}$ | $0.48_{\pm 0.17}$ | $0.35_{\pm 0.16}$ | $\mathbf{0.30_{\pm 0.12}}$ |
| FT | $0.83_{\pm 0.01}$ | $\mathbf{0.01_{\pm 0.01}}$ | $0.01_{\pm 0.00}$ | $0.05_{\pm 0.06}$ | $0.53_{\pm 0.07}$ | $0.77_{\pm 0.09}$ | $\mathbf{0.84_{\pm 0.00}}$ | $0.76_{\pm 0.09}$ | $0.79_{\pm 0.20}$ |

Table 36: Results highlighting the performance of RELOAD on the tabular Census Income dataset.

## C.8 ABLATION: LEARNING RATE $\eta_p$ AND THRESHOLD $\alpha$

We study the effect of different learning rates on the unlearning performance exhibited by the RELOAD algorithm. For this study, we select the case of randomly forgetting 10% of the training data from a ResNet-18 model trained on CIFAR-100.

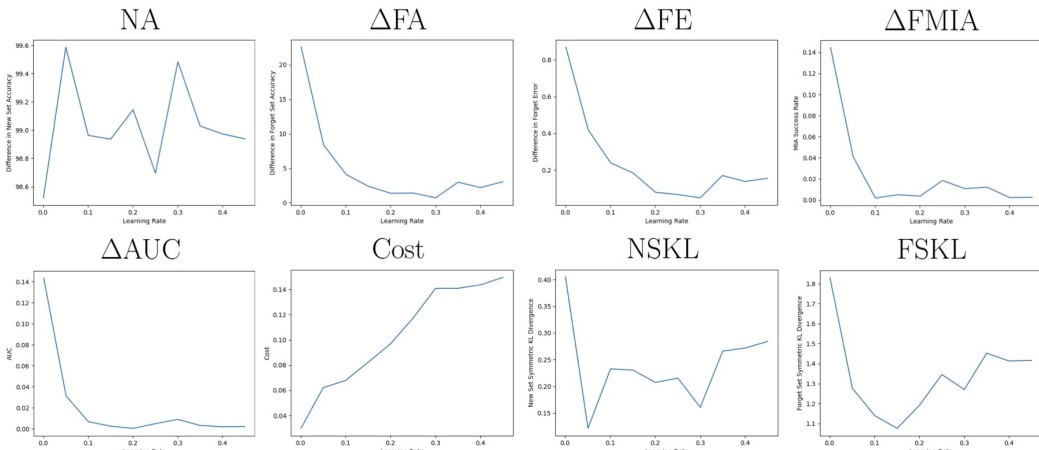

Figure 19: Impact of Learning Rate ($\eta$) on RELOAD performance.

As shown in Figure 19, we observe that the choice of learning rate has a significant impact on performance. This is particularly true in the case of ΔFA, ΔFE, ΔFMIA, and ΔAUC measurements - which are the primary metrics evaluating how well the model has forgotten $\mathcal{D}_{forget}$. Based on these plots, we choose $\eta = 0.33$.

Figure 20 shows the effect of varying the proportion of the parameters that are selected for reinitialization ($\alpha$). We observe that the choice of threshold has an impact on the performance of the RELOAD algorithm and that its selection involves a tradeoff between the different metrics we consider. Thus, the best choice of $\alpha$ should ideally be selected through a hyperparameter search.

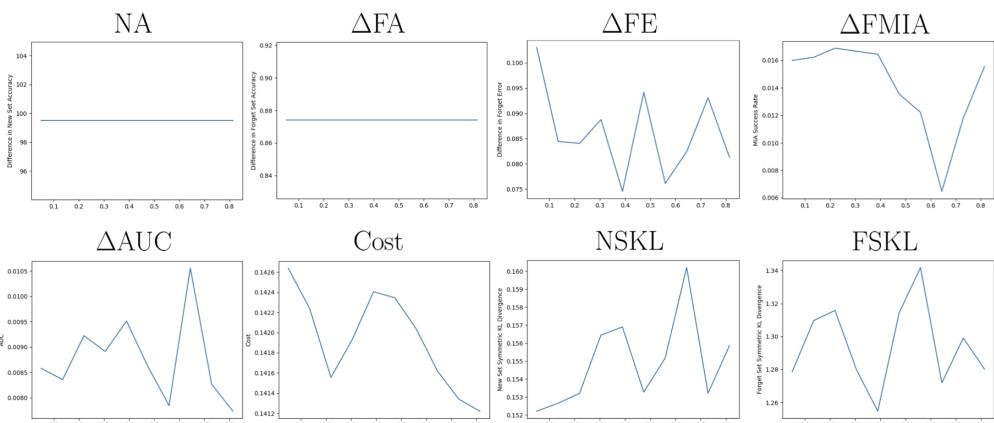

Figure 20: Impact of Threshold ($\alpha$) on RELOAD performance

