# OpenReview forum: "Mitigating Privacy Risk via Forget Set-Free Unlearning"
_ICLR.cc/2026/Conference — ICLR 2026 Poster_

### Official Review · Reviewer_dom5 · 2025-10-25

**Soundness:** 3
**Presentation:** 3
**Contribution:** 2
**Rating:** 4
**Confidence:** 4

**Summary:**

Existing study requires the model provider to keep the requested unlearned data information until the unlearning is completed. This paper explores and defines a new unlearning setting, Partially Blind Unlearning (PBU), in which no direct access to the unlearning data is required. Under this setting, the author implements a three-fold method that leverages cached gradients from the training stage. The method RELOAD consists of three steps, combining previous studies. RELOAD gives the unlearned model under PBU by following 1. Compute gradient difference, perform a single gradient ascent step, and then fine-tune on retained datasets. The author then provides comprehensive experiments on both classical unlearning tasks unlearning on vision models, and unlearning tasks on language models.

**Strengths:**

1. The paper provides a new privacy-oriented perspective on the problem of unlearning. The author made an important observation and defined a new setting for exploring safe unlearning. The motivation is well-explained.
2. The paper integrates several existing methods in solving new and practically important problems. This provides a more modular and interpretable unlearning algorithm.
3. The evaluation is comprehensive and representative. The experiments marked the performance of RELOAD on both small-scale vision models and on language models.

**Weaknesses:**

1. In Section 2.3, the author states that we can infer the loss gradient on the forget dataset with the original model by computing the difference between the loss gradient on the original dataset and the retained dataset. This, however, works with several assumptions that are not explicitly stated, such as the assumption that loss is additively computed across samples. I am concerned that for contrastive learning tasks or models trained with layer normalization, this formula may not function as intended. I would appreciate further discussion on this point.
2. While the overall unlearning method provides more interpretability compared to existing unlearning methods and achieves promising results, the methodological innovation is somewhat incremental. The core part of the method, including gradient ascent, selective weight re-initialization, and fine-tuning are existing method. The contribution of the method lies mostly in how to combine the methods under PBU setting.

**Questions:**

It seems like RELOAD is achieving good results in the baseline section. The setting for the experiments is not strict to PBU. This indicates that  RELOAD is producing better results and outperforms baseline methods that do use the forget set. If so, why is PBU important to this method?

---

> ### Author Response · Authors · 2025-11-28
>
> We thank Reviewer dom5 for their insightful feedback. We are encouraged that the reviewer finds our motivation “well-explained”, that our method addresses “new and practically important problems”. We value the assessment that our work provides a "modular and interpretable" solution.
> Below, we hope to address all of the reviewer’s concerns.
> # Weaknesses
> > In Section 2.3, the author states that we can infer the loss gradient on the forget dataset with the original model by computing the difference between the loss gradient on the original dataset and the retained dataset. This, however, works with several assumptions that are not explicitly stated, such as the assumption that loss is additively computed across samples. I am concerned that for contrastive learning tasks or models trained with layer normalization, this formula may not function as intended. I would appreciate further discussion on this point.
>
> We appreciate this insightful observation. Indeed, our recovering of the loss gradient on the forget dataset assumes the loss is computed additively across samples, which is the standard case for the supervised losses primarily targeted in this work. We will make sure to include this assumption in Section 2.1. As it is a core assumption to the functioning of RELOAD.
> Since layer normalization operates on a per-sample basis, it does not introduce cross-sample dependencies within a batch. Therefore, the additive property of the gradients holds, and our subtraction method remains mathematically valid for architectures like Transformers.
>
> To empirically validate this, we conducted additional experiments on Vision Transformers (ViT-B/16), which employ Layer normalization. As shown in the newly added table below, RELOAD performs exceptionally well on a ViT, reaching optimal unlearning statistics. These results confirm that for LayerNorm-based architectures, the gradient derivation functions as intended. The full experimental details are in the revised manuscript in Section 3.5.
>
> ### Table: 10% Random Forgetting on CIFAR-10 (ViT)
>
> | Method                | RA (↑)            | ΔFA (↓)             | ΔFE (↓)            | ΔFMIA (↓)         | RSKL (↓)         | FSKL (↓)         |
> |-----------------------|-------------------|----------------------|---------------------|--------------------|-------------------|-------------------|
> | **Reload (ResNet-18)** | 99.49± 0.10 | 1.83± 0.83 | 0.05± 0.04 | 0.00± 0.00 | 0.12± 0.01 | 0.53± 0.07 |
> | **Reload (ViT)**     | 99.45± 0.12    | 0.53± 0.50        | 0.79± 0.1        | 0.01± 0.01      | 0.19± 0.03     | 8.7± 0.11      |
> | **Retrained (Baseline, ResNet-18)** | 99.99± 0.01 | 94.40± 0.72       | 0.23± 0.08      | 0.50± 0.01      | -                 | -                 |
> | **Retrained (Baseline, ViT)** | 99.91± 0.02 | 54.11± 0.50       | 4.31± 0.09       | 0.51± 0.01      | -                 | -                 |
>
> We acknowledge that more caution is required for models using Batch Normalization (e.g., ResNet-18). Since removing the forget set alters the batch statistics (mean/variance) for the retained set, the gradients are technically coupled. However, in practice, we observe that this "approximation mismatch" is negligible. As detailed in our main results, RELOAD achieves competitive performance on ResNet-18, suggesting the method is robust enough to handle the minor gradient drift introduced by batch-dependent statistics.
>
> Finally, we greatly appreciate the insight on contrastively trained models. We believe that investigating the effect of a RELOAD-style unlearning on latent representations of a contrastively-trained model is an exciting future direction. Crucially, the focus in such a setting shifts from measuring traditional performance gain or drop with respect to the retrain baseline, to assessing the identifiability and similarity of the learned representations between the unlearned model and the retrain baseline. For instance, future work could focus on metrics that quantify how closely the representation space of the unlearned model aligns with that of the retrained model, such as centered kernel alignment (CKA).

---

> ### Author Response · Authors · 2025-11-28
>
> > While the overall unlearning method provides more interpretability compared to existing unlearning methods and achieves promising results, the methodological innovation is somewhat incremental. The core part of the method, including gradient ascent, selective weight re-initialization, and fine-tuning are existing method. The contribution of the method lies mostly in how to combine the methods under PBU setting.
>
> While we acknowledge that RELOAD leverages established concepts (like gradient ascent), we believe the novelty lies in the synergy and the specific formulations we introduce to make PBU possible.
>
> - The "RAINBOW" of Unlearning: Much like the "Rainbow" agent in Reinforcement Learning demonstrated that integrating distinct improvements yields a state-of-the-art system, RELOAD integrates these components to solve a previously intractable constraint (PBU).
>
> - Selective Re-initialization: While re-initialization exists in literature, our approach is not generic. We introduce a specific, selective re-initialization strategy guided by the Knowledge Values. We formulate and justify the design of these Knowledge Values to particularly target what is integral to navigating the loss landscape in a manner suitable for unlearning. This targeted approach allows us to selectively remove information from weights most important for predicting.
>
> The primary innovation is proving that these components can be orchestrated to unlearn without the forget set, a capability previous methods lacked.
>
> # Questions
> > It seems like RELOAD is achieving good results in the baseline section. The setting for the experiments is not strict to PBU. This indicates that RELOAD is producing better results and outperforms baseline methods that do use the forget set. If so, why is PBU important to this method?
>
> Thank you for bringing up this crucial point. PBU was the specific problem setting we set out to solve: "Can we unlearn data that we are no longer allowed to access?"
>
> The fact that RELOAD outperforms baselines that do have access to the forget set is a significant finding that validates the PBU premise. It demonstrates that access to the specific data points to be forgotten is not actually required for effective unlearning.
>
> PBU is important because it represents the strictest (and safest) privacy standard. In real-world scenarios (e.g., GDPR "Right to be Forgotten"), a provider is often required to delete the data immediately. PBU allows the model to unlearn that data even after the raw records are deleted. We strongly believe that since RELOAD achieves this and beats forget-set aware baselines, future unlearning frameworks should prioritize privacy-preserving (partially-blind) approaches rather than retaining data for the sake of unlearning algorithms.
>
> # Final remarks
> Thank you once again for your helpful feedback. We hope that our rebuttal addresses your questions and concerns, and we kindly ask that you consider a fresher evaluation of our paper if you are satisfied with our responses. We are also more than happy to answer any further questions that arise.

---

### Official Review · Reviewer_2ZVC · 2025-10-29

**Soundness:** 3
**Presentation:** 3
**Contribution:** 3
**Rating:** 6
**Confidence:** 2

**Summary:**

This paper tackles the problem of machine unlearning without retaining the forget set, a long-standing practical limitation in enforcing the “right to be forgotten.” The authors introduce RELOAD, a partially-blind unlearning algorithm that relies only on cached gradients from the final training step and the retained dataset, avoiding the need to store sensitive data. Through gradient ascent, selective reinitialization, and fine-tuning, RELOAD achieves performance comparable to retraining from scratch while being significantly more efficient. Experiments on image classification and large language models demonstrate promising results.

**Strengths:**

1. The paper formalises the partially-blind unlearning setting, which is a realistic and privacy-preserving variant of traditional unlearning.
This addresses an important gap between regulatory demands (e.g., GDPR) and existing technical capabilities.
2. The motivation connecting dataset risk and model risk is conceptually strong and provides a clear societal justification for this work.
3. RELOAD elegantly combines gradient-based unlearning, structured sparsity, and fine-tuning in a simple yet effective pipeline.
The “knowledge value” mechanism for selective reinitialization is particularly interesting.
4. Across diverse tasks (CIFAR, SVHN, and Llama-2), RELOAD shows comparable or better performance than methods requiring the actual forget set. Its efficiency (<8 min for Llama2-7B) suggests real practical potential.

**Weaknesses:**

1. Limited theoretical justification. The paper mainly relies on intuition and empirical validation. While the derivation of ∇L(Dforget) = ∇L(D) − ∇L(Dretain) is sound, the guarantees of approximate unlearning (e.g., bounds on residual influence) are not formally analysed.
2. Dependence on cached gradients. Storing full-model gradients at the end of training may be expensive for large-scale models, potentially offsetting some of the claimed efficiency or privacy advantages.
3. While results are impressive, the experiments could be broadened — e.g., include more realistic privacy benchmarks or human-sensitive data domains.
4. Some ablation studies (e.g., varying α in reinitialization, or using partial gradient caching) are deferred to the appendix but would strengthen the main text.

**Questions:**

See weaknesses.

---

> ### Author Response · Authors · 2025-11-28
>
> We thank the reviewer for their thorough evaluation of our work. We feel encouraged that the reviewer feels the paper “addresses an important gap between regulatory demands (e.g., GDPR) and existing technical capabilities”, that our motivation connecting dataset risk and model risk is “conceptually strong”, and “provides a clear societal justification for the work”. We appreciate that the reviewer highlights our “simple yet effective pipeline”, and that our algorithm’s efficiency on Llama2-7B suggests “real practical potential”.
> In the following, we hope to address your specific concerns and questions below.
> # Weaknesses
> > Limited theoretical justification. The paper mainly relies on intuition and empirical validation. While the derivation of ∇L(Dforget) = ∇L(D) − ∇L(Dretain) is sound, the guarantees of approximate unlearning (e.g., bounds on residual influence) are not formally analysed.
>
> We acknowledge that we do not provide closed-form theoretical bounds on the residual influence or differential privacy guarantees in this work, and defer formal theoretical analysis to future work.
>
> We do argue that in the absence of tight bounds, rigorous empirical auditing becomes the standard for validity. To that end, we rely on differences in Membership Inference Attacks (ΔMIA) between reload and the retrain baseline as a practical proxy for privacy guarantees.
>
> A ΔMIA close to zero indicates that the privacy risk of our model is equivalent to that of a model that never saw the data. Our results consistently demonstrate a negligible ΔMIA, empirically validating that RELOAD successfully approximates the retrained baseline, thereby removing the influence of the forget set as effectively as retraining from scratch.
>
> > Dependence on cached gradients. Storing full-model gradients at the end of training may be expensive for large-scale models, potentially offsetting some of the claimed efficiency or privacy advantages.
>
> This is an important practical consideration, thank you for raising this insightful comment! We have addressed this in the revision with a quantization ablation:
>
> We performed experiments storing the cached gradients in FP16 quantization rather than FP32. Specifically, we experimented with storing the cached gradients ∇θ​L(D) in float16 instead of float32, and then inflating back to float32 at RELOAD runtime, halving the storage cost of stored gradients.
>
> As shown below, performance deterioration across key unlearning and utility metrics is minimal and generally within the standard deviation of the full-precision results. The full experimental details are in the revised manuscript in Section 3.5.
> ### Table: 10% Random Forgetting on CIFAR-10 (ResNet-18) with Quantized Cached Gradients
> | Method                   | RA (↑)            | ΔFA (↓)             | ΔFE (↓)            | ΔFMIA (↓)         | RSKL (↓)         | FSKL (↓)         |
> |--------------------------|-------------------|----------------------|---------------------|--------------------|-------------------|-------------------|
> | **Reload (Un-Quantized)** | 99.99± 0.01    | 0.46± 0.57        | 0.76± 0.08       | 0.01± 0.01      | 0.44± 0.03     | 4.04± 0.1      |
> | **Reload (Quantized)**   | 99.99± 0.01    | 0.46± 0.57        | 0.76± 0.08       | 0.01± 0.01      | 0.44± 0.03     | 4.04± 0.1      |
> | **Retrained (Baseline)** | 99.52± 0.16    | 36.22± 0.49       | 2.25± 0.03       | 0.51± 0.01      | -                 | -                 |
>
>
> These results indicate that gradient quantization is a highly effective and low-cost solution for mitigating the storage overhead of RELOAD in models where full gradients are required, significantly lowering  the hardware barrier for deployment.
> For the entity unlearning experiments designed for language models such as Llama2-7B, the setup does not require caching gradients. Instead, we borrow the core mechanistic insights of RELOAD to design an efficient, gradient-free prompting solution that effectively unlearns targeted entities without the need for additional storage overhead.

---

> ### Author Response · Authors · 2025-11-28
>
> > While results are impressive, the experiments could be broadened — e.g., include more realistic privacy benchmarks or human-sensitive data domains.
>
> We agree with your suggestion to broaden the experimental scope to include human-sensitive domains. We would like to emphasize that our evaluation framework is strictly designed around privacy-sensitive metrics, specifically the Forget Membership Inference Attack ($\Delta$FMIA). Unlike utility metrics (like accuracy), $\Delta$FMIA directly quantifies the reduction in "model risk" by measuring whether an attacker can distinguish if specific data points were part of the training set.
>
> To demonstrate RELOAD's capacity to minimize this specific privacy risk in human-centric domains, we extended our evaluation to the Adult Census Income dataset, a standard benchmark involving sensitive personal attributes.The results confirm that RELOAD excels at minimizing privacy leakage even on human data.
>
> As detailed in the table below, RELOAD achieves a $\Delta$FMIA of 0.00, indicating that the unlearned model is indistinguishable from a model retrained from scratch regarding membership privacy risk. This validates that our method effectively neutralizes the "dataset risk" 5 associated with sensitive human entities, while retaining high model utility.
>
> ### Table: Results on Census Income — Randomly forgetting 30% of the data
>
> | Method   | RA (↑)            | ΔFA (↓)           | ΔFE (↓)           | ΔFMIA (↓)         | Cost (↓)          |
> |----------|--------------------|--------------------|--------------------|--------------------|--------------------|
> | **Retrained (Baseline)**  | 0.84± 0.01      | 0.82± 0.01      | 0.48± 0.00      | 0.51± 0.13      | 1.00± 0.00      |
> | **Reload**   | 0.84± 0.01      | 0.02± 0.01      | 0.00± 0.00      | 0.00± 0.00      | 0.29± 0.18      |
> | **GA**       | 0.38± 0.22      | 0.54± 0.04      | 0.55± 0.04      | 0.06± 0.07      | 0.13± 0.02      |
> | **FT**       | 0.83± 0.01      | 0.01± 0.01      | 0.01± 0.00      | 0.05± 0.06      | 0.53± 0.07      |
> > Some ablation studies (e.g., varying α in reinitialization, or using partial gradient caching) are deferred to the appendix but would strengthen the main text.
>
> We agree that these parameters are central to understanding the robustness of RELOAD. We have revised the manuscript to move the ablation study on VisionTransformers and with Quantized Gradients from the Appendix to the Main Results section (Section 3.5), leveraging the increased page limit. This provides a more complete picture of the algorithm's stability across model types and gradient caching techniques within the primary page limit.
> # Final remarks
>
> We thank the reviewer once again for their constructive feedback. We hope that our rebuttal addresses your questions and concerns, and we kindly ask that you consider a fresher evaluation of our paper if you are satisfied with our responses. We are also more than happy to answer any further questions that arise.

---

### Official Review · Reviewer_WbA5 · 2025-10-30

**Soundness:** 3
**Presentation:** 3
**Contribution:** 3
**Rating:** 6
**Confidence:** 3

**Summary:**

The paper proposes RELOAD, a partially-blind unlearning (PBU) method that aims to remove the influence of a forget set without access to the forget data. The method includes three parts: 1) an ascent step using cached final-training gradients, 2) re-initialisation of parameters with low “knowledge value,” and 3) fine-tuning on the retain set to maintain model utility. Experiments include classic unlearning, entity unlearning, and corrective unlearning.

**Strengths:**

1. The problem of forget-set-free unlearning is novel and well-motivated. The RELOAD enables machine unlearning without retaining the raw forget set by using cached final-step gradients.

2. The empirical results are strong across benchmarks. RELOAD can preserve model utility while improving forget quality.

3. The method is efficient. For Llama-2-7B, it uses 7% of weights and <0.025% of retained data, finishing in 8 minutes on a single GPU. This result suggests that the method is practical.

**Weaknesses:**

1.	The paper does not provide a theoretical justification for why a single gradient-ascent update using $\nabla_\theta L(D)-\nabla_\theta L(D_{\text{retain}})$ and selective re-initialization of low-KV weights can remove the influence of the forget data. As a result, it is unclear when RELOAD succeeds or fails beyond the reported scenarios.

2.	The paper claims RELOAD “allow user data to be immediately removed when a request for deletion is made, eliminating the continued accumulation of dataset risk,” but RELOAD requires the retain set to compute gradients and to perform fine-tuning. This means that the retained data is still at risk, and the ideal method should not use any training data during the unlearning process.

3.	The robustness of RELOAD across different numbers of requests is unclear. When the forget set is extremely small (e.g., unlearning only one data point), $\nabla_\theta L(D_{\text{forget}})=\nabla_\theta L(D)-\nabla_\theta L(D_{\text{retain}})$ becomes a very small residual between two nearly identical large gradients; this makes the ascent step near zero and may fail to truly forget that sample. In addition, when the forget request is large (e.g., 30% of the data), the ascent and re-initialization would degrade model utility, while fine-tuning may fail to recover the utility given the limited retain set.

**Questions:**

Please see the Weaknesses section for all questions and clarification requests.

---

> ### Author Response · Authors · 2025-11-28
>
> We thank the reviewer for their constructive feedback. We are encouraged that the reviewer finds our PBU setting “novel and well-motivated”, appreciates RELOAD’s “strong results across benchmarks”, and that the method is “efficient”.
> Below, we hope to address all specific questions and concerns.
> # Weaknesses
> > The paper does not provide a theoretical justification for why a single gradient-ascent update using and selective re-initialization of low-KV weights can remove the influence of the forget data. As a result, it is unclear when RELOAD succeeds or fails beyond the reported scenarios.
>
> We appreciate you raising this point. We have expanded Section 2.3 in the revised manuscript to provide a more thorough explanation of the effect produced due to the interaction of these components.
>
> Crucially, our ablation studies in Appendix C.2 and C.3 confirm that these components are non-redundant and essential:
> - Ablation without Ascent: We demonstrate that running RELOAD without the specific gradient ascent step fails to sufficiently distance the model from the forget set, resulting in poor unlearning performance.
> - Threshold Sensitivity: We evaluated various selective re-initialization thresholds. The results show that deviations from our calculated low-KV quantile either fail to remove the sensitive information (if too few weights are re-initialized) or significantly degrade model utility (if too many are re-initialized).
>
> These results confirm that the combination of targeted ascent and specific low-KV re-initialization is strictly necessary for the algorithm's success.
>
> > The paper claims RELOAD “allow user data to be immediately removed when a request for deletion is made, eliminating the continued accumulation of dataset risk,” but RELOAD requires the retain set to compute gradients and to perform fine-tuning. This means that the retained data is still at risk, and the ideal method should not use any training data during the unlearning process.
>
> We acknowledge the reviewer’s point that the retain set is utilized during fine-tuning. However, we emphasize a critical distinction regarding the nature of the risk and consent:
> The users comprising the retain set have not requested deletion; therefore, their data remains valid for training purposes under current usage agreements.
>
> Targeted Liability Reduction: The "risk" RELOAD specifically targets is the legal and ethical liability associated with holding data after consent has been revoked (the forget set). Practitioners are under strict obligations to remove this specific subset of data immediately.
>
> While we agree with the reviewer that an idealized method would use zero data, RELOAD is designed for the practical reality of Partially-Blind Unlearning (PBU). It allows us to focus entirely on eliminating the influence of the data we are legally obligated to remove, thereby strengthening data security for the specific subset where practitioners face the highest liability if a breach or retention violation were to occur.

---

> > ### Author Response · Authors · 2025-11-28
> >
> > > The robustness of RELOAD across different numbers of requests is unclear. When the forget set is extremely small (e.g., unlearning only one data point), becomes a very small residual between two nearly identical large gradients; this makes the ascent step near zero and may fail to truly forget that sample. In addition, when the forget request is large (e.g., 30% of the data), the ascent and re-initialization would degrade model utility, while fine-tuning may fail to recover the utility given the limited retain set.
> >
> > Thank you for raising this observation! RELOAD is designed to handle this specifically through its selection mechanism. Our selection of weights for re-initialization is based on a quantile of the Knowledge Value (KV), not an absolute magnitude threshold. Even if the residual gradients are small, the algorithm still identifies the weights with the lowest relative value for re-initialization. To prevent numerical errors or vanishing gradients from rendering the ascent step ineffective in these small-batch scenarios, we utilize a Laplace smoothing constant.
> >
> > As evidence of this robustness, we highlight our results to our experiments on 100-sample forgetting (representing approx. 0.22% of the dataset). In these cases, RELOAD successfully removed the influence of the specific samples without the failure modes suggested.
> >
> > Furthermore, we performed extensive experiments simulating large-scale unlearning requests (up to 30% of the dataset), reported in Appendix B.4.2. The results demonstrate that RELOAD effectively repairs model utility even after such significant deletions. Because the re-initialization is selective, the "core" weights representing the general distribution (the top-KV weights) remain largely intact. The subsequent fine-tuning step is then sufficient to realign the model to the remaining 70% of the data. The experimental data confirms that utility is recovered to levels comparable to retraining from scratch, validating the method's stability even under high-volume unlearning requests.
> >
> > # Final remarks
> > Thank you once again for your helpful feedback. We hope that our rebuttal addresses your questions and concerns, and we kindly ask that you consider a fresher evaluation of our paper if you are satisfied with our responses. We are also more than happy to answer any further questions that arise.

---

### Official Review · Reviewer_VSdc · 2025-10-31

**Soundness:** 2
**Presentation:** 3
**Contribution:** 2
**Rating:** 4
**Confidence:** 3

**Summary:**

This paper addresses a significant limitation in existing machine unlearning techniques, namely the requirement to retain access to the “forget set” (the data to be removed) during the unlearning process. The authors introduce the concept of partially-blind unlearning (PBU) and propose RELOAD, a framework that performs unlearning without direct access to the forget set. RELOAD leverages cached gradients from the final epoch of training and combines gradient ascent, selective weight reinitialization, and fine-tuning on retained data. The paper claims strong empirical results showing that RELOAD can efficiently approximate retraining from scratch and even outperform some forget set-dependent approaches, including applications to both image classification and large language models.

**Strengths:**

* The problem formulation of unlearning without the forget set is timely, novel, and practically relevant for privacy compliance (e.g., GDPR).

* Methodologically sound integration of gradient ascent, weight reinitialization, and fine-tuning into a coherent framework.

* Empirical results show promising efficiency and competitive performance, especially for large models such as Llama2-7B.

**Weaknesses:**

* Limited robustness to model updates

The method assumes availability of final-epoch gradients representing the entire training data. In real-world pipelines where models are fine-tuned or continuously updated, these cached gradients may no longer capture the forget set’s influence, reducing unlearning effectiveness. Evaluating RELOAD under fine-tuning or continual learning scenarios is necessary.

* Unclear privacy guarantees

Although RELOAD is “partially blind,” cached gradients can still leak sensitive information. Prior works (e.g., Geiping et al., 2020) show that gradient inversion can reconstruct data. The paper lacks a quantitative privacy leakage analysis to support its safety claims.

* Limited ablation and sensitivity study

While some ablations are included, deeper exploration of how hyperparameters (e.g., ascent rate, reset proportion) affect performance is missing. This limits confidence in robustness across settings.

* Storage overhead

The approach removes the need to store the forget set but introduces the requirement to store full-model gradients, which can be large for modern networks. The practical feasibility of gradient caching is not analyzed.

**Questions:**

1. How would RELOAD perform if the model undergoes fine-tuning or incremental updates after initial training?

2. What is the approximate storage cost of ∇θL(D) for large models such as Llama2-7B, and could this be mitigated via gradient compression?

3. Have the authors tested for information leakage from cached gradients using existing gradient inversion techniques?

4. How does RELOAD handle overlapping or correlated data between forget and retain sets?

5. Can the method scale to federated or distributed training settings where only local gradients are available?

---

> ### Author Response · Authors · 2025-11-28
>
> We thank reviewer VSdc for their thoughtful assessment. We are encouraged that the reviewer finds our PBU formulation “timely, novel, and practically relevant for privacy compliance”, our method a “coherent framework”, and that our empirical results show “promising efficiency and competitive performance”. Below, we hope to address all concerns raised.
>
> # Weaknesses
> 1. Robustness to Model Updates (Fine-tuning & Continual Learning)
>
> > The method assumes availability of final-epoch gradients [...] Evaluating RELOAD under fine-tuning or continual learning scenarios is necessary.
>
> Thank you for this insight. We agree that theoretically, significant parameter changes during fine-tuning could reduce the fidelity of the cached gradients relative to the current model state. Ideally, gradients should be recomputed post fine-tuning/continual learning on the whole dataset to retain theoretical soundness.
>
> To investigate the practicality of applying RELOAD after fine-tuning, we conducted a new ablation simulating a fine-tuning scenario leveraging CIFAR-10 and CIFAR-10.1 as follows. A ResNet-18 model is trained on CIFAR-10, achieving 99.76% test accuracy while reaching 30.8% accuracy on CIFAR-10.1 test. The model is then fine-tuned on CIFAR-10.1, reaching 99.2% accuracy on CIFAR-10.1 test.
>
> We report in the table below results of RELOAD in scenarios: 1. Unlearning samples from CIFAR-10 after fine-tuning, 2. Unlearning samples from CIFAR-10.1 after finetuning, and 3. Unlearning samples from both CIFAR-10 and 10.1 after finetuning. The full experimental details for these experiments are in the revised manuscript in Appendix C.6.
>
> ### Table 1: 10% Random Forgetting from CIFAR-10 after finetuning on CIFAR-10.1 (ResNet-18)
>
> | Method              | RA (↑)            | ΔFA (↓)             | ΔFE (↓)            | ΔFMIA (↓)         | RSKL (↓)         | FSKL (↓)         |
> |---------------------|-------------------|----------------------|---------------------|--------------------|-------------------|-------------------|
> | **Reload (ours)**   | 98.16± 0.01    | 0.57± 0.46        | 0.06± 0.03      | 0.01± 0.01      | 0.91± 0.02    | 3.45± 0.06    |
> | **Retrained (Baseline)** | 92.34± 0.6 | 34.57± 0.00       | 2.33± 0.00       | 0.51± 0.01      | -                 | -                 |
>
>
> ### Table 2: 10% Random Forgetting from CIFAR-10.1 after finetuning on CIFAR-10.1 (ResNet-18)
>
> | Method              | RA (↑)            | ΔFA (↓)             | ΔFE (↓)            | ΔFMIA (↓)         | RSKL (↓)         | FSKL (↓)         |
> |---------------------|-------------------|----------------------|---------------------|--------------------|-------------------|-------------------|
> | **Reload (ours)**   | 98.89± 0.19    | 2.85± 2.05        | 0.20± 0.13       | 0.05± 0.04      | 0.51± 0.02     | 3.95± 0.27     |
> | **Retrained (Baseline)** | 99.73± 0.11 | 32.4± 2.78       | 2.70± 0.11       | 0.54± 0.05      | -                 | -                 |
>
>
> ### Table 3: 10% Random Forgetting from CIFAR-10 and CIFAR-10.1 after finetuning on CIFAR-10.1 (ResNet-18)
>
> | Method              | RA (↑)            | ΔFA (↓)             | ΔFE (↓)            | ΔFMIA (↓)         | RSKL (↓)         | FSKL (↓)         |
> |---------------------|-------------------|----------------------|---------------------|--------------------|-------------------|-------------------|
> | **Reload (ours)**   | 99.69± 0.13    | 3.88± 1.52        | 0.27± 0.11       | 0.02± 0.01      | 1.04± 0.13     | 3.97± 0.15     |
> | **Retrained (Baseline)** | 89.39± 2.89 | 35.83± 0.93      | 2.34± 0.05       | 0.52± 0.02      | -                 | -                 |
>
>
> We observe that across 10 seeds, RELOAD exhibits stronger performance across key metrics with respect to the retrain baseline on all three experiments. This suggests that the continual caching of gradients post fine-tuning proves RELOAD effective in scenarios where the base model is fine-tuned on distributionally shifted data.
>
> An inherent requirement for this robustness is that gradients must be re-computed across the cumulative dataset (original + fine-tuning data) after significant model updates. While this necessitates continual storage of the training set during the model's active lifecycle, we argue this aligns with standard industry operations wherein data holders typically preserve their full datasets for versioning, debugging, and compliance until a specific deletion request is received.
> > Although RELOAD is “partially blind,” cached gradients can still leak sensitive information. Prior works (e.g., Geiping et al., 2020) show that gradient inversion can reconstruct data. The paper lacks a quantitative privacy leakage analysis to support its safety claims.

---

> ### Author Response · Authors · 2025-11-28
>
> > Have the authors tested for information leakage from cached gradients using existing gradient inversion techniques?
>
> We agree that this is a critical point. As is discussed in Section 2.3 on partial blindness, including Geiping et al. 2020, successful inversion attacks rely on knowing or approximating the exact cardinality of the batch used for aggregation. In the PBU setting, the adversary lacks knowledge of the precise size of the forget set being removed, a crucial prerequisite for existing methodologies. Furthermore, these works highlight that reconstruction fidelity rapidly degrades with increasing batch size, yielding often unrecognizable results even for small aggregations. Given that RELOAD operates on gradients aggregated over the entire, massive training set D, the signal contributed by any individual forgotten data point is so diluted that it renders existing reconstruction techniques ineffective.
>
> > While some ablations are included, deeper exploration of how hyperparameters (e.g., ascent rate, reset proportion) affect performance is missing. This limits confidence in robustness across settings.
>
> We appreciate the desire for a thorough robustness analysis. We wish to clarify that a comprehensive set of ablation and sensitivity studies covering all critical components and hyperparameters of RELOAD is already included in our manuscript, specifically in Appendix C.2 and C.3.
> In Appendix C.2, we analyze the contribution and sensitivity of each core component of the RELOAD framework (gradient ascent, normalisation, cosine similarity to compute knowledge values). In Appendix C.3, we provide a detailed sensitivity study on key hyperparameters, including the learning rate and the reset proportion.
> We will enhance the discussion and thoroughly articulate the criticality of these components with respect to the detailed ablation results presented in the Appendix.
>
> # Questions
> > How would RELOAD perform if the model undergoes fine-tuning or incremental updates after initial training?
> Please see the above additional experiments on CIFAR-10 and CIFAR-10.1 fine-tuning.
> > What is the approximate storage cost of ∇θL(D) for large models such as Llama2-7B, and could this be mitigated via gradient compression?
>
> For the entity unlearning experiments designed for language models such as Llama2-7B, the setup does not require caching gradients. Instead, we borrow the core mechanistic insights of RELOAD to design an efficient, gradient-free prompting solution that effectively unlearns targeted entities without the need for additional storage overhead.
> We sincerely thank the reviewer for the insightful suggestion regarding gradient compression as a mitigation strategy for the storage cost in non-LLM architectures. We have acted on this by conducting new experiments evaluating the effect of gradient quantization.
> Specifically, we experimented with storing the cached gradients ∇θ​L(D) in float16 instead of float32, and then inflating back to float32 at RELOAD runtime, halving the storage cost of stored gradients.
> As shown below, performance deterioration across key unlearning and utility metrics is minimal and generally within the standard deviation of the full-precision results. The full experimental details are in the revised manuscript in Section 3.5.
> ### Table: 10% Random Forgetting on CIFAR-10 (ResNet-18) with Quantized Cached Gradients
>
> | Method                   | RA (↑)            | ΔFA (↓)             | ΔFE (↓)            | ΔFMIA (↓)         | RSKL (↓)         | FSKL (↓)         |
> |--------------------------|-------------------|----------------------|---------------------|--------------------|-------------------|-------------------|
> | **Reload (Un-Quantized)** | 99.99± 0.01    | 0.46± 0.57        | 0.76± 0.08       | 0.01± 0.01      | 0.44± 0.03     | 4.04± 0.1      |
> | **Reload (Quantized)**   | 99.99± 0.01    | 0.46± 0.57        | 0.76± 0.08       | 0.01± 0.01      | 0.44± 0.03     | 4.04± 0.1      |
> | **Retrained (Baseline)** | 99.52± 0.16    | 36.22± 0.49       | 2.25± 0.03       | 0.51± 0.01      | -                 | -                 |
>
>
> These results indicate that gradient quantization is a highly effective and low-cost solution for mitigating the storage overhead of RELOAD in models where full gradients are required, affirming the reviewer's foresight on this approach.

---

> > ### Author Response · Authors · 2025-11-28
> >
> > > How does RELOAD handle overlapping or correlated data between forget and retain sets?
> >
> >
> > We appreciate the opportunity to clarify this crucial aspect of RELOAD's robustness. We specifically address this scenario in Section 3.2.
> > In these experiments, we simulate a strong case of correlation by using RELOAD to unlearn an entire batch of samples belonging to the same class, ensuring the retain set contains numerous examples of the same class (high correlation).
> > As detailed in the discussion and corresponding tables in Section 3.2, RELOAD demonstrates high efficacy in this challenging setting. Across key metrics, our method closely approximates the retraining baseline, achieving the lowest ΔFMIA and FSKL across all methods. This confirms that RELOAD successfully isolates the influence of the specific forgotten samples without catastrophically degrading the utility derived from highly correlated retained data.
> > > Can the method scale to federated or distributed training settings where only local gradients are available?
> >
> >
> > We sincerely thank the reviewer for this forward-thinking suggestion; adapting RELOAD to Federated Learning (FL) environments presents a highly promising and novel research direction.
> > RELOAD is naturally suited for FL since it uses gradients, the primary information exchanged. We envision a process involving Local Unlearning followed by Global Coordination. Consider a FL example where hospitals deploy the same (FL) trained model across all hospital data:
> > Local Gradient Caching: Each hospital caches the aggregate gradients of its own local dataset.
> > Local Unlearning: When a patient submits a "right to be forgotten" request, the hospital executes a Local RELOAD using only its locally cached gradients, successfully removing the patient's influence from its local model.
> > Model Coordination: Due to divergent local unlearning requests, models will drift. To ensure global compliance and prevent data reconstruction risks from stale parameters, a periodic Coordinated RELOAD or aggregation of local unlearned updates would be necessary to generate and redistribute a new, globally compliant model.
> > Although this distributed extension has not been empirically explored, the concept of leveraging local cached gradients for efficient local unlearning, followed by coordinated global updates, provides a clear and theoretically sound pathway for applying RELOAD to large-scale, privacy-sensitive FL environments. We plan to pursue this in future work.
> > # Final remarks
> > We thank the reviewer once again for their helpful feedback. We hope that our rebuttal addresses their questions and concerns, and we kindly ask that they consider a fresher evaluation of our paper if they are satisfied with our responses. We are also more than happy to answer any further questions that arise.

---

### Author Response · Authors · 2025-11-28

We thank all reviewers for their insightful and constructive feedback. We are encouraged to hear that they found our work novel (VSdc, WbA5, dom5), practically relevant for privacy with a strong motivation (VSdc, 2ZVC), and appreciated the coherence of our overall framework (VSdc, 2ZVC, dom5). We are also pleased that reviewers highlighted our promising efficiency and strong empirical results (VSdc, WbA5, 2ZVC). We appreciate the reviewers’ recognition and now provide a unified response addressing their main concerns through additional clarification and new analyses.

# Clarifying the Theoretical Basis and Practical Validity of RELOAD (WbA5, 2ZVC, dom5)

While we do not provide closed-form bounds on residual influence, RELOAD’s effectiveness can be understood and validated both conceptually and empirically.
Gradient-based unlearning mechanism: RELOAD approximates ∇L(D_forget) by subtracting the retained-set gradient from the full-training gradient (∇L(D) − ∇L(D_retain)), under the standard assumption of additively computed losses across samples. This assumption holds for most supervised losses and LayerNorm-based architectures, including Transformers and Vision Transformers, as validated in our experiments. For BatchNorm-based models, minor approximation errors occur but do not materially affect unlearning performance.


Necessity of ascent and selective reinitialization: Ablation studies (Appendix C.2–C.3) demonstrate that both components are essential. Without the gradient ascent step, RELOAD fails to sufficiently remove the influence of the forget set. Similarly, choosing a low-KV reinitialization threshold that is too high or too low either retains sensitive information or degrades model utility.


Empirical validation as a practical proxy: In the absence of formal theoretical bounds, we measure the effectiveness of RELOAD by comparing it to the gold-standard retrain baseline (training from scratch on the retained dataset). Specifically, ΔMIA quantifies the difference in susceptibility to membership inference attacks between RELOAD and the retrained model. Our results consistently show negligible ΔMIA, indicating that RELOAD successfully removes the influence of the forget set to a degree comparable with full retraining.
Together, these points explain why RELOAD works across diverse architectures and support the robustness of our method, while acknowledging that formal theoretical guarantees remain an exciting avenue for future work.

# Practical Considerations: Gradient Storage, Efficiency, and Targeted Risk (VSdc, 2ZVC)

We thank the reviewers for raising concerns regarding the storage, robustness, and privacy implications of caching gradients in RELOAD. For models where full gradients are required, we demonstrate that storage overhead can be effectively mitigated through gradient quantization. In our experiments on ResNet-18, storing ∇θL(D) in FP16 rather than FP32 halves the memory footprint while producing negligible differences in unlearning and utility metrics, remaining well within the standard deviation of full-precision results, we report our results in the table below. This highlights that RELOAD’s gradient caching is practical and deployable even for moderately large models.
### Table: 10% Random Forgetting on CIFAR-10 (ResNet-18) with Quantized Cached Gradients

| Method                   | RA (↑)            | ΔFA (↓)             | ΔFE (↓)            | ΔFMIA (↓)         | RSKL (↓)         | FSKL (↓)         |
|--------------------------|-------------------|----------------------|---------------------|--------------------|-------------------|-------------------|
| **Reload (Un-Quantized)** | 99.99± 0.01    | 0.46± 0.57       | 0.76± 0.08       | 0.01± 0.01      | 0.44± 0.03    | 4.04± 0.10      |
| **Reload (Quantized)**   | 99.99± 0.01    | 0.46± 0.57        | 0.76± 0.08      | 0.01± 0.01    | 0.44± 0.03     | 4.04± 0.1     |
| **Retrained (Baseline)** | 99.52± 0.16   | 36.22± 0.49       | 2.25± 0.03       | 0.51± 0.01     | -                 | -                 |

For large language models such as Llama2-7B, our entity unlearning experiments do not require storing gradients at all. Instead, we leverage RELOAD’s mechanistic insights to design a gradient-free, efficient prompting approach, eliminating any additional storage overhead . Finally, regarding privacy and data risk, RELOAD specifically targets the “forget set” — the subset of data for which consent has been revoked — while the retained data remains valid for training (in particular, for RELOAD finetuning). This ensures that legal and ethical obligations are addressed immediately, reducing liability while focusing resources on the highest-risk subset. Together, these results confirm that RELOAD is both practical and effective across a range of architectures and deployment scenarios.

---

> ### Author Response · Authors · 2025-11-28
>
> # Final remarks
> Changes and edits are now reflected throughout the manuscript and color-highlighted, including new ablations and edits suggested during this review process.
>
> We hope these clarifications and additional analyses help reviewers appreciate the empirical rigor, feasibility, and societal relevance of RELOAD. Once again, we sincerely thank all reviewers for their feedback as it enabled us to improve the manuscript with additional results, ablations, and clarifications that strengthen the contributions of our work. We kindly ask that the reviewers consider a fresher evaluation of our paper if they are satisfied with our responses, and are more than happy to answer any further questions.

---

### Meta-Review · Area_Chair_Meat · 2026-01-06

**Summary:**

The submission presents a novel unlearning scheme that combines existing techniques (specifically, gradient ascent on the forget set gradient, selective weight reinitialization, and fine-tuning on the retained set) to perform unlearning without the use of the forget set. The empirical results show extremely strong performance compared to existing unlearning baselines. The main concerns raised by the reviewers were (C1) the computational overhead and the privacy risk of caching loss gradients over the whole training set, (C2) the lack of any theoretical justification for the proposed scheme, and (C3) the fact that the proposed RELOAD scheme is a simple recombination of existing unlearning schemes. In my opinion, the authors provide reasonable responses to C1, while acknowledging C2 and C3, but highlighting that the submission provides a significant amount empirical justification for the proposed scheme while demonstrating the non-trivial aspect of the recombination via ablations.

Overall, I recommend that this submission be accepted.

However, I would like to bring up one concern (not brought up by any of the reviewers) which I hope the authors will consider in future revisions:
In usual unlearning scenarios, while we assume access to the forget set (which this submission avoids), we usually do not require access to the full retain set, and usually a subset of the retain set is utilized. The way RELOAD is developed (the forget set gradient is computed by using the full training data gradient and the full retain set gradient), it is not clear whether RELOAD will be as successful if only a subset of the retain set is available. It would be helpful to a reader to properly understand this.

**Reviewer Concerns:**

The following concerns were raised by the reviewers:

- Reviewers raised concerns regarding storage of "final-epoch gradients" especially when models are continually trained (such as being fine-tuned for specific tasks) and it is not clear what the "final-epoch gradients" are.
  - The authors provide new experiments in a continual fine-tuning setup.
  - However, it requires the recomputation of the gradients over the whole training set after the model has been fine-tuned, which makes it less practical.

- Storing gradients for complete training data might have significant computational overhead.
  - The authors respond by providing additional results where the gradients are quantized for storage.

- If gradients are stored for the unlearning period, the privacy risk from gradient inversion attacks are not appropriately discussed.
  - The authors claim that batch aggregation makes it hard for privacy attacks on gradients to succeed.

- It is weird that FT by itself is not included in Table 1 especially since the FT phase of RELOAD is fine-tuned on the retained set until convergence. Also, for the results in the appendix with FT, it is not clear if there is some form of parity maintained for the number of fine-tuning steps between FT and RELOAD.
  - One can consider the ablation removing ascent step and selective reinitialization as the FT baseline here. But I do not think that is included here.
  - The results in the response for 2ZVC seem to show that FT has very close performance to RELOAD on teh Census Income data. This seems to bring into question the utility of the ascent and reinitialization steps.
  - Furthermore, a critical ablation of using varying fraction of the retain set is missing, answering the question of whether we need the whole retain set for the RELOAD algorithm.

- Assumption for decomposable loss functions, and thus not applicable to models trained with contrastive losses or with (batch) normalization.
  - The authors respond by highlighting that most losses are decomposable, and the proposed RELOAD cannot be trivially applied to models with batch normalization or models trained with contrastive losses.

- Limited ablations of newly introduced hyperparameters.
  - The authors respond by saying that ablations are already present in the appendix.

- Lack of any theoretical justification for the proposed RELOAD scheme, especially justifying the different steps (such as ascent and selective reinitialization).
  - The authors acknowledge this limitation but argue that their thorough empirical performance and careful ablations provide ample justification for the steps in RELOAD.

**Reviewer Scores:**

In my opinion, all reviewer concerns were adequately addressed, and all reviewers would raise their current scores (6, 4, 6, 4) by at least 1.

---

### Decision · Program_Chairs · 2026-01-26

Accept (Poster)